# Technological evolution of large-scale blue hydrogen production toward the U.S. Hydrogen Energy Earthshot

Wanying Wu [1], Haibo Zhai [1,2,3] ✉ & Eugene Holubnyak[2]

Hydrogen potentially has a crucial role in the U.S. transition to a net-zero emissions economy. Learning from large-scale hydrogen projects will boost technological evolution and innovation toward the U.S. Hydrogen Energy Earthshot. We apply experience curves to estimate the evolving costs of blue hydrogen production and to further examine the economic effect on technological evolution of the Inflation Reduction Act's tax credits for carbon sequestration and clean hydrogen. Learning-by-doing alone can decrease the production cost of blue hydrogen. Without tax incentives, however, it is hard for blue hydrogen production to reach the cost target of $1/kg $H_2$. Here we show that the breakeven cumulative production capacity required for gas-based blue hydrogen to reach the $1/kg $H_2$ target highly depends on tax credit, natural gas price, inflation rate, and learning rates. We make recommendations for hydrogen hub development and for accelerating technological progress toward the Hydrogen Energy Earthshot.

Clean hydrogen has the potential to help achieve 10% economy-wide emissions reductions by 2050 relative to 2005, promote energy security and resilience, and develop a new economy in the United States[1]. In 2030, the hydrogen economy could create about 100,000 net new jobs for the development of new capital projects and clean hydrogen infrastructure[2]. The U.S. Bipartisan Infrastructure Law has appropriated $9.5 billion for clean hydrogen for the U.S. Department of Energy (DOE)[1]. Both zero- and low-carbon hydrogen production technologies are key options in a diverse toolbox enabling the transition to a sustainable and equitable clean energy future[1]. In October 2023, the U.S. DOE announced $7 billion to launch seven Regional Clean Hydrogen Hubs across the nation[3]. Some regional hubs will use water and natural gas as the feedstock for renewable-powered electrolysis and steam methane reforming (SMR) with carbon capture and storage (CCS) to produce clean hydrogen, which is also called green and blue hydrogen in practice, respectively. Blue hydrogen is often viewed as a near-term bridge to a zero-carbon hydrogen economy. Given potential high methane leakage, however, there are scientific debates on the competitiveness of blue hydrogen[4,5], which makes a serious call for methane abatement.

The U.S. is making significant efforts to accelerate progress through historic investments and additional policies and incentives for clean hydrogen[1]. The U.S. National Clean Hydrogen Strategy and Roadmap has outlined strategic pathways for annual clean hydrogen production of 10 million metric tons (MMT) by 2030, 20 MMT by 2040, and 50 MMT by 2050 in the U.S.[1]. Although renewable-powered green hydrogen has much less carbon emissions than blue hydrogen, the current cost of green hydrogen production can be several times higher[6,7], as shown later. Fossil fuel-based hydrogen production with CCS or blue hydrogen is among a portfolio of pathways for clean hydrogen. In 2021, the U.S. DOE launched the Energy Earthshots Initiative that aims to accelerate breakthroughs of more abundant, affordable, and reliable clean energy solutions by 2030[8]. To catalyze technological innovation and scale in clean hydrogen, this initiative included a hydrogen shot that aims to decrease the cost of clean hydrogen to $1 per 1 kilogram in 1 decade[8], which is called the Hydrogen Energy Earthshot.

The Inflation Reduction Act (IRA) of 2022 provides a set of tax credits to stimulate the deployment of clean hydrogen technologies[9]. The IRA contains two incentive provisions: a new Section 45 V Tax

[1]College of Engineering and Physical Sciences, University of Wyoming, Laramie, WY, USA. [2]School of Energy Resources, University of Wyoming, Laramie, WY, USA. [3]Department of Engineering and Public Policy, Carnegie Mellon University, Pittsburgh, PA, USA. ✉e-mail: hzhai@uwyo.edu

Credit for clean hydrogen production and an enhanced Section 45Q Tax Credit for carbon sequestration[9]. The 45 V tax credit is available for hydrogen projects with life cycle greenhouse gas (GHG) emissions of less than 4.0 kilograms of carbon dioxide ($CO_2$) equivalent per kilogram of hydrogen during the 10-year period and ranges from $0.6 to 3.0 per kilogram of hydrogen. The production credit varies with the level of life cycle GHG emissions. In addition, the IRA has enhanced the 45Q tax credit to $85 per metric ton of $CO_2$ stored in saline reservoirs and $60 per metric ton of $CO_2$ used for enhanced oil recovery (EOR) or other industrial applications for up to twelve years. The sequestration threshold required for eligible CCS projects has been lowered to 18,750 metric tons of $CO_2$ per year for power plants and 12,500 metric tons per year for other facilities. These tax incentives would facilitate large-scale blue hydrogen production.

In 2020, global hydrogen production reached 90 million metric tons per annum (MMTA), of which 72 MMTA were pure hydrogen[10], while the U.S. hydrogen production reached about 10 MMTA[11]. Globally, SMR and coal gasification without carbon abatement accounted for 74% and 24% of the pure hydrogen production[10], respectively. Similarly, they accounted for 99% of the U.S. hydrogen production in 2020, of which 95% was made by SMR[11]. SMR and coal gasification can be coupled with CCS to produce low-carbon hydrogen. However, less than 1% of the global hydrogen is produced currently from fossil fuel resources with CCS[12]. Table 1 summarizes fossil fuel-based blue hydrogen projects installed around the world. Globally, these projects produced 0.7 MMTA of blue hydrogen from gas, coal, and oil resources and captured 10 MMTA of $CO_2$ in 2020[10]. In the U.S., the blue hydrogen projects produced 0.23 MMTA of hydrogen in 2021[10].

The U.S. DOE's National Energy Technology Laboratory (NETL) has provided a comparative assessment of the performance and cost of state-of-the-art fossil fuel-based hydrogen production technologies, including reforming and gasification without and with CCS[7]. The levelized cost of hydrogen production without carbon abatement ranges roughly from $1.1 to 2.6 per kilogram of hydrogen and varies with feedstock type or production technology. The addition of CCS for low-carbon hydrogen increases the levelized cost by more than 50% for the reforming production and 20% for the gasification production[7]. The largest contributor to the levelized cost is the fuel cost for the reforming production, whereas it is the capital cost for the gasification production[7].

The production process of blue hydrogen often involves multiple subsystems, such as SMR and CCS. Future costs of individual subsystems will likely decline as a result of learning-by-doing as they scale up. However, the progress in learning may vary by subsystem. Schoots et al.[13] analyzed hydrogen cost data observed during the period of 1940 to 2007 and found that the learning rate of SMR, defined as the fractional reduction in cost for each doubling of cumulative installed capacity, is 11 ± 6% in the investment, but there is no cost reduction in the overall production cost, which implies a zero-learning rate in operating and maintenance (O&M) costs. Rubin et al.[14] and IEAGHG[15] reported that the learning rate of gasification is 14% for capital cost and 12% for O&M cost. CCS is an essential technology for blue hydrogen production. Rubin et al.[14] reported that the learning rate of a carbon capture system is 11% for capital cost and 22% for O&M cost. Deployment of large-scale CCS projects will decrease the future cost of carbon capture for advanced technologies[16]. Although a large number of learning curve studies have been done in the past years, almost no breakthroughs have been made in estimating the learning rates for SMR, gasification, and CCS beyond the two pioneering studies by Schoots et al.[13] and Rubin et al.[14]. Recent studies frequently adopted the learning rates from the two pioneering studies for a variety of applications[17-25]. However, few studies provide a thorough assessment of the overall technological learning of blue hydrogen produced from fossil fuel resources and examine the potential effect of tax incentives on technological evolution.

Projections of technological evolution are crucial to research and development in advanced technology, market analysis and forecast, investment analysis, resource planning, and decision- and policy-

**Table 1 | Operational blue hydrogen production facilities**[a]

| Feedstock | Project Name | Country | Online Year | H₂ Production Capacity | | Captured $CO_2$ (million metric tons/year) |
|---|---|---|---|---|---|---|
| | | | | (m³/hour)[b] | (10³ · metric tons/year) | |
| Natural Gas | PCS Nitrogen | United States | 2013 | 31,344 | N/A | 0.25 |
| | Port Arthur | United States | 2013 | 125,376 | 118 | 1.00 |
| | Enid Fertilizer | United States | 1982 | 87,764 | N/A | 0.70 |
| | Port Jerome | France | 2015 | 12,538 | 39 | 0.10 |
| | Quest | Canada | 2015 | 125,376 | 300 | 1.00 |
| | Nutrien (Former Agrium) Fertilizer | Canada | 2020 | 37,613 | N/A | 0.30 |
| | Al Reyadah CCUS | United Arab Emirates | 2016 | 47,876 | N/A | 0.80 |
| Coal | Sinopec Qilu Petrochemical CCS | China | 2022 | 41,892 | N/A | 0.70 |
| | Sinopec Zhongyuan Oilfield EOR | China | 2015 | 5985 | N/A | 0.10 |
| | Changqing Oil Field EOR | China | 2015 | 2992 | N/A | 0.05 |
| | Great Plains Synfuel Plant and Weyburn-Midale | United States | 2000 | 179,536 | N/A | 3.00 |
| Oil | Coffeyville Fertilizer Plant | United States | 2013 | 125,376 | N/A | 1.00 |
| | Shell Heavy Residue Gasification CCU - Pernis Refinery[c] | Netherlands | 2005 | 23,938 | 1,000 | 0.40 |
| | Karamay Dunhua Oil Technology CCUS EOR Project | China | 2015 | 5985 | N/A | 0.10 |
| | North West Sturgeon Refinery | Canada | 2020 | 77,799 | N/A | 1.30 |
| | Horizon Oil Sands | Canada | 2009[d] | 26,212 | N/A | 0.44 |

[a]The sources of data: IEA[28,31].
[b]The hydrogen production capacity can be converted to the mass-based capacity based on the hydrogen density of 0.089 kg/m³ (IEA[28]).
[c]The carbon capture and utilization project will be converted to a carbon capture, utilization, and storage project in 2024 (IEA[28]).
[d]The source of data: IEA[31].

making. The major objectives of this study are to estimate the future evolving costs of major blue hydrogen production pathways from large-scale deployment, including state-of-the-art SMR and coal gasification with CCS, and to examine the economic effect on the technological evolution of the IRA's tax credits for clean hydrogen. This study evaluates both the 45Q and 45 V tax credits and compares their economic role in promoting blue hydrogen production toward the Hydrogen Energy Earthshot. This study demonstrates how learning-by-doing will lower the cost of blue hydrogen production in the future and how much capacity of blue hydrogen projects should be installed to reach a cost target. This study offers an outlook for large-scale blue hydrogen production from abated fossil resources and reveals its potential policy-driven evolutionary trends, including their dependence on key factors.

## Results

This study first characterizes greenhouse gas emissions and costs of commercial technologies for blue hydrogen production and then develops technological learning and diffusion models to assess the future costs and evolutionary trajectories of blue hydrogen production without and with tax incentives toward the U.S. Hydrogen Energy Earthshot. A series of parametric analyses are further performed to reveal the dependence of the overall hydrogen production cost on key factors, such as fuel price, carbon capture cost uncertainties, learning rates, and inflation rate.

### Current blue hydrogen production

This study adopts state-of-the-art reforming and gasification technologies as a point of reference to explore the evolutionary trends of blue hydrogen production driven by learning-by-doing. The current performance and cost estimates of these technologies are obtained from the recent NETL study[7]. The majority of hydrogen produced in the U.S. is made via steam methane reforming (SMR). In addition, the cost of blue hydrogen produced by SMR with carbon capture and storage (CCS) is similar to that by autothermal reforming with CCS, but the on-site and life-cycle emissions from the SMR process are less[7]. This study, therefore, focuses on SMR with CCS for gas-based blue hydrogen production. In the meantime, an oxygen-blown, entrained-flow Shell-type gasifier is employed with CCS for coal-based blue hydrogen production[7]. Supplementary Table 1 in Supplementary Note 1 summarizes the major techno-economic parameters and assumptions made for blue hydrogen production plants using natural gas and coal resources as the feedstocks, which include the project book lifetime, capacity factor, hourly production capacity, fuel price, and fixed charge rate. In addition, the land and water footprints per unit of hydrogen produced by these plants and the amounts of $CO_2$ sequestration are also reported in Supplementary Table 2. Blue hydrogen plants produce high-purity hydrogen (99.9 vol.%) at the pressure of 6.48 MPa and transport the captured $CO_2$ at the pressure of 15.3 MPa for storage in saline reservoirs, which are typical design conditions. This study reports the cost results in 2018 U.S. real dollars unless otherwise noted.

For the given assumptions, the total levelized cost of hydrogen (LCOH) is $1.64/kg $H_2$ for SMR with CCS and $3.09/kg $H_2$ for coal gasification with CCS. In comparison, the plant LCOH is 88.4% higher for gasification production than the reforming production, which indicates that the integration of SMR with CCS is much more competitive for blue hydrogen. In addition, the on-site stack $CO_2$ emissions from hydrogen production by SMR with CCS are 0.4 kg $CO_2$/kg $H_2$, which is much less than that (1.4 kg $CO_2$/kg $H_2$) from gasification with CCS[7]. In addition to the stack $CO_2$ emissions, there may be fugitive GHG emissions from various sources at an SMR production plant, mainly from the piping equipment and fittings[26]. However, fugitive GHG emissions are about 0.05% of the stack GHG emissions[26], which indicates that plant methane leakage is not a serious issue.

A hydrogen production plant is decomposed into major subsystems. The components included in each of the subsystems defined for gas- and coal-based production plants are reported in Supplementary Tables 3 and 4, respectively. Figure 1a, b show the distribution by subsystem in the plant LCOH for the two production plants, respectively. The contributions of individual subsystems to the overall production cost are different. Given the gas price of $4.2/GJ, SMR and associated fuel consumption collectively account for 65.9% of the plant LCOH for gas-based production. Please note that at the gas-based hydrogen plant with CCS, the fuel combustion unit generates thermal energy for not only SMR but also the carbon capture process's solvent regeneration. Given the coal price of $57.3/metric ton, gasification and associated fuel consumption collectively account for 52.1% of the coal-based production. Thus, the progress of individual subsystems in learning will have a different effect on the overall production cost in the future. In addition, the fuel costs account for 50.0% and 14.2% of the plant LCOH for the gas- and coal-based production cases, respectively. Natural gas price is a key factor influencing gas-based blue hydrogen production.

Currently, hydrogen is mainly produced by SMR without CCS in the U.S., which is often called gray hydrogen. Compared to it, the blue hydrogen production by SMR with CCS can decrease the stack $CO_2$ emission intensity by 96% but increase the LCOH by 55%[7]. The resulting $CO_2$ avoidance cost by blue hydrogen is $65 per metric ton of $CO_2$. In contrast, the green hydrogen production by polymer electrolyte membrane electrolyzers almost has no stack $CO_2$ emissions but a high LCOH value ranging from $3.0 – 7.5/kg $H_2$[27]. The resulting $CO_2$ avoidance cost by green hydrogen relative to gray hydrogen varies from $212 – 689 per metric ton of $CO_2$, which is much higher than that by blue hydrogen. Obviously, there are tradeoffs in $CO_2$ avoidance cost and emission savings between the blue and green hydrogen production pathways. The details of emission and cost data and $CO_2$ avoidance cost estimation are available in Supplementary Note 2. Please note that the choice of a reference case affects the $CO_2$ avoidance cost.

### Future costs of blue hydrogen production without and with tax credits

A blue hydrogen production plant consists of numerous subsystems. However, the maturity status of individual subsystems and their initial installed capacity are different. As a result, learning rates and initial installed capacity vary by subsystem. Thus, a component-based learning curve model is employed to construct a plant-level learning curve based on individual subsystems' learning rates and initial installed capacity. In addition, the technological learning is evaluated in terms of the cumulative installed capacity of blue hydrogen instead of the number of new hydrogen production plants. To characterize the evolving costs of blue hydrogen produced from natural gas and coal resources in the future, this study first constructs learning curves for the total as-spent capital (TASC) and total operating and maintenance (TOM) cost of individual subsystems at each plant and then establishes the learning curve of the plant LCOH as a function of cumulative production capacity. To construct a learning curve for either the TASC or the TOM, initial installed capacity, initial cost, and learning rate have to first be determined. As discussed above, the initial TASC and TOM of individual subsystems are derived from the NETL study[7] and summarized in Supplementary Tables 6–10 in Supplementary Note 3 and Supplementary Tables 11 and 12 in Supplementary Note 4. The initial installed capacity (Supplementary Table 13) and learning rates of individual subsystems are collected mainly from numerous well-established studies and summarized in Table 2, in which bracketed values indicate uncertain ranges related to the base values. Both the initial installed capacity and learning rates vary significantly by subsystem. There are also high uncertainties in learning rates for both capital and O&M costs.

Blue hydrogen production without tax credits. At a global scale, the initial installed capacity of hydrogen production in 2021 was

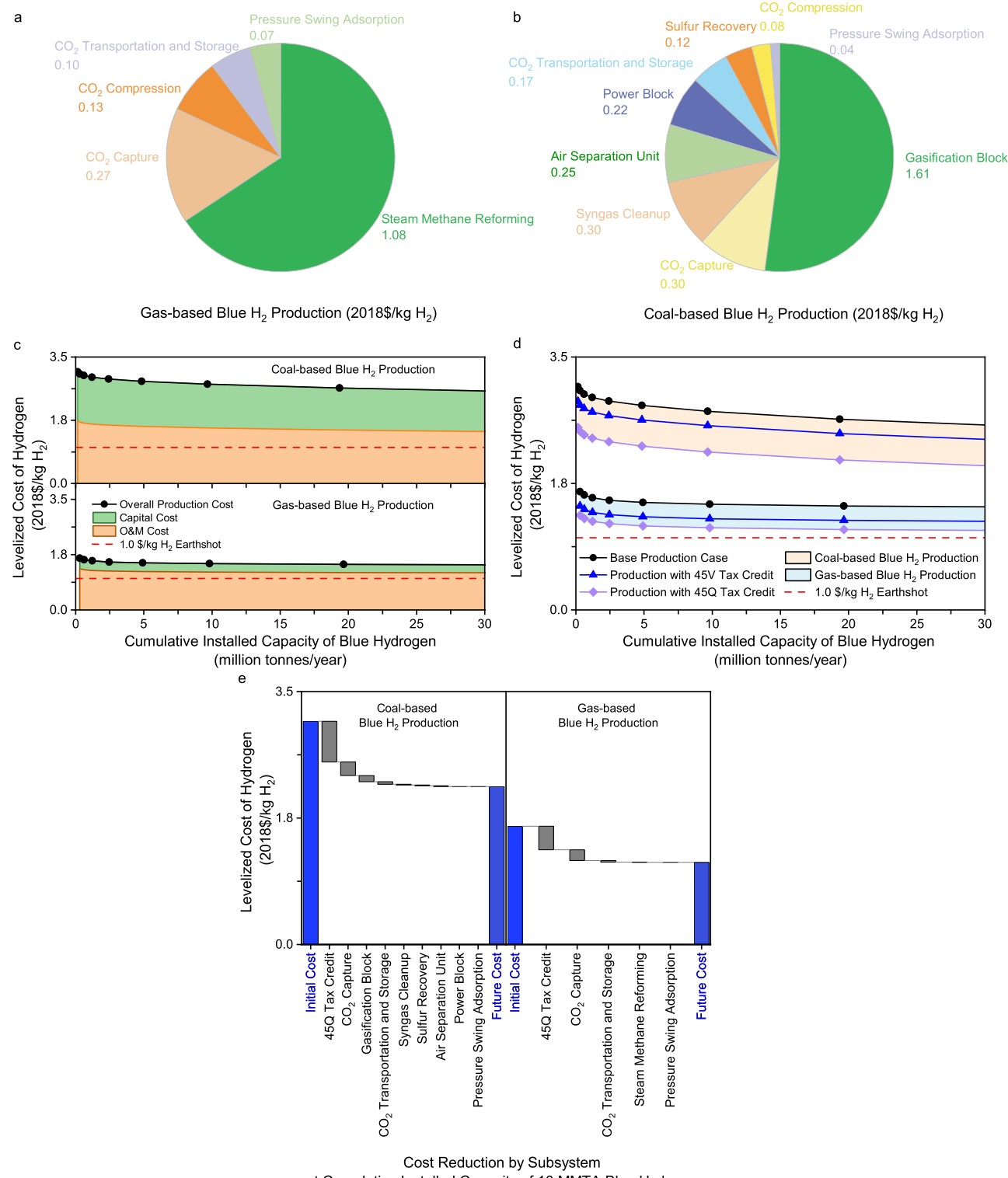

**Fig. 1 | Initial and future costs of blue hydrogen production without and with tax incentives. a** Distribution of initial levelized cost for gas-based H$_2$ production. **b** Distribution of initial levelized cost for coal-based H$_2$ production. **c** Learning curves for coal-based and gas-based H$_2$ production capital and operating and maintenance (O&M) costs without tax incentives. **d** Learning curves for overall levelized cost of coal-based and gas-based H$_2$ production without and with tax incentives. **e** Future cost reductions for coal-based and gas-based H$_2$ production with tax incentives.

estimated to be 0.31 MMTA for gas-based blue hydrogen and 0.15 MMTA for coal-based blue hydrogen[12,28]. Figure 1c shows the learning curves for levelized capital and O&M costs and plant LCOH for fossil-based hydrogen production. A comparison between different cost categories over a range of cumulative production capacity indicates

that the overall levelized cost of blue hydrogen will still be affected largely by TOM in the future, especially for gas-based production. In addition, a comparison between the two methods implies that SMR with CCS would continue to be more economically competitive for blue hydrogen production than gasification with CCS.

**Table 2 | Initial installed capacity and learning rates of individual subsystems for blue hydrogen production**

| Feedstock | Subsystems | Initial Installed Capacity (GW_th) | Learning Rates | | Sources of Data | |
|---|---|---|---|---|---|---|
| | | | Capital Cost | O&M Cost | Initial Installed Capacity | Learning Rates |
| Natural Gas | Steam Methane Reforming | 280[a] | 0.11 (0.05–0.17) | 0.00[f] | IEA[12,28] | Schoots et al.[13] |
| | Pressure Swing Adsorption | 366[a] | 0.11 (0.05–0.17) | 0.00[f] | IEA[12,28] | Schoots et al.[13] |
| | CO_2 Capture | 1.5[a] | 0.11 (0.06–0.17) | 0.22 (0.10–0.30) | IEA[12,28] | Rubin et al.[14]; IEAGHG[15] |
| | CO_2 Compression | 25[b] | 0.00 (0.00–0.10) | 0.00 (0.00–0.10) | Rubin et al.[14] | Rubin et al.[14]; IEAGHG[15] |
| | CO_2 Transportation and Storage | 2.4[c] | Not Applicable[d] | 0.05 | IEA[12,28] | NETL[51] |
| Coal | Air Separation Unit | 125[b] | 0.10 (0.05–0.15) | 0.05 (0.00–0.10) | Rubin et al.[14] | Rubin et al.[14]; IEAGHG[15] |
| | Gasification Block | 173 | 0.14 (0.07–0.21) | 0.12 (0.05–0.20) | Higman[50] | Rubin et al.[14]; IEAGHG[15] |
| | Syngas Cleanup | 173[e] | 0.10 (0.05–0.10) | 0.10 (0.05–0.10) | Higman[50] | Industrial Economics, Inc. and E.H. Pechan & Associates, Inc.[52] |
| | Sulfur Recovery | 125[b] | 0.11 (0.06–0.17) | 0.22 (0.10–0.30) | Rubin et al.[14] | Rubin et al.[14]; IEAGHG[15] |
| | Pressure Swing Adsorption | 411[a] | 0.11 (0.05–0.17) | 0.00[f] | IEA[12,28] | Schoots et al.[13] |
| | Power Block | 600[b] | 0.05 (0.03–0.09) | 0.00 | Rubin et al.[14] | Rubin et al.[14]; IEAGHG[15]; Zhai[16] |
| | CO_2 Capture | 0.9[a] | 0.11 (0.06–0.17) | 0.22 (0.10–0.30) | IEA[12,28] | Rubin et al.[14]; IEAGHG[15] |
| | CO_2 Compression | 25[b] | 0.00 (0.00–0.10) | 0.00 (0.00–0.10) | Rubin et al.[14] | Rubin et al.[14]; IEAGHG[15] |
| | CO_2 Transportation and Storage | 2.4[c] | Not Applicable[d] | 0.05 | IEA[12,28] | NETL[51] |

[a]The initial installed capacity of individual subsystems is estimated based on the global hydrogen production capacity. See Supplementary Equations (2) and (3) for the details.
[b]The capacity is converted from electric power (GW_e) to equivalent thermal power (GW_th), referring to a net thermal efficiency of 40%, which is a typical thermal efficiency of an integrated gasification combined cycle power plant or a supercritical coal power plant. See Supplementary Equation (4) for the details.
[c]It is assumed that CO_2 transportation and storage subsystem has the total initial installed capacity as the CO_2 capture subsystems at hydrogen production plants.
[d]The CO_2 transportation & storage subsystem is treated as the O&M cost component.
[e]It is assumed that the initial installed capacity of the syngas cleanup subsystem is the same as the gasification block subsystem.
[f]The O&M learning rate is assumed to be zero as no learning-by-doing behavior was found for the overall hydrogen production activity based on historical data (Schoots et al.[13]).

The annual demand for clean hydrogen produced from renewable and decarbonized fossil resources in the U.S. may reach 10 million metric tons of hydrogen per year by 2030[1]. As shown in Fig. 1c, the costs decline via incremental improvements to current technologies when the cumulative production capacity increases. When it reaches 10 MMTA, the capital and O&M costs decrease by 20.0% and 8.3% from the current levels for gas-based production, respectively. There are similar cost reductions for coal-based production. As a result, the plant LCOH decreases to $1.46/kg H_2 by 10.7% for the gas-based production and $2.75/kg H_2 by 10.9% for the coal-based production after 10 MMTA of hydrogen production capacity. Although experience learned from large-scale deployed projects will help to lower the future costs of blue hydrogen production, it is hard for both reforming and gasification technologies with CCS to reach the cost target of $1/kg H_2.

The plant LCOH trends are affected largely by key subsystems' capital and O&M learning rates, along with feedstock prices. For gas-based production, SMR and CCS are the key subsystems that dominate the plant LCOH, as discussed above. However, as shown in Table 2, there are no reductions gained in O&M costs from deploying SMR (including associated components), pressure swing adsorption (PSA) for hydrogen purification, and CO_2 compression. For the given natural gas price of $4.2/GJ, therefore, it is difficult to reach the cost target of $1/kg H_2, even when the cumulative production capacity goes beyond 10 MMTA.

Blue hydrogen production with tax credits. Both the 45Q and 45 V tax credits are to promote investment in clean hydrogen technologies and then lower the cost of hydrogen production. For blue hydrogen projects with CCS, however, a taxpayer cannot simultaneously claim both 45 V and 45Q tax credits during a given period. To claim either the 45Q tax credit or the 45 V tax credit, facilities must be placed in service before January 1st, 2033[29]. The credit is available for such qualified facilities for a period. The period of credit availability is common to eligible facilities, regardless of their start-of-service time.

In this study, it is assumed that the captured CO_2 is stored in saline reservoirs, which earns a carbon-sequestration credit of $85 per metric ton of CO_2 for 12 years. As mentioned earlier, the 45 V tax credit depends on the life cycle emissions of hydrogen production, which include greenhouse gas emissions from plant stacks, fuel supply, electric power supply, and CO_2 sequestration or management. The life cycle emissions were estimated by the NETL to range from 3.1 to 8.9 kg CO_2-eq/kg H_2 for the gas-based blue hydrogen in the 90% confidence interval between the 5th and 95th percentile values and from 3.4 to 8.9 kg CO_2-eq/kg H_2 for the coal-based blue hydrogen, which is driven mainly by the uncertainty in fuel supply[7]. The largest contributor among the multiple stages to the life cycle emissions is the fuel supply[7]. The median estimate of life cycle emissions is 4.6 kg CO_2-eq/kg H_2 for the gas-based blue hydrogen and 4.1 kg CO_2-eq/kg H_2 for the coal-based blue hydrogen[7], which is close to the threshold value of 4.0 kg CO_2-eq/kg H_2 required to

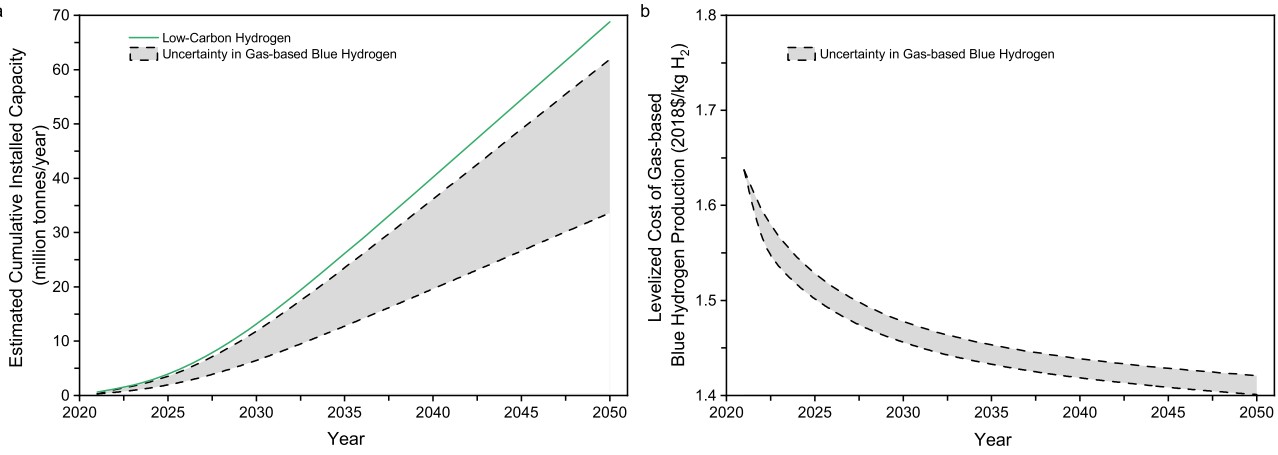

**Fig. 2 | Diffusion of cumulative installed capacity of low-carbon hydrogen and time-based learning curves of gas-based blue hydrogen. a** Diffusion of cumulative installed capacity. **b** Time-based learning curves of blue hydrogen production cost.

claim the minimum tax credit for clean hydrogen. Blue hydrogen projects have a fair possibility of earning a 45 V tax credit. Thus, the production tax credit for hydrogen projects is assumed to be $0.6 per kilogram of $H_2$ for 10 years. This assumption is optimistic for blue hydrogen in this study. However, there is no 45 V tax credit if the life cycle emissions of specific blue hydrogen projects are more than 4.0 kg $CO_2$-eq/kg $H_2$. See Supplementary Note 5 for additional information about life cycle emissions and tax credits. Figure 1d shows the learning curves for the plant LCOH for the gas- and coal-based hydrogen production with tax incentives.

Tax incentives lower the plant LCOH of hydrogen production. When the cumulative production capacity reaches 10 MMTA, the overall cost of hydrogen produced by SMR with CCS declines to $1.14 and $1.26 per kilogram of hydrogen produced with the 45Q and 45 V tax credits, respectively. They are 22.2% and 13.7% less than the LCOH without tax incentives, respectively. There are similar cost reductions with gasification-based production. These results indicate that the 45Q tax credit provides more economic incentives for blue hydrogen projects than the 45 V tax credit.

Tax incentives decrease the time-related learning experience necessary to reach a cost target. However, Fig. 1d shows that with either a hydrogen production tax credit or a carbon-sequestration tax credit, it is still hard for coal gasification with CCS to produce blue hydrogen at a cost of $1/kg $H_2$. In contrast, with the carbon-sequestration tax credit claimed for hydrogen projects, the cost of blue hydrogen produced by SMR with CCS approximates the Hydrogen Energy Earthshot, as shown in Fig. 1d.

Learning-by-doing will reduce the cost of hydrogen production for coal- and gas-based blue hydrogen. Figure 1e shows the cost reduction by subsystem and by the 45Q tax credit when the cumulative installed capacity of blue hydrogen reaches 10 MMTA. For blue hydrogen produced from both coal and gas resources, the overall cost reduction will be driven largely by the carbon-sequestration tax credit and the improvement in carbon capture. In contrast, other subsystems, such as SMR and PSA, will make limited contributions because they are mature technologies and have no or limited reductions from an additional 10 MMTA deployment in their future costs. These results indicate the importance of continued support from both public and private sectors for CCS-related research, development, and demonstration programs at federal and state levels.

### Time-based diffusion of blue hydrogen production

It is helpful for hydrogen energy planning to explore if certain production capacity and cost targets can be achieved by 2030. A new study reports the cumulative installed capacity of low-carbon

hydrogen production over time-based on globally announced, planned, and committed projects through 2030[30]. A diffusion-of-innovation model was established based on the current and future low-carbon hydrogen capacities through 2030 to explore the time-based diffusion of gas-based blue hydrogen over a long-term planning horizon through 2050.

Figure 2a shows the cumulative installed capacity estimates for global low-carbon hydrogen production over time. The gas-based blue hydrogen capacity accounts for 49% of the total low-carbon hydrogen capacity given in Table 1 and is estimated to be 90% in 2030 in terms of the International Energy Agency's hydrogen project databases[28,31]. Given the changing shares over time, Fig. 2a also shows a range of cumulative installed capacity for gas-based blue hydrogen in a particular year. The cumulative installed capacity of the global gas-based blue hydrogen may range from 6 to 12 MMTA in 2030, which implies that it would be hard for the blue hydrogen production by SMR with CCS alone in the U.S. to reach 10 MMTA in 2030.

Figure 1c shows the overall plant LCOH as a function of cumulative installed capacity for gas-based blue hydrogen, whereas Fig. 2a shows the cumulative installed capacity over time. Combining them together, Fig. 2b shows the overall plant LCOH of gas-based blue hydrogen production without tax credit over time. The result shown in Fig. 2b implies that for the fuel price and learning rates given in the base case, it would also be difficult for gas-based blue hydrogen to reach the ambitious cost target of $1/kg $H_2$ by 2030 in normal scenarios without aggressive incentives and game-changing technologies.

### Sensitivity of blue hydrogen production cost to key factors

Massive deployment of hydrogen projects will lower future costs for clean hydrogen production. Tax incentives for clean hydrogen will further decrease production costs and accelerate the technological evolution toward the Hydrogen Energy Earthshot. However, it is hard for coal gasification with CCS to reach the cost target of $1/kg $H_2$ for clean hydrogen. In contrast, the tax-incentivized production for blue hydrogen by SMR with CCS has the potential to reach the Hydrogen Energy Earthshot. Blue hydrogen projects announced in the U.S. will mainly employ gas-based reformation technologies with CCS[31]. The future production costs and their evolutionary trends are affected by natural gas price, carbon removal system cost, and learning rates in capital and O&M costs, as well as inflation when the cost is estimated in nominal dollars. In the U.S., natural gas prices are highly volatile. There are also high uncertainties in learning rates for many subsystems, which are shown in Table 2. The sensitivity analysis, therefore, is performed for the gas-based blue hydrogen with a focus on natural gas price,

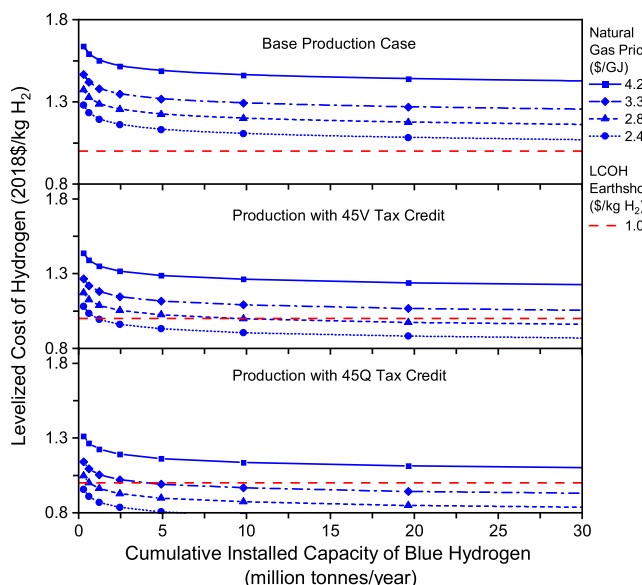

**Fig. 3 | Effect of natural gas price on future levelized cost of gas-based blue hydrogen production in three cases** including base production case, production with a 45V tax credit, and production with a 45Q tax credit.

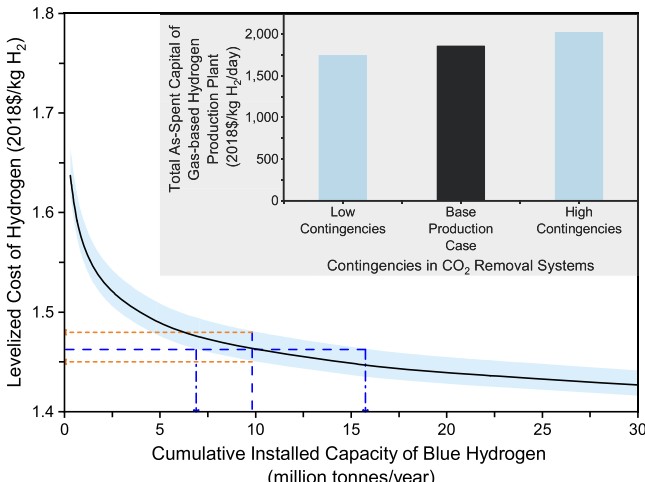

**Fig. 4 | Effects of process and project contingencies of $CO_2$ removal systems on initial capital cost and future levelized cost of gas-based blue hydrogen production.** The light blue shading area represents the uncertainties in the levelized cost of hydrogen production, which are driven by the uncertain cost estimates of carbon removal systems. The orange dash lines represent that the levelized cost of hydrogen varies from \$1.45 to 1.48/kg $H_2$ at the cumulative installed capacity of 10 MMTA, while the blue dash lines represent that to reach the cost of \$1.46/kg $H_2$, the cumulative installed capacity requirements vary from 7 to 16 MMTA.

carbon removal system cost uncertainties, learning rates, and inflation rate. In each parametric analysis, other parameters were kept at the base case values given in Table 2 and Supplementary Tables 1, 15, and 16 unless otherwise noted.

Effect of natural gas price. For blue hydrogen produced by SMR with CCS, the feedstock cost accounts for 50.0% of the plant LCOH, with an assumed natural gas price of \$4.2/GJ. In the past years from 2017 to 2022, the annual average Henry Hub gas prices ranged from \$1.9/GJ to \$6.1/GJ[32]. Thus, it is necessary to examine the economic benefit of low gas prices for blue hydrogen production. Figure 3 shows the effect of natural gas prices on the plant LCOH for hydrogen production without and with tax incentives. Obviously, the plant LCOH is highly sensitive to the natural gas price. Cheap natural gas resources help SMR-CCS decrease the cumulative production capacity necessary to reach the Hydrogen Energy Earthshot.

When the 45Q tax credit is claimed for hydrogen projects, the cumulative production capacity necessary to reach the cost target of \$1/kg $H_2$ is 4.9 and 0.6 MMTA if the gas price declines to \$3.3/GJ and \$2.8/GJ, respectively. As shown in Fig. 3, the initial plant LCOH is already less than \$1/kg $H_2$ if the natural gas price is \$2.4/GJ. When the 45 V tax credit is claimed, the cumulative production capacity should reach 9.8 and 1.2 MMTA to achieve the cost target when the gas price declines to \$2.8/GJ and \$2.4/GJ, respectively. Without tax incentives like 45Q and 45 V and increased learning rates, however, it is still difficult for this production method, even with cheap natural gas resources to reach the Hydrogen Energy Earthshot.

It is also worth noting that the gas-based hydrogen industry may have a sizable effect on the natural gas markets in the U.S., depending on the scale of blue hydrogen production in the future. For example, the production of 10 MMTA hydrogen by SMR with CCS would consume 1.9 billion GJ of natural gas per year, which is equivalent to about 17% of the national industrial natural gas consumption in 2022[33].

Effect of carbon removal system cost uncertainties. There are uncertainties in the process and project contingencies of two $CO_2$ removal systems employed for producing low-carbon hydrogen from natural gas resources. Such uncertainties affect the TASC and LCOH of a hydrogen production plant. The process contingency depends on the maturity level of a technology, whereas the project contingency depends on the availability of site-specific project details. In the base

case, the process contingency is 18% of the bare erected cost (BEC) for the Cansolv system and 0% for the MDEA system, while the project contingency is 25% of the sum of BEC, engineering, construction management, home office and fees, and process contingency and 25% for the Cansolv unit and 20% for the MDEA unit[7]. A parametric analysis is then conducted to reveal the collective impacts of uncertain processes and project contingencies, which take into account low and high contingencies. In the low contingencies scenario, the process contingency is 10% for the Cansolv system and 0% for the MDEA system, while the project contingency is 10% for both the $CO_2$ removal systems. In the high contingencies scenario, the process contingency is 40% for both the $CO_2$ removal systems, while the project contingency is 30% for both the $CO_2$ removal systems[34].

As shown in Fig. 4, the uncertainties in carbon removal system cost estimates have a sizable effect on the hydrogen production plant's TASC. As a result, the plant LCOH varies from \$1.45 to 1.48/kg $H_2$ at the cumulative installed capacity of 10 MMTA. To reach the cost of \$1.46/kg $H_2$, the cumulative installed capacity requirements vary from 7 to 16 MMTA. These results imply that cost uncertainties in carbon removal systems may result in pronounced variations in the estimation of the cumulative installed capacity necessary to reach a cost target.

Effect of learning rates. Learning rates directly drive future cost trends. In particular, the O&M learning rates of SMR, PSA, and $CO_2$ compression largely influence the pace of cost reductions toward the Hydrogen Energy Earthshot as their costs and associated fuel or electricity consumption collectively dominate the plant LCOH. In the base case, the O&M learning rates are zero for the three subsystems. However, Table 2 shows that there are uncertainties in learning rates, which can vary by 50% or more relative to the base values for some subsystems. Given such high uncertainties, it is important to examine the sensitivity of future cost trends to learning rates.

Additional scenarios are explored to examine the effect on the overall LCOH of increases in both the capital and O&M learning rates of individual subsystems with an emphasis on the increased O&M learning rates for SMR, PSA, and $CO_2$ compression. In these scenarios, the capital and O&M learning rates are elevated for individual subsystems to be 25% and 50% higher than the base values, except for SMR, PSA, and $CO_2$ compression. The O&M learning rates are increased for the

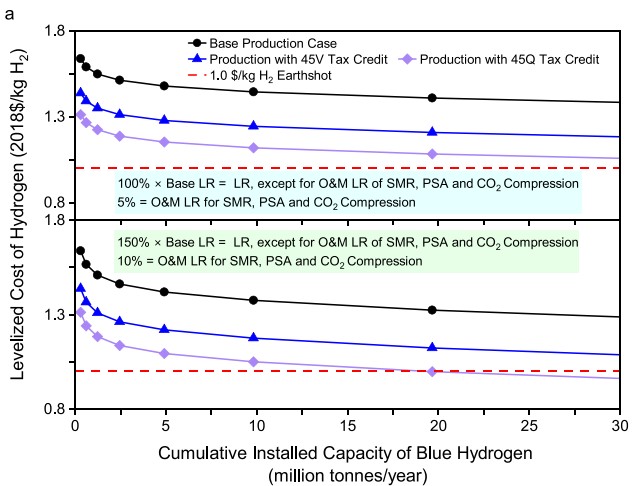

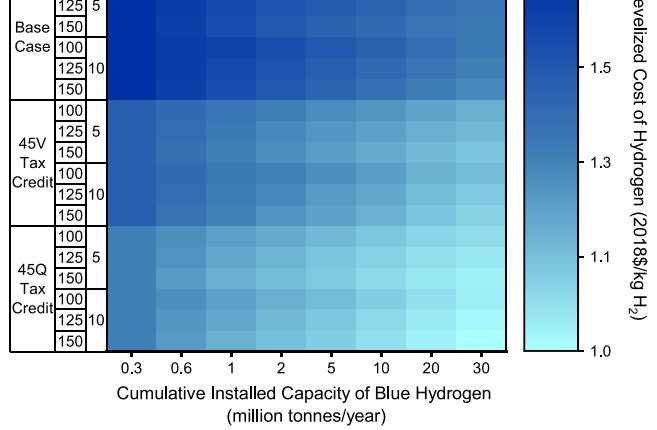

**Fig. 5 | Sensitivity of LCOH by SMR with CCS to learning rates. a** LCOH under two boundary scenarios of learning rates. **b** LCOH under the range of 100%–150% time-based learning rates, except for O&M cost learning rates, which are equal to 5%–10% for SMR, PSA, and CO$_2$ compression. Note to Fig. 5: P$_1$ means a percentage relative to the base learning rate, whereas P$_2$ means the learning rate on an absolute basis.

Note to abbreviations: CCS means carbon capture and storage; LCOH means levelized cost of hydrogen; LR means learning rate; O&M means operating and maintenance; PSA means pressure swing adsorption; and SMR means steam methane reforming.

three subsystems to 5% and 10% on an absolute basis. Figure 5 shows the sensitivity of the plant LCOH to the increased learning rates.

The increases in learning rates, especially the O&M learning rates of SMR, PSA, and CO$_2$ compression, obviously lower the cumulative production capacity necessary to reach a cost target. The results shown in Fig. 5, however, also imply that without tax incentives for clean hydrogen, it would still be challenging for blue hydrogen produced from expensive natural gas resources to reach the Hydrogen Energy Earthshot by 2030, even if the progress in learning is to accelerate substantially. If the O&M learning rates of SMR, PSA, and CO$_2$ compression reach more than 5%, massive deployment of blue hydrogen projects claimed with 45Q tax incentives can decrease the plant LCOH to $1/kg H$_2$. Figure 5a, b show that with an O&M learning rate of 10% for the three subsystems, the breakeven cumulative production capacity is 20 MMTA or more, which is also affected by other subsystems' learning rates.

**Effect of inflation rate.** In general, this study estimates the cost of hydrogen production in real dollars. When the cost is estimated in nominal dollars, however, both the initial and future LCOH estimates vary with the inflation rate as it affects the discount rate, fixed charge rate, and levelization factor. A parametric analysis was further performed for the inflation rate to quantify its effect on the evolving cost of gas-based blue hydrogen production toward the Hydrogen Energy Earthshot. Figure 6 shows the learning curves of blue hydrogen production with inflation. Figure 6a, b show that at a given level of cumulative installed capacity, the LCOH in nominal dollars increases when the inflation rate increases from 1% to 3%. As a result, blue hydrogen production may not reach the cost target of $1/kg H$_2$ for both scenarios without and with a 45Q tax credit even when the cumulative installed capacity reaches 30 MMTA. Figure 6c further shows that with an inflation rate of 3%, the future LCOH may get close to the cost target when cheap natural gas resources are used as the feedstock to produce blue hydrogen with a cumulative installed capacity of up to 30 MMTA.

Figure 6a, b also compare the learning curves of blue hydrogen production between the two scenarios without and with inflation. As shown in Fig. 6a for the scenario without a 45Q tax credit, the reduction in hydrogen production cost from deploying the cumulative installed capacity of 10 MMTA can be offset by an inflation rate of 1%. There is a similar result at the cumulative installed capacity of 5 MMTA

for the scenario with a 45Q tax credit, as shown in Fig. 6b. All these results imply that inflation would remarkably raise challenges for blue hydrogen production to reach the Hydrogen Energy Earthshot in the near future.

## Discussion

This study reveals opportunities and challenges while creating the new hydrogen economy. The capex learning rates of green hydrogen production are 9% and 13% for alkaline electrolysis and polymer electrolyte membrane electrolysis, respectively[35,36], which are similar to those for SMR and PSA. However, the cost competitiveness of green hydrogen relative to blue hydrogen in the future remains an open question as the future production cost also depends on initial installed capacity, renewable electricity cost, tax credit, and other factors[27,36–38]. Experience learned from the deployment of blue hydrogen projects will be helpful in lowering future costs of hydrogen production by both SMR and gasification with CCS. In comparison between the two hydrogen production methods, SMR with CCS will continue to be more economically competitive. For the given estimates of learning rates for SMR and CCS, the overall cost of blue hydrogen production without tax incentives can be decreased by 10.7% from the current level to $1.46/kg H$_2$ when the cumulative production capacity reaches 10 MMTA. In addition to the learning rates, the overall LCOH is also affected significantly by natural gas price. Without tax incentives, however, it is still hard for SMR with CCS to reach the Hydrogen Energy Earthshot even when natural gas resources are cheap, which implies an urgent need for radical innovation in technology. With tax incentives, the breakeven cumulative installed production capacity, which is required for SMR with CCS to reach the cost target of $1/kg H$_2$, highly depends on the gas price and can be much less than 10 MMTA when gas prices are not more than $3.3/GJ. In contrast, when natural gas resources are not cheap, the breakeven cumulative production capacity required for tax-incentivized SMR with CCS is at a level of 20 MMTA or more even if the learning rates for capital and O&M costs are increased remarkably. In a short summary of these findings, tax credit, natural gas price, and learning rates are the most significant factors that collectively determine the breakeven cumulative production capacity required for SMR with CCS to reach the Hydrogen Energy Earthshot. However, inflation can remarkably elevate challenges for blue hydrogen to reach the cost target.

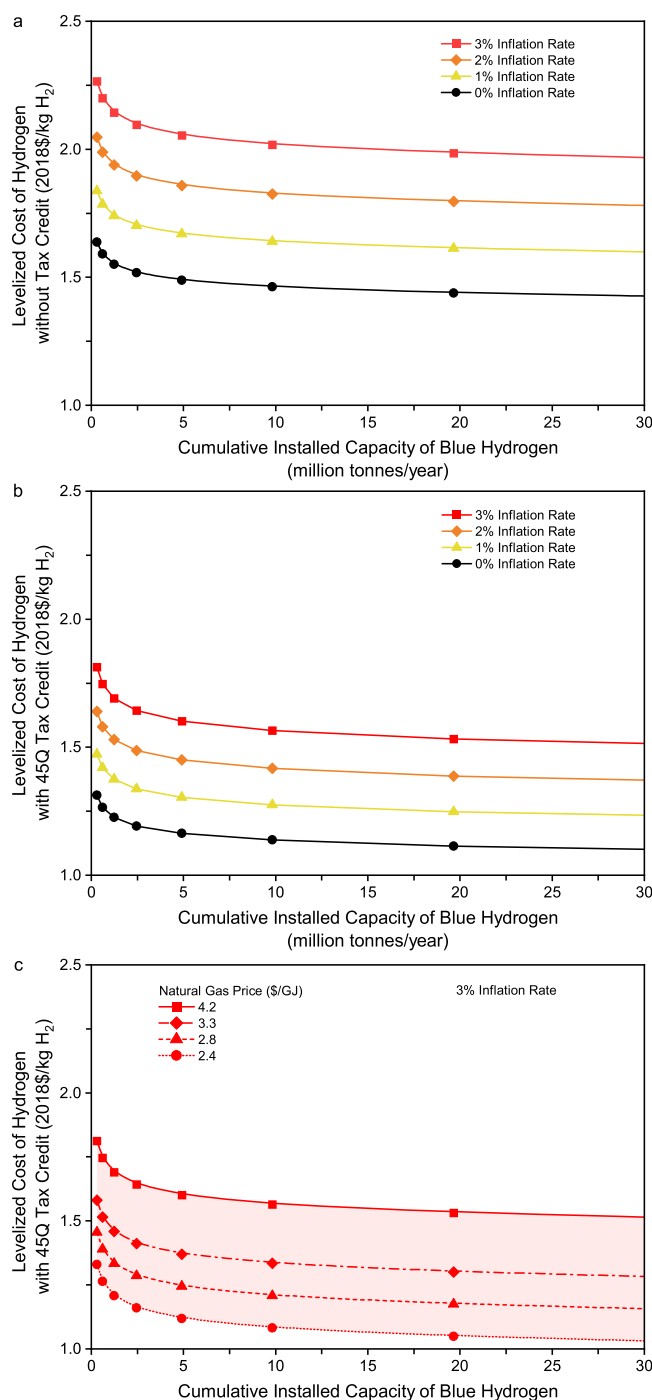

**Fig. 6 | Effect of inflation rate on future cost of gas-based blue hydrogen production without and with 45Q tax credit. a** Levelized cost of hydrogen production with a gas price of $4.2/GJ and without a 45Q tax credit. **b** Levelized cost of hydrogen production with a gas price of $4.2/GJ and 45Q tax credit. **c** Levelized cost of hydrogen production with a 3% inflation rate and 45Q tax credit.

offtake[1,2]. The hesitancy to long-term, scaled contracts is influenced by numerous factors, such as lack of price certainty, unavailability and reliability of large-scale hydrogen supply, near-term policy implementation uncertainty, and long-term political uncertainty[1,2]. For blue hydrogen projects, enhancements in tax credits for carbon sequestration can improve the economics of hydrogen production. For example, an extension of the 45Q tax credit period from the current 12 years to 18 years would significantly reduce the cumulative installed capacity required for gas-based blue hydrogen projects to reach the Hydrogen Energy Earthshot, as demonstrated in Supplementary Fig. 3. Extending the period of the 45Q tax credit for blue hydrogen projects can be considered an option to secure financing and promote long-term offtake.

Production of blue hydrogen requires fossil fuel resources, water, and land. For illustrative purposes, Supplementary Fig. 4 shows the annual natural resource requirements for SMR with CCS as a function of cumulative hydrogen production capacity based on the plant design and performance given in Supplementary Table 1. The hydrogen production of 10 MMTA by SMR with CCS would annually require 1.9 billion GJ of natural gas resources, withdraw 0.30 km$^3$ of water resources, and occupy 25.1 km$^2$ of land resources. Deployment of large-scale hydrogen projects toward the Hydrogen Energy Earthshot will pronouncedly affect multiple natural resources and local resource planning. In addition to the production cost, massive blue hydrogen production should also be planned in the context of resource sustainability.

Tax incentives for either clean hydrogen production or carbon sequestration accelerate technological learning to reduce the time-related cumulative production capacity necessary to reach a cost target for hydrogen production and in turn, lower natural resource consumption and environmental impacts. In comparison between the two types of tax credits, blue hydrogen projects gain more economic value from the 45Q tax credit than the 45 V tax credit. For post-combustion CCS, enhancing the $CO_2$ removal efficiency from 90% to 99% or more can achieve net-zero emissions while increasing the $CO_2$ avoidance cost by only 3–13% or less[39]. Thus, deep CCS for 99% $CO_2$ capture can be considered for blue hydrogen production to nearly fully remove site emissions and in the meantime, receive increased economic benefits from the 45Q tax credit.

Radical innovation in technology and systems integration is urgently needed to reach the Hydrogen Energy Earthshot by 2030. In addition to deep CCS, advanced hydrogen technologies, such as thermal and catalytic pyrolysis of natural gas, dry reforming of methane, and chemical looping, should be researched, developed, and demonstrated (RD&D) for buying down learning curves[40]. Autothermal reforming and partial oxidation can be considered an option for greenfield investment in blue hydrogen[41]. There is a strong need for collaborative support and joint efforts on RD&D from both the public and private sectors. However, it may be still challenging for advanced hydrogen technologies to reach the cost target by 2030. Improvements in hydrogen technology alone may not be enough to reach the cost target of $1/kg H$_2$[42]. Cost reduction pathways beyond technology advancement for clean hydrogen production should also be explored extensively. As SMR without carbon abatement is the most widely-used process for hydrogen production now, retrofitting CCS to existing SMR facilities can substantially decrease the investment and then the overall production cost for blue hydrogen. While building new production plants to meet the significantly increasing demand for clean hydrogen, retrofit of CCS or deep CCS is a cost reduction option for blue hydrogen in the near term. In addition, advances in solvent regeneration for CCS or deep CCS can lower the energy penalty to improve economic viability[43]. By-product sales, $CO_2$ valuation, hydrogen production integration with other energy systems, and optimal siting of production plants are the additional options for cost reductions toward the Hydrogen Energy Earthshot[42].

The global production capacity of low-carbon hydrogen will reach 12.3 MMTA by 2030 based on the announced, planned, and committed projects[30]. The low-carbon hydrogen capacity in North America will reach 6.8 MMTA by 2030[30]. However, only 1.8 MMTA[30] and 1.5 MMTA[2] of the announced projects in North America and the U.S. have reached the final investment decision (FID), mainly because many announced projects have not yet secured financing and nailed down contracted

Competing strategies and supportive policy and regulatory actions should be made rapidly on both the hydrogen demand and supply sides at both federal and state levels in alignment with the innovation expansion. A variety of high-level strategies are needed on the demand side to promote the widespread use of low-carbon hydrogen in industrial, transportation, and power sectors and then establish large-sale markets for low-carbon hydrogen[1,2]. To jumpstart a clean hydrogen economy, a cluster approach can be employed on the supply side to establish regional production-transportation-demand networks by co-locating feedstock supply, hydrogen production, and carbon sequestration with multiple end-users and by utilizing existing infrastructure, such as pipeline infrastructure for natural gas, $CO_2$, and $H_2$ transportation and geological reservoirs for $CO_2$ storage. To scale the regional hydrogen economy, secured investments in hydrogen production and supporting infrastructure are required with funding from both public and private sectors, plus subsidies and tax incentives. In addition, deploying large hydrogen production plants instead of small ones can improve engineering economics at a plant level. Given the important role of CCS in producing competitive blue hydrogen, continued support for large-scale demonstration projects should be boosted in the near term to reduce the CCS cost and its uncertainty. Investments in blue hydrogen should be prioritized to lock down sufficient financial resources for the most competitive technologies in the near term. Economic and policy incentives can be tailored with emphasis on gas-based blue hydrogen to catalyze its widespread deployment and technological evolution because of the pronounced cost advantage relative to coal-based blue hydrogen. Extending the 45Q tax credit from the current 12-year period to a longer period for gas-based blue hydrogen projects would remarkably lower the time-related cumulative installed capacity necessary to reach the Hydrogen Energy Earthshot. In addition to the enhanced tax incentives, emission trading can be another driver to facilitate low-carbon technology deployment and then accelerate technological evolution[44].

Although the future cost of blue hydrogen is expected to decrease from cumulative experience in conjunction with tax incentives, blue hydrogen may not be economically competitive with gray hydrogen in the near term. To jumpstart a clean hydrogen economy, therefore, a carrot-and-stick policy approach can be employed to stimulate low-carbon hydrogen deployment and constrain gray hydrogen production from unabated fossil resources.

Numerous tax-incentivized blue hydrogen projects with cheap feedstocks will be needed at a total capacity of roughly several million metric tons of hydrogen per year to reach the Hydrogen Energy Earthshot, which highly depends on the several key factors discussed above. However, the current deployment of SMR with CCS for blue hydrogen production is lagging far behind estimates of what is required. Regional hydrogen hubs will be needed across the country to produce clean hydrogen at the targeted cost level. Large-scale blue hydrogen production will consume multiple types of natural resources, such as fossil, water, and land resources, and geological reservoirs and transportation infrastructure for $CO_2$ storage, and affect local management and planning of these natural resources. The availability and price of these natural resources vary by region or location in the country. Thus, siting blue hydrogen production plants should take into account the co-location and co-availability of these natural resources. In contrast, green hydrogen would ideally be sited in regions with high penetration of solar and wind energy.

Several caveats accompany this study. A single-factor top-down model using constant learning rates is applied to characterize technological evolving trends in the future. However, it is a simplified representation of future cost trends for large-scale new technologies or process designs[38]. Technological evolution is projected for blue hydrogen based on the empirical experience of continuing improvements to current technologies, which is similar to other learning curve studies on CCS[14,16,38]. However, technology innovation may progress in a complex pattern instead. Sensitivity analysis is performed to explore the potential consequences of uncertain learning rates but without the quantification of the likelihood of the outcomes due to a lack of sufficient empirical data. If new game-changing technologies were to be created and deployed widely, however, the resulting learning could be accelerated substantially to result in greater cost reductions or less cumulative production capacity requirements to reach the Hydrogen Energy Earthshot than those estimated in this study.

Last but not least, it is very important to prevent upstream methane emissions while reforming natural gas with CCS to produce blue hydrogen. Methane leakage along the natural gas supply chain can jeopardize the role of natural gas in the energy transition to a low-carbon or net-zero future, even when CCS is deployed[45,46]. A high methane leakage rate of 3.5% or more can elevate the blue hydrogen's carbon footprint and make it uncompetitive or even unviable in a hydrogen economy[4]. To reduce or avoid the risks of committing to high-emitting blue hydrogen, stringent standards and regulations should be imposed to limit methane leakage and promote the deployment of the best available technologies for methane abatement[46].

## Methods

This study applies empirical learning curves to characterize the evolving costs of blue hydrogen production without and with future tax incentives. The section presents a diffusion-of-innovation model, a component-based learning curve model, and an economic metric for technology evaluation and then discusses the sources of data used for the model formulation and cost estimation.

### Diffusion-of-innovation model

The diffusion of innovation describes how a new technology would spread over time. An S-shaped curve is often used to measure the diffusion over time, in which the adoption rate increases during the early stage, reaches a maximum level at the point of inflection, and decreases until the diffusion curve saturates[47]. To estimate the annual installed capacity of low-carbon hydrogen over time, the S-shaped diffusion function is employed[48,49]:

$$cc_t = \frac{cc_{sat}}{1 + \frac{cc_{sat} - cc_0}{cc_0} \cdot e^{-r \cdot t}} \quad (1)$$

Where $cc_t$ is the annual installed capacity of low-carbon hydrogen in a particular year "$t$" (million metric tons per annum, MMTA); $cc_{sat}$ is the saturation level of annual installed capacity (MMTA); $cc_0$ is the initial annual installed capacity in the start year (MMTA); r is the growth rate (fraction); and t is a particular year after the start period. The function coefficients are estimated by regression based on current and future low-carbon hydrogen production capacities through 2030[12,28,30]. Additional details about the regression and diffusion function are available in Supplementary Note 7. Once the annual installed capacity in future years is determined, the cumulative annual installed capacity can be estimated as a function of time.

### Component-based learning curve model

A learning curve represents an empirical relationship between production unit cost and cumulative production capacity over time. The improved performance gained from time-related experience translates to lower the cost of hydrogen production. The one-factor learning curve model is used widely to characterize cost trends of new technologies or systems[14,16,38]. In this model, the future cost of a technology is estimated as a function of cumulative installed production

capacity[38]:

$$C = a \cdot x^b \tag{2}$$

Where $C$ is the unit cost at the cumulative installed capacity; $x$ is the ratio of cumulative installed capacity relative to initial capacity; $a$ is the constant unit cost at the initial installed capacity; and $b$ is the constant learning parameter. The fractional reduction in cost from each doubling of cumulative installed capacity is defined as the learning rate and is calculated as[38]:

$$LR = 1 - 2^b \tag{3}$$

Where LR is the learning rate, and factor $2^b$ is the progress ratio. The cost of interest can be either the total capital cost or the total operating and maintenance (O&M) cost. To construct a learning curve for a technology with respect to its either total capital cost or total O&M cost, three types of model parameters have to be specified, including the initial cost, initial installed capacity, and learning rate. Capital and O&M learning rates can be estimated using empirical data for mature technologies or an analogous approach for advanced technologies. For example, the learning rates for SMR were derived from its historical installed capacity and cost data[13], whereas the learning rates for post-combustion carbon capture were estimated by referring to those of post-combustion flue-gas desulfurization as they are technically analogous[14,38]. The data collected for these parameters are discussed later.

A hydrogen production plant involves multiple technologies or subsystems, which have different values regarding the three parameters defining a learning curve. At blue hydrogen production, individual subsystems lie at different levels of technological maturity. Learning rates and initial installed capacity count on maturity level and then vary by subsystem. For example, at a gas-based blue production plant, SMR and PSA are mature subsystems, whereas carbon capture has not been deployed widely, though it is commercially available. As a result, the O&M learning rates are zero for SMR and PSA but 22% for carbon capture. Thus, a component-based learning curve model is applied to estimate the total cost of hydrogen production at a certain level of cumulative installed capacity as the sum of individual subsystem costs[38]:

$$C = \sum_{i=1}^{Num} a_i \cdot x^{b_i} \tag{4}$$

$$b_i = \log(1 - LR_i) / \log(2) \tag{5}$$

Where $C$ is the total cost per plant at the cumulative installed capacity; $a_i$ is the initial cost of subsystem "i" to produce the first unit; $b_i$ is the learning parameter for subsystem "i"; $LR_i$ is the learning rate of subsystem "i"; and Num is the number of subsystems at a plant. The gas- and coal-based production plants are shown in Supplementary Note 1 and Supplementary Figs. 1 and 2 are decomposed into five subsystems and nine subsystems, respectively. The individual subsystems are reported in detail for the two plants in Supplementary Tables 3 and 4, respectively.

## Cost metric for technology evaluation
The component-based learning curve model is applied to project the future total capital cost and total O&M cost of individual subsystems and an overall plant as a function of cumulative installed capacity and then estimate the overall cost of hydrogen production for a given cumulative installed capacity. The cost metric considered for technology evaluation is the levelized cost of hydrogen and is estimated in

real dollars[7,34].

$$LCOH_R = LCC_R + LOM_R + LFP_R \tag{6}$$

$$LCC_R = \frac{TASC_R \cdot FCR_R}{(CF \cdot AH) \cdot KG_{H2}} \tag{7}$$

$$LOM_R = \frac{OM_R}{(CF \cdot AH) \cdot KG_{H2}} \tag{8}$$

$$LFP_R = \frac{FC_R \cdot FR}{KG_{H2}} \tag{9}$$

Where the subscript "R" means the real dollars; $LCOH_R$ is the levelized cost of hydrogen of a blue hydrogen production plant ($/kg $H_2$); $LCC_R$ is the levelized capital cost ($/kg $H_2$); $LOM_R$ is the non-fuel levelized operating and maintenance cost ($/kg $H_2$); $LFP_R$ is the levelized fuel price ($/kg $H_2$); $TASC_R$ is the total as-spent capital of a blue hydrogen production plant ($); $FCR_R$ is the fixed charge rate (fraction/year); CF is the plant capacity factor (%); AH is the total annual hours (8760 h); $KG_{H2}$ is the hourly hydrogen production rate (kg $H_2$/hour); $OM_R$ is the total non-fuel operating and maintenance (O&M) cost ($/year), including both the fixed and non-fuel variable O&M costs; $FC_R$ is the natural gas cost ($/GJ) or coal cost ($/metric ton); FR is the hourly natural gas flow rate (GJ/hour) or coal flow rate (metric ton/hour). When estimating the LCOH in real dollars, the FCR has to be determined using Eq. (10) to (13), which varies with the project book lifetime and a panel of financial variables[34]:

$$FCR_R = \frac{CRF_R^{nonfuel} - ETR \cdot PV_{plant}}{1 - ETR} \tag{10}$$

$$CRF_R^{nonfuel} = \frac{ATWACC_R \cdot (1 + ATWACC_R)^{BL}}{(1 + ATWACC_R)^{BL} - 1} \tag{11}$$

$$PV_{plant} = CRF_R^{nonfuel} \cdot \sum_{n=1}^{m} \frac{d_n}{(1 + ATWACC_R)^n} \tag{12}$$

$$ATWACC_R = PC_{equity} \cdot ROE_R + PC_{debt} \cdot kd_R \cdot (1 - ETR) \tag{13}$$

Where the superscript "nonfuel" represents the non-fuel component; $CRF_R$ is the capital recovery factor (fraction/year); ETR is the effective tax rate (%); $PV_{plant}$ is the present value of tax depreciation expense of a blue hydrogen project (fraction/year); $ATWACC_R$ is the after-tax weighted average cost of capital (%); BL is the project book lifetime (year); $d_n$ is the tax depreciation fraction in a year (n) (fraction); m is the number of years of depreciation (year); $PC_{equity}$ is the percent of equity (%); $PC_{debt}$ is the percent of debt (%); $ROE_R$ is the real rate of return on equity (%); $kd_R$ is the real rate of cost of debt (%). The financial parameters and their data sources are detailed in Supplementary Table 17 and the resulting FCR is available in Supplementary Table 18.

Inflation affects discount rate, ATWACC, FCR, and other factors. To evaluate the effect of inflation on the cost of blue hydrogen production, the LCOH is estimated in nominal dollars. The detailed estimation of the LCOH in nominal dollars is reported in Supplementary Note 8.

When either a clean-hydrogen production tax credit or a carbon-sequestration tax credit is considered for a blue hydrogen project, the levelized tax credit (LTC) and the levelized cost of hydrogen with a tax credit are estimated in real dollars using Equation (14a) or Equation

(14b) and Equation (15), respectively:

$$LTC_{R,45v} = \frac{TC_{45v} \cdot AHP_{H2} \cdot CP_{45v}}{BL \cdot AHP_{H2}} \tag{14a}$$

$$LTC_{R,45Q} = \frac{TC_{45Q} \cdot ACS_{CO2} \cdot CP_{45Q}}{BL \cdot AHP_{H2}} \tag{14b}$$

$$LCOH_{R,TC} = LCC_R + LOM_R + LFP_R - LTC_R \tag{15}$$

Where $LCOH_{R,TC}$ is the levelized cost of hydrogen with a tax credit ($/kg $H_2$); $LTC_{R,45V}$ and $LTC_{R,45Q}$ are the levelized tax credit of Sections 45 V and 45Q over the book lifetime of a blue hydrogen project, respectively ($/kg $H_2$); $AHP_{H2}$ is the annual hydrogen production (kg $H_2$/year); $ACS_{CO2}$ is the annual $CO_2$ sequestration amount (metric ton $CO_2$/year); BL is the book lifetime of a hydrogen project (years); $CP_{45V}$ is the 45 V credit period (years); $CP_{45Q}$ is the 45Q credit period (years); $TC_{45V}$ is the 45V bonus rate ($/kg $H_2$), which depends on the life cycle GHG emissions ("well-to-gate" emissions); and $TC_{45Q}$ is the 45Q bonus rate ($/metric ton $CO_2$).

## Data sources

To construct a learning curve, initial installed capacity, initial cost, and learning rate must be specified. As discussed below, the data for these parameters are collected from various sources for individual subsystems.

The initial TASC and total O&M cost of individual subsystems are derived from the NETL's recent study on state-of-the-art commercial technologies for blue hydrogen production[7]. The NETL study provides the estimates of total plant cost for individual processes or systems, as well as the owner's cost and the TASC and O&M costs for the overall plant. The cost breakdowns are applied to the NETL's plant-level cost estimates to come up with the initial TASC and total O&M cost for individual subsystems. The cost allocation methods and relevant cost estimates are detailed in Supplementary Notes 3 to 4 and Supplementary Tables 6 to 12.

The initial installed capacity of individual subsystems is estimated based on the current global installed capacity unless noted otherwise, which implies conservative projections of technological learning. The installed capacity is measured in equivalent gigawatts of thermal energy. When the installed capacity is measured in electric power capacity, it is converted to equivalent gigawatts of thermal energy with an assumed thermal efficiency of 40%. Thermal energy capacity is further converted to the mass-based production capacity of hydrogen in terms of its higher heating value. The data of initial installed capacity are mainly derived or collected from the International Energy Agency's Global Hydrogen Review[12,28], a highly-cited learning curve article[14], and the Global Syngas Technology Council syngas database[50]. Additional details of initial installed capacity are available in Supplementary Note 4 and Supplementary Table 13.

The data on capital and O&M cost learning rates are mainly collected from three learning curve studies on reforming hydrogen production and power generation with CCS, respectively[13–15]. Additional learning rate data come from other studies[16,51,52].

## Data availability

The source data generated in this study have been deposited in the Figshare database: [https://doi.org/10.6084/m9.figshare.23821926]. The data can also be obtained by contact with the corresponding author.

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

## Acknowledgements
This study is supported by the University of Wyoming's School of Energy Resources. The authors greatly acknowledge Holly Krutka, Scott Quillinan, Edward Rubin, Zitao Wu, and Landen Fuller for assistance with this study. The views and opinions of this article are those of the authors alone and do not necessarily state or reflect those of the United States Government or any agency thereof.

## Author contributions
H.Z. conceived and guided the research; W.W. performed the experiments; W.W. and H.Z. did the analysis; E.H. contributed to the discussion; H.Z. and W.W. wrote the original manuscript; and all the authors reviewed and edited the manuscript.

## Competing interests
The authors declare no competing interests.
