## [Peer Review File · Nature Communications]

Technological evolution of large-scale blue hydrogen production toward the U.S. Hydrogen Energy EarthshotReviewers' Comments:

Reviewer #1:

Remarks to the Author:

The authors present an analysis on future blue hydrogen costs in the US based on different technologies, learning rates and natural gas prices. They compare the effect of two different tax credits that are in place, in the perspective of reaching a target cost of 1 USD/kg for blue hydrogen. The results are of interest for the research community, given the importance of the topic. Unfortunately, I believe that the present version of the manuscript has some important issues, and it is not suitable for publication on an international journal such as Nature Communications. I report my comments below.

The economic analysis provided by the authors is detailed and include many different sub-systems and parameters. However, in a general perspective, I believe that other very important aspects are not discussed at all. The crucial point is the CO₂ emission savings that can be achieved by blue hydrogen, especially compared to green hydrogen, which appears to be the elephant in the room in this analysis. The authors briefly mention lifecycle emission ranges at page 11, but no additional information is given nor discussed. I think indeed this is a crucial point that should be addressed to support the main results. The authors could also estimate a cost per avoided tonne of CO₂ emission to be compared to other technologies and studies.

The literature review seems quite limited, especially given the large number of studies on the subject in the last years. The learning rates reported by the authors (page 5) rely on literature that is more than 15 years old, which seems quite outdated considering the recent technological evolution. I think that using more recent data would improve the quality and reliability of the results.

Furthermore, CCS in real applications has shown a lower economic performance than expected, and I think this potential issue should be addressed in the discussion and maybe also in the sensitivity analysis.

An additional issue about CCS is that the authors do not seem to consider any investment cost for storage (tables 10 and 11). I think this is a major issue, as these systems should be built, and are also expected to require an important part of the tax credit that the authors are completely allocating to blue hydrogen production. This aspect should be better discussed in the paper, especially considering the important debate about the suitability of CCS.

The authors consider a tax credit that can be claimed for lifecycle GHG emissions lower than 4 kgCO₂/kgH₂. However, the GHG emission ranges that they provide at page 11 show median values that are much higher than 4. How do they justify this assumption? Please clarify this issue.

The results of the authors are expressed with respect to the potential blue hydrogen deployment capacity. How these capacities compare to the expected demand of blue hydrogen by 2030? I think this information is fundamental to put the results in the right perspective.

The sensitivity analysis (or at least the discussion) may also address the potential effect of different levels of inflation (and possibly other economic parameters), given the significant variations over the last years. Would this affect the reliability of the results?

Additional aspects to be addressed:

Please include line numbers when submitting a paper for review.

What is the U.S. Hydrogen Earthshot reported in the title and the abstract? The authors never explicitly define it. Please clarify it for the readers.

In the introduction, a quick comparison of blue hydrogen with green hydrogen should be provided. Also, I think some information on the actual emissions of current blue hydrogen projects should be illustrated, to inform the readers about the current levels of greenhouse gas emissions associated with this technology. Although they are already reported by the authors in Table 2 in the supplementary material, I think this is a crucial performance indicator of the technology. Also, a description of the contribution of different stages on the lifecycle emissions could be provided.

Page 6: the authors compare the LCOH of SMR and gasification, but I think they should also mention the two different levels of GHG emissions. Gasification is not only more expensive than SMR, as

authors correctly highlight, but it also leads to (1) a lower capture rate, (2) a higher net GHG emission intensity compared to SMR.

Page 6: When discussing the cost distribution in subsystems I think that the reference fuel cost for gas and coal should be provided, as their variability significantly affects the shares that are discussed. This information is of course provided by the authors together with the other parameters of the study, and also addressed later in the sensitivity analysis (at least for natural gas), but I think that including it here in the discussion may help the readers in better understanding the results that are described.

The y scale of Figure 1 should report the 1 USD/kg that is the target aimed by the US policy.

Page 18, end of the page: it seems to me quite hard that advanced technologies can contribute to the cost target by 2030, given the short time horizon. Please clarify this aspect in the discussion.

Page 19: Location of blue hydrogen hubs should also take into account the availability of CO₂ storage sites and related infrastructure. This is an important aspect that is not mentioned by the authors.

There are some typos in the manuscript (e.g. page 13, "nature gas", page 17 "summery", etc.).

Reviewer #2:

Remarks to the Author:

The presented manuscript "Technological evolution of large-scale blue hydrogen production toward the U.S. Hydrogen Earthshot" by W. Wu, H. Zhai, and E. Holubnyak studies the role that blue hydrogen will play in the decarbonisation of US energy system and in reaching the US Hydrogen Earthshot. The authors take data from a recent NETL study on the production cost of blue hydrogen and analyse future cost reductions through "learning-by-doing", where they assume exponential learning curves associated with increased technology deployment in coming years.

In summary, the manuscript investigates a highly relevant and timely topic of research. Yet, I see the need for substantial improvement to meet the standards expected by research articles published in Nature Communications.

Here is a list of my main points of criticism:

- 1) The manuscript does not connect to the wider context of and societal/political/scientific debate about hydrogen and, specifically, blue hydrogen.
- 2) The manuscript has its findings based on a few key assumptions that are not sufficiently substantiated or discussed.
- 3) There is a lot of room for improvement for the presentation of figures and text.

1) Wider context

- For instance, the manuscript starts out by stating that hydrogen may potentially play a crucial role in the net-zero transition, but then focusses only on cost reduction throughout the entire rest of the paper without discussing emission savings at all. There exists a wider scientific debate about the role of natural gas in the net-zero transition (e.g. doi:10.1038/s41467-023-41105-z), about the impact of methane leakage (e.g. doi:10.1038/s41467-023-41527-9), and about how clean blue hydrogen actually is (doi:10.1002/ese3.956, doi:10.1002/ese3.1126, doi:10.1002/ese3.1154). I would expect a manuscript aiming to be published in Nature Communications to at least briefly touch upon those points and to present a balanced and nuanced picture of the role of blue hydrogen that goes beyond cost (even though I understand that this is the focus of the manuscript). I suggest the authors add at least a short discussion on the supply-chain CO₂ and CH₄ emissions of natural gas (and coal, although this is perhaps not as relevant) in the introduction and/or conclusions sections, perhaps also something on the risks of fossil lock-ins and the risks/opportunities of CCS.
- There is a brief mention of fossil fuel, water, and land resource demand, but with no clear interpretation or comparison.
- Equally, natural gas and coal are discussed alongside each other purely based on a cost perspective, even though these two energy carriers are very dissimilar and also play crucially different roles in the net-zero transition.

- Next, I would expect to see the current project pipeline discussed. It would be very relevant to compare the 10MMTA number to announced blue-hydrogen projects in the US and to the share of announced projects with FID. Or could you at least estimate how many projects of what scale would need to be realised to reach that number?
- Moreover, I would expect some mentioning of other colours of hydrogen, especially green hydrogen. Specifically, it'd be interesting to compare current cost estimates and projections of future cost reductions of green hydrogen (there are loads of studies on this) to those estimated in the presented study. Moreover, a short comparison of emission intensities would be suitable here. Again, discussing project announcements and hence likely trends of installed capacity until 2030 would be relevant to compare.
- A big obstacle for hydrogen and large-scale industrial projects is still the lack of technological certainty. Many announced projects (see for instance the IEA hydrogen project database) still lack an FID. Perhaps the authors could discuss this in the US context and if/how their results could lay out a clearer plan for not only cost competitiveness but also for security of investment for blue hydrogen projects.
- Finally, I would expect to see more depth added to the conclusions section and for a more explicit outline for competing strategies and policy options (e.g. further subsidies/financial incentives; focussing on maturing one technology only; incentives to build several small vs few big plants; securing investments; speeding up planning procedures; etc).

2) Key assumptions and validity of results

- The key results of the manuscript crucially depend on the learning rates assumed for the cost reductions of the individual components. The authors assume these values from two primary sources (Rubin et al from 2007, and Schoots et al from 2008). These cited studies are now more than 15 years old and the data they refer to ranges from the 60s to the early 2000s. I wonder how reliable and valid the derived learning curves are for today's SMR and CO₂ capture technologies. I understand that these numbers are very tricky to derive with certainty, and one often has to resort to regressions of historical data. Yet, given that this is such a key assumption of this manuscript, I would expect a much more detailed analysis and discussion of these assumed parameters. My first impression is that a learning rate for SMR is somewhat high, given that this is a rather established technology.
- Also note that modern plant designs will have to ensure minimal CH₄ leakage (less than 0.1%), which will necessitate extra efforts and could thereby easily drive up SMR cost and hence offset other learning effects.
- Moreover, it would be interesting if the manuscript was able to differentiate more clearly, how learning-by-doing could drive down the cost of those specific components and if/how R&D could support this effort.
- The article focusses on SMR as the main technology for blue hydrogen. However, at least to my understanding, the most suitable technology for blue hydrogen with high capture rates would be ATR. Can the authors comment on why they decided to choose SMR instead of ATR?

3) Presentation of figures and text

- As mentioned above, the whole article would greatly benefit from more discussion of context and connecting to the wider scientific debate. Specifically, the introduction and conclusions sections require more contextualisation.
- I don't know how this article is doing on the word-count limit, but I suspect it's rather tight. I would suggest moving a lot more of the technical discussions from the main text (e.g. "Current blue hydrogen production") into the Methods section and really only discuss the results and the key assumptions (learning rates). This would give more space for discussing context rather than stating methodological details or points that the reader can read off from the figures.
- The figures convey little information and don't allow the reader to easily grasp the main results of the manuscript at one glance. I also can't help but noticing that the figures were created in Excel. Often (not always) I find that this does not allow to create advanced plots that can densely present a lot of information in a yet elegant and comprehensible way. For a publication in Nature Communications, I would expect the authors to work hard to create an iconic figure capturing the key

results and conveying these along with an accompanying message/story in a very comprehensible manner. E.g., I would suggest to have one main figure consisting of 1) the pie chart (which is good!), 2) a stacked-bar waterfall diagram for each technology demonstrating the different components of cost savings at 10 MMTA installed capacity and with the tax credits, and 3) the learning curves of the main assumptions without the point markers (or do they serve a purpose?) combined with shaded uncertainty corridors indicating the results of the sensitivity analysis (only gas price OR learning rate OR perhaps both combined).

In summary, the authors investigate a very relevant and topical research question, but the way in which their manuscript is currently written, it doesn't match the portfolio of Nature Communications and would be more suitable for a more field-specific technical journal. Irrespective of whether the authors choose to resubmit to NComms, I would encourage them to substantially revise their paper by: a) adding more context, b) pointing out key take-aways more clearly, c) improving the graphical presentation of their results, and d) more thoroughly discussing/substantiating the key assumptions of their work (mainly the learning rate).

Some further points/comments/questions/remarks:

- What is your assumption for the interest rate/WACC? (I can't remember reading it and couldn't find it with Ctrl+F.) I wonder if the assumed WACC could also be a basis for another sensitivity analysis.
- My understanding is you are currently deducting the tax credit from the levelised cost, i.e. thereby spreading the credit across the assumed book lifetime. Perhaps it would make sense to rather discuss in more detail what will happen when the tax credit ceases? I.e. the later that projects start, the less of the credit period they will be able to obtain. Your current main figures have product capacity on the lower axis. If you assume a certain deployment rate, you could plot these figures as a function of time. The resulting figure would allow to see the trade-off between waiting till the production cost drops lower via learning vs. less tax credit (I assume the tax credit would still be the much bigger effect, hence indicating it'd be more profitable to construct now rather than later).
- Two main takeaways for me from the manuscript are: a) substantial cost reductions via "learning-by-doing" can only be achieved if technologies are sufficiently deployed, and b) coal-based blue hydrogen seems to be uncompetitive compared to gas-based. Wouldn't it therefore be sensible to recommend focussing policy support/industry efforts on gas-based blue hydrogen to reach sufficient deployment and hence cost reductions of this one technology instead of splitting up efforts between the two options?
- Eq. (6) looks odd. Shouldn't the term proportional to the VOM be independent of the CF? I.e. there is no cost for fuel consumption during the period when the plant is not producing.
- Irrespective of what you choose to do with your figures, could you make sure the x-axis ticks are the same across all of them? E.g. it's hard to find the 1\$/kg marker. Perhaps you could also add a dashed horizontal line to that graph?
- Given you're trying to compare cost reductions against the 1\$/kg target, it'd be valuable to discuss the role of inflation. I noticed you're using \$2018 numbers from the NETL study. It's almost a bit of a silly question because I presume the 1\$/kg target is more of a political slogan than the result of an elaborate economic analysis, but I wonder how much inflation today and in coming years would make it harder to reach that specific nominal value.
- Can you say more explicitly what is meant by "constant dollars"?
- Typo: summery

Reviewer #3:

Remarks to the Author:

This paper looks at the projected cost of blue hydrogen as a function of different learning rates, taking into account the IRA tax credits for hydrogen. It relies on a recent comprehensive report by NETL, which provides process level comparisons of state-of-the-art, fossil-based hydrogen production technologies. The authors use their analysis to assess the likelihood of reaching the US Hydrogen

Earthshot goal of hydrogen at 1 \$/kg by 2030.

Given the importance of hydrogen in future global energy systems, and the uncertainties around technology costs, understanding the cost evolution of hydrogen technologies is a useful and timely addition to the literature. The paper focusses heavily on learning rates and gas prices. However, the novelty of the analysis and depth of the discussion could be improved.

Firstly, the assumptions for learning rate seem simplistic and lack a discussion/justification of how they apply to an industry that is looking to evolve a mature technology: SMR is a commercially mature process, but coupling it with CCS is not. The discussion of learning rate discovery is limited to two sentences in the methods section which simply states that 'learning rates are mainly collected from two highly-cited learning curve articles'. In particular, the authors mention the option of retro fitting existing SMR plants briefly in the discussion, but do not include this important pathway in their analysis. The manuscript needs a more in-depth discussion of how learning rates can be applied and developed to take the maturity of certain sub-systems, and the options for retro fitting into account. I note that the authors do seem to apply different learning rates to different subsystems, but do not explain if/how they considered the varying maturity of the sub-systems.

Secondly, they identify gas prices as a critical parameter, but do not take the opportunity to discuss how a large blue hydrogen industry could distort or affect the gas markets in the US – essentially providing a large capacity market for US gas as its usage in other sectors declines. Given that they are calculating large cumulative hydrogen production capacities, this would seem to be a very salient point.

Both of the points mentioned above are critical to understand the implications of the findings of the analysis in the paper, and such discussion is expected in high impact, interdisciplinary journals like Nat. Comms.

Thirdly, the authors do not accurately reflect the LCA emissions for blue hydrogen in the paper. The authors provide a range of emissions intensities for blue hydrogen, then state in the supplementary that 'In this study, all hydrogen production facilities are assumed to be eligible for the bonus rate of tax credit of Section 45V'. However, the NETL report that they cite for emissions data clearly states that the median LCA emissions intensities will exceed the 4 kgCO₂/kgH₂ limit, and hence that blue hydrogen production will not necessarily qualify for the 45V tax credit. The authors never acknowledge this and the LCA emissions are never directly compared to the requirements for the 45V tax credit (which is tucked away in the supplementary in Table 13). This is a crucial oversight at best, and misleading at worst.

I recommend that the authors revise the manuscript to take account of the issues mentioned above, and my detailed comments below, before resubmission.

Detailed comments

Too much of the critical information and context is buried in the supplementary. While I recognise that it is necessary to keep to word limits, the authors should provide more of an overview of the system boundaries they consider in the manuscript itself. This includes a discussion of how learning rates are chosen and applied as discussed above. For example, the NETL report assumes that CO₂ and H₂ exit the plants at particular pressures, and take into account the energy and plant required for this. It is important to state these assumptions and clarify why they are chosen.

Autothermal reforming is projected to be a cost-effective blue hydrogen technology in many other studies as well as the IEA estimates. The authors should justify why they chose not to include it – especially since the NETL report provides data on ATR plants.

The authors should provide hydrogen production capacity in such a way that it can be compared between data sets. Table 1 quotes H₂ production capacity in cubic meters per hour; Table 2 in GWth; and Figures 1-3 in Mt H₂/year. (I note that Table S12 has both Mt/yr and GWth). Given the focus on learning rates as a function of installed capacity, it is useful for the reader to compare these results to the existing capacity. The authors should clearly provide current installed capacities as Mt H₂/y.

Blue hydrogen plants tend to be large, so even a large production capacity increase could represent a small number of new plants, and hence limited learnings. Is this taken into account in the learning rate analysis?

Given that the aim is to assess the likelihood of reaching the US Hydrogen Earthshot goal of hydrogen at 1 \$/kg by 2030, the authors should comment on how long it would take to reach the installed capacities required to drive the costs down – would this occur within the next 7 years? This is obviously a difficult task, but should be at least mentioned. For example, how many blue hydrogen plants that would need to be deployed? how quickly can blue hydrogen plants typically be deployed? what is the current pipeline of announced projects?

The authors state that 'the cost of blue hydrogen produced by SMR with CCS approximates to the Hydrogen Earthshot, as shown in Fig. 1e.', however, this is difficult to see on the graph. Suggest that the authors add a line to indicate the earth shot cost – as is done in Fig 2 b-d and Fig 3.

Issues with range of LCA given and discussion around tax credit 45V:

The authors state that "the production tax credit for hydrogen projects is \$0.6 per kilogram of H₂ for 10 years as the life cycle GHG emissions of gas- and coal-based production plants for blue hydrogen are estimated to be 3.13 to 8.86 and 3.40 to 8.87 kg CO₂-equivalent per kilogram of hydrogen, respectively". However, looking into the NETL report, the median is 4.6 kgCO₂/kgH₂ (Exhibit 3-52, pg 126). The assumption that blue hydrogen plants will get the 45V tax credit is clearly flawed, and the lack of transparency around this is misleading. One way to get round this would be to 'look under the hood' of the NETL analysis and highlight under which conditions blue hydrogen plants could meet the emissions requirements. This would correct the misleading assumption and improve the utility of the analysis in the paper.

The comments from all the three reviewers are very valuable, which helped us significantly improve the manuscript to reach a new height of quality for publication. To express our sincere appreciation, we expanded the acknowledgement section by adding: “In addition, the authors also want to deliver special thanks to the three anonymous reviewers for their great comments, which improved the quality of this study during the revision process.”

REVIEWER COMMENTS

Reviewer #1 (Remarks to the Author):

Comment: The authors present an analysis on future blue hydrogen costs in the US based on different technologies, learning rates and natural gas prices. They compare the effect of two different tax credits that are in place, in the perspective of reaching a target cost of 1 USD/kg for blue hydrogen.

Response: The authors greatly appreciate the reviewer for time and valuable comments. Noted.

Comment: The results are of interest for the research community, given the importance of the topic. Unfortunately, I believe that the present version of the manuscript has some important issues, and it is not suitable for publication on an international journal such as *Nature Communications*. I report my comments below.

Response: As the reviewer pointed out, this manuscript addresses a timely important topic on low-carbon hydrogen production and provides important results, findings, and new insights, which are of interest to the community in this field. We have performed a large amount of additional quantitative and qualitative analyses, discussions, and clarifications to address the reviewer and other two reviewers’ critical comments, which have helped us significantly improve the manuscript. We think the revised manuscript is a fit for broad readership of multidisciplinary journals like *Nature Communications*.

Comment: The economic analysis provided by the authors is detailed and include many different sub-systems and parameters. However, in a general perspective, I believe that other very important aspects are not discussed at all. The crucial point is the CO₂ emission savings that can be achieved by blue hydrogen, especially compared to green hydrogen, which appears to be the elephant in the room in this analysis. The authors briefly mention lifecycle emission ranges at page 11, but no additional information is given nor discussed. I think indeed this is a crucial point that should be addressed to support the main results. The authors could also estimate a cost per avoided tonne of CO₂ emission to be compared to other technologies and studies.

Response: We appreciate the reviewer for pushing us to think more broadly about this critical topic. We collected additional data of stack CO₂ emissions from the production of grey hydrogen, blue hydrogen, and green hydrogen to quantify the CO₂ emission savings by blue hydrogen relative to grey hydrogen and further compare it with green hydrogen. We further estimated the CO₂ avoidance costs by blue and green hydrogen and made a comparison between the two low-carbon hydrogen production pathways. Additional information of life cycle emissions was added to the main paper.

While revising the main paper in response to this comment, we added to the end of Section Current Blue Hydrogen Production a new paragraph: “Currently, hydrogen is mainly produced by SMR without CCS in the U.S., which is often called grey hydrogen. Compared to it, the blue hydrogen production by SMR with CCS can decrease the stack CO₂ emission intensity by 96% but increase the LCOH by 55%⁷. The resulting CO₂ avoidance cost by blue hydrogen is \$65 per metric ton of CO₂. In contrast, the green hydrogen production by polymer electrolyte membrane electrolyzers almost has no stack CO₂ emissions but a high LCOH value ranging from \$3.0–7.5/kg H₂²⁷. The resulting CO₂ avoidance cost by green hydrogen relative to grey hydrogen varies from \$212–689 per metric ton of CO₂, which is much higher than that by blue hydrogen. Obviously, there are tradeoffs in CO₂ avoidance cost and emission savings between the blue and green production pathways. The details of emission and cost data and CO₂ avoidance cost estimation are available in Supplementary Note 2.”

We added a new supplementary section:

Supplementary Note 2: Cost of CO₂ Avoided by Blue and Green Hydrogen

The cost of CO₂ avoided by blue or green hydrogen relative to grey hydrogen can be estimated in terms of the production plant’s CO₂ emission intensity and levelized cost of hydrogen:

$$CCA = \frac{LCOH_{\text{blue/green}} - LCOH_{\text{grey}}}{EI_{\text{grey}} - EI_{\text{blue/green}}} \quad (S1)$$

Where CCA is the cost of CO₂ avoided (\$/kg CO₂); LCOH is the levelized cost of hydrogen (\$/kg H₂); EI is the stack CO₂ emission intensity of a production plant (kg CO₂/kg H₂); the subscript of “blue” represents the gas-based blue hydrogen production, the subscript of “green” represents the green hydrogen production, and the subscript of “grey” represents the gas-based grey hydrogen production without carbon capture.

Supplementary Table 5 summarizes the stack CO₂ emission intensity and LCOH of grey, blue, and green hydrogen production plants, which are collected from the literature^{1,4}. The resulting costs of CO₂ avoided by blue and green hydrogen relative to grey hydrogen are also provided in Supplementary Table 5.

Supplementary Table 5. Hydrogen Production Emission, Production Cost, and Carbon Avoidance Cost

Production Technology	Stack Emission Intensity (kg CO ₂ /kg H ₂)	Levelized Cost of Hydrogen (2018\$/kg H ₂)	Cost of CO ₂ Avoided (\$/metric ton CO ₂)
Steam Methane Reforming (SMR) (Grey)	9.35	1.06	Reference
SMR with CCS (Blue)	0.38	1.64	65
Distributed Polymer Electrolyte Membrane Electrolyzers (Green)	0 ^a	5.17 (3.0–7.5) ^b	440 (212-689)

^aZero carbon emissions are assumed for the electrolysis powered by solar photovoltaics as it produces almost no direct emissions during operation⁵.

^bThe cost was estimated for a production capacity of 1,500 kg H₂/day with an effective electricity price of 7.55 ¢/kWh⁴. The cost was further adjusted to 2018 dollars using the annual average consumer price index for all urban consumers⁶.

To clarify the life cycle emission estimation in detail, we expanded the statement in Page 11 of the original version to read: “As mentioned earlier, the 45V tax credit depends on the life

cycle emissions of hydrogen production, which include greenhouse gas emissions from plant stacks, fuel supply, electric power supply, and CO₂ sequestration or management. The life cycle emissions were estimated by the National Energy Technology Laboratory to range from 3.1 to 8.9 kg CO₂-eq/kg H₂ for the gas-based blue hydrogen in the 90% confidence interval between the 5th and 95th percentile values and from 3.4 to 8.9 kg CO₂-eq/kg H₂ for the coal-based blue hydrogen, which is driven mainly by the uncertainty in fuel supply⁷. The largest contributor among the multiple stages to the life cycle emissions is the fuel supply⁷. The median estimate of life cycle emissions is 4.6 kg CO₂-eq/kg H₂ for the gas-based blue hydrogen and 4.1 kg CO₂-eq/kg H₂ for the coal-based blue hydrogen⁷, which is close to the threshold value of 4.0 kg CO₂-eq/kg H₂ required to claim the minimum tax credit for clean hydrogen. Blue hydrogen projects have a fair possibility of earning a 45V tax credit. Thus, the production tax credit for hydrogen projects is assumed to be \$0.6 per kilogram of H₂ for 10 years. This assumption is optimistic for blue hydrogen in this study. However, there is no 45V tax credit if the life cycle emissions of specific blue hydrogen projects are more than 4.0 kg CO₂-eq/kg H₂. See Supplementary Note 5 for additional information about life cycle emissions and tax credits.”

Comment: The literature review seems quite limited, especially given the large number of studies on the subject in the last years. The learning rates reported by the authors (page 5) rely on literature that is more than 15 years old, which seems quite outdated considering the recent technological evolution. I think that using more recent data would improve the quality and reliability of the results.

Response: This study mainly cited the two pioneering learning curve studies by Rubin et al. (2007) and Schoots et al. (2008). Although a large number of technological learning studies have been done in the past years, almost no breakthroughs have been made on learning rates beyond the two pioneering studies in the past years. The learning rates from the two pioneering studies have been frequently adopted in the recent studies. In other words, there are no more rigorous estimates of learning rates in the literature than the two pioneering studies cited in our manuscript. To make it clear, we added to the relevant paragraph in the introduction section on Page 5 of the original version new sentences:

“Although a large number of learning curve studies have been done in the past years, almost no breakthroughs have been made in estimating the learning rates for CCS, gasification, and SMR beyond the two pioneering studies by Rubin et al. (2007)¹⁴ and Schoots et al. (2008)¹³. Recent studies frequently adopted the learning rates from the two pioneering studies for a variety of applications¹⁷⁻²⁵.”

The added references include:

- Böhm, H., Zauner, A., Rosenfeld, D. C. & Tichler, R. Projecting cost development for future large-scale power-to-gas implementations by scaling effects. *Appl. Energy* **264**, 114780 (2020).
- Fan, J. L., Li, Z., Li, K. & Zhang, X. Modelling plant-level abatement costs and effects of incentive policies for coal-fired power generation retrofitted with CCUS. *Energy Policy* **165**, 112959 (2022).
- Fan, J. L., Xu, M., Yang, L., Zhang, X. & Li, F. How can carbon capture utilization and storage be incentivized in China? A perspective based on the 45Q tax credit provisions. *Energy Policy* **132**, 1229–1240 (2019).

- George, J. F., Müller, V. P., Winkler, J. & Ragwitz, M. Is blue hydrogen a bridging technology? The limits of a CO₂ price and the role of state-induced price components for green hydrogen production in Germany. *Energy Policy* **167**, 113072 (2022).
- Kang, J. N., Wei, Y. M., Liu, L., Han, R., Chen, H., Li, J., Wang, J. W. & Yu, B. Y. The prospects of carbon capture and storage in China's power sector under the 2 °C target: a component-based learning curve approach. *Int. J. Greenhouse Gas Control* **101**, 103149 (2020).
- Lee, H., Lee, J. & Koo, Y. Economic impacts of carbon capture and storage on the steel industry - A hybrid energy system model incorporating technological change. *Appl. Energy* **317**, 119208 (2022).
- Malhotra, A., & Schmidt, T. S. Accelerating low-carbon innovation. *Joule* **4**, 2259–2267 (2020).
- Nicodemus, J. H. Technological learning and the future of solar H₂: A component learning comparison of solar thermochemical cycles and electrolysis with solar PV. *Energy Policy* **120**, 100–109 (2018).
- Yang, L., Xu, M., Yang, Y., Fan, J. & Zhang, X. Comparison of subsidy schemes for carbon capture utilization and storage (CCUS) investment based on real option approach: Evidence from China. *Appl. Energy* **255**, 113828 (2019).

Comment: Furthermore, CCS in real applications has shown a lower economic performance than expected, and I think this potential issue should be addressed in the discussion and maybe also in the sensitivity analysis.

Response: There are uncertainties in CCS cost estimates. As the reviewer suggested, we performed a parametric analysis on CCS systems' process and project contingences and added the relevant analysis, results, and discussion to the sensitivity analysis section:

“There are uncertainties in the process and project contingences of two CO₂ removal systems employed for producing low-carbon hydrogen from natural gas resources. Such uncertainties affect the total as-spent capital (TASC) and LCOH of a hydrogen production plant. The process contingency depends on the maturity level of a technology, whereas the project contingency depends on the availability of site-specific project details. In the base case, the process contingency is 18% of the bare erected cost (BEC) for the Cansolv system and 0% for the MDEA system, while the project contingency is 25% of the sum of BEC, engineering, construction management, home office and fees, and process contingency and 25% for the Cansolv unit and 20% for the MDEA unit⁷. A parametric analysis is then conducted to reveal the collective impacts of uncertain process and project contingencies, which takes into account low and high contingencies. In the low contingencies scenario, the process contingency is 10% for the Cansolv system and 0% for the MDEA system, while the project contingency is 10% for both the CO₂ removal systems. In the high contingencies scenario, the process contingency is 40% for both the CO₂ removal systems, while the project contingency is 30% for both the CO₂ removal systems³⁴.

As shown in Figure 4, the uncertainties in CCS cost estimates have a sizable effect on the hydrogen production plant's TASC. As a result, the plant LCOH varies from \$1.45-1.48/kg H₂ at

the cumulative installed capacity of 10 MMTA. To reach the cost of \$1.46/kg H₂, the cumulative installed capacity requirements vary from 7 to 16 MMTA. These results imply that cost uncertainties in CCS systems may result in pronounced variations in the estimation of the cumulative installed capacity necessary to reach a cost target.”

Fig. 4 Effects of process and project contingencies of CO₂ removal systems on initial capital cost and future levelized cost of gas-based blue hydrogen production.

Comment: An additional issue about CCS is that the authors do not seem to consider any investment cost for storage (tables 10 and 11). I think this is a major issue, as these systems should be built, and are also expected to require an important part of the tax credit that the authors are completely allocating to blue hydrogen production. This aspect should be better discussed in the paper, especially considering the important debate about the suitability of CCS.

Response: The investment in CO₂ storage was considered. Both the CO₂ transport and storage costs were considered in this study and were treated as O&M costs (\$/metric ton CO₂) in the overall cost estimation instead of a capital cost component, which are shown in Supplementary Tables 10 and 11 of the original manuscript. To make it clearer, we added to Supplementary Table 1 the total CO₂ transport and storage cost: “total CO₂ transport and storage cost: \$10/metric ton CO₂.”

Comment: The authors consider a tax credit that can be claimed for lifecycle GHG emissions lower than 4 kgCO₂/kgH₂. However, the GHG emission ranges that they provide at page 11 show median values that are much higher than 4. How do they justify this assumption? Please clarify this issue.

Response: The median values of life cycle emissions are close to the threshold value of 4.0 kg CO₂-eq/kg H₂. As we addressed in response to a similar comment above, we expanded the discussion with additional information: “The life cycle emissions were estimated by the National Energy Technology Laboratory to range from 3.1 to 8.9 kg CO₂-eq/kg H₂ for the gas-based blue hydrogen in the 90% confidence interval between the 5th and 95th percentile values and from 3.4 to 8.9 kg CO₂-eq/kg H₂ for the coal-based blue hydrogen, which is driven mainly by the uncertainty in fuel supply⁷. The largest contributor among the multiple stages to the life cycle emissions is the fuel supply⁷. The median estimate of life cycle emissions is 4.6 kg CO₂-eq/kg H₂ for the gas-based blue hydrogen and 4.1 kg CO₂-eq/kg H₂ for the coal-based blue hydrogen⁷,

which is close to the threshold value of 4.0 kg CO₂-eq/kg H₂ required to claim the minimum tax credit for clean hydrogen. Blue hydrogen projects have a fair possibility of earning a 45V tax credit. Thus, the production tax credit for hydrogen projects is assumed to be \$0.6 per kilogram of H₂ for 10 years. This assumption is optimistic for blue hydrogen in this study. However, there is no 45V tax credit if the life cycle emissions of specific blue hydrogen projects are more than 4.0 kg CO₂-eq/kg H₂.”

Comment: The results of the authors are expressed with respect to the potential blue hydrogen deployment capacity. How these capacities compare to the expected demand of blue hydrogen by 2030? I think this information is fundamental to put the results in the right perspective.

Response: The U.S. Department of Energy’s National Clean Hydrogen Strategy and Roadmap report provides an estimate of total national clean hydrogen from both renewable and decarbonized fossil resources by 2030 instead of blue hydrogen alone. To address this comment on the broad perspective, we added to the relevant results on Page 10 a new sentence: “The annual demand for clean hydrogen produced from renewable and decarbonized fossil resources in the U.S. may reach 10 million metric tons of hydrogen per year by 2030¹.”

We further added to the Discussion section a new paragraph:

“The global production capacity of low-carbon hydrogen will reach 12.3 MMTA by 2030 based on the announced, planning and committed projects³⁰. The low-carbon hydrogen capacity in North America will reach 6.8 MMTA by 2030³⁰. However, only 1.8 MMTA³⁰ and 1.5 MMTA^{1,2} of the announced projects in North America and the U.S. have reached final investment decision (FID), mainly because many announced projects have not yet secured financing and nailed down contracted offtake^{1,2}. The hesitancy to long-term, scaled contracts is influenced by numerous factors, such as lack of price certainty, unavailability and reliability of large-scale hydrogen supply, near-term policy implementation uncertainty, and long-term political uncertainty^{1,2}. For blue hydrogen projects, enhancements in tax credits for carbon sequestration can improve the economics of hydrogen production. For example, an extension of the 45Q tax credit period from current 12 years to 18 years would significantly reduce the cumulative installed capacity required for gas-based blue hydrogen projects to reach the Hydrogen Energy Earthshot, as demonstrated in Supplementary Fig. 3. Extending the period of the 45Q tax credit for blue hydrogen projects can be considered an option to secure financing and promote long-term offtake.”

Supplementary Fig. 3. Effect of 45Q tax credit period on future cost of gas-based blue hydrogen production.

Comment: The sensitivity analysis (or at least the discussion) may also address the potential effect of different levels of inflation (and possibly other economic parameters), given the significant variations over the last years. Would this affect the reliability of the results?

Response: The \$1/kg target is the U.S. DOE’s initiative goal, which has no specific information regarding cost type (real vs. nominal dollars) and inflation. This study reports cost results in real dollars. To address this comment, we conducted an additional parametric analysis and estimated the cost results in nominal dollars to demonstrate the effect of inflation rate on the future cost of hydrogen production toward the Hydrogen Energy Earthshot. The results imply that inflation would remarkably raise challenges for blue hydrogen production to reach the cost target. We added to the Sensitivity Analysis section the new analysis and relevant results:

“Inflation Rate

In general, this study estimates the cost of hydrogen production in real dollars. When the cost is estimated in nominal dollars, however, both the initial and future LCOH estimates vary with inflation rate as it affects discount rate, fixed charge rate, and levelization factor. A parametric analysis was further performed for inflation rate to quantify its effect on the evolving cost of gas-based blue hydrogen production toward the Hydrogen Energy Earthshot. Fig. 6 shows the learning curves of blue hydrogen production with inflation. Fig. 6a and 6b show that at a given level of cumulative installed capacity, the LCOH in nominal dollars increases when the inflation rate increases from 1% to 3%. As a result, blue hydrogen production may not reach the cost target of \$1/kg H₂ for both the scenarios without and with 45Q tax credit even when the cumulative installed capacity reaches 30 MMTA. Fig. 6c further shows that with an inflation rate of 3%, the future LCOH may get close to the cost target when cheap natural gas resources are used as the feedstock to produce blue hydrogen with the cumulative installed capacity of up to 30 MMTA.

Fig. 6a and 6b also compare the learning curves of blue hydrogen production between the two scenarios without and with inflation. As shown in Fig. 6a for the scenario without 45Q tax credit, the reduction in hydrogen production cost from deploying the cumulative installed capacity of 10 MMTA can be offset by an inflation rate of 1%. There is a similar result for the scenario with 45Q tax credit, as shown in Fig. 6b. All these results imply that inflation would

remarkably raise challenges for blue hydrogen production to reach the Hydrogen Energy Earthshot in the near future.”

Fig. 6 Effect of inflation rate on future cost of gas-based blue hydrogen production without and with 45Q tax credit. (a) Levelized cost of hydrogen production with gas price of \$4.2/GJ and without 45Q tax credit; (b) Levelized cost of hydrogen production with gas price of \$4.2/GJ and 45Q tax credit; (c) Levelized cost of hydrogen production with 3% inflation rate and 45Q tax credit.

To explain how to estimate the LCOH in nominal dollars, we added a new supplementary session:

“Supplementary Note 8: Hydrogen Production Cost Estimation in Nominal Dollars

The levelized cost of hydrogen and is estimated in nominal dollars¹:

$$\text{LCOH}_N = \text{LCC}_N + \text{LOM}_N + \text{LFP}_N \quad (\text{S12})$$

$$\text{LCOH}_N^{\text{TC}} = \text{LCC}_N + \text{LOM}_N + \text{LFP}_N - \text{LTC}_N^{45\text{Q}} \quad (\text{S13})$$

Where the subscript “N” means the nominal dollars; LCOH_N is the levelized cost of hydrogen (\$/kg H₂); LCC_N is the levelized capital cost (\$/kg H₂); LOM_N is the non-fuel levelized operating and maintenance cost (\$/kg H₂); LFP_N is the levelized fuel price (\$/kg H₂); $\text{LCOH}_N^{\text{TC}}$ is the levelized cost of hydrogen with a tax credit (\$/kg H₂); $\text{LTC}_N^{45\text{Q}}$ is the levelized tax credit over the project book lifetime of a blue hydrogen project (\$/kg H₂).

The levelized capital cost in nominal dollars is estimated using Supplementary Equation (14), while the fixed charge rate is estimated using Supplementary Equations (15) to (21)^{2,3,15}:

$$\text{LCC}_N = \frac{\text{TASC}_R \cdot \text{FCR}_N}{(\text{CF} \cdot \text{AH}) \cdot \text{KG}_{\text{H}_2}} \quad (\text{S14})$$

$$\text{FCR}_N = \frac{\text{CRF}_N^{\text{nonfuel}} - \text{ETR} \cdot \text{PV}_{\text{plant}}}{1 - \text{ETR}} \quad (\text{S15})$$

$$\text{CRF}_N^{\text{nonfuel}} = \frac{\text{ATWACC}_N \cdot (1 + \text{ATWACC}_N)^{\text{BL}}}{(1 + \text{ATWACC}_N)^{\text{BL}} - 1} \quad (\text{S16})$$

$$\text{PV}_{\text{plant}} = \text{CRF}_N^{\text{nonfuel}} \cdot \sum_{n=1}^m \frac{d_n}{(1 + \text{ATWACC}_N)^n} \quad (\text{S17})$$

$$\text{ATWACC}_N = \text{PC}_{\text{equity}} \cdot \text{ROE}_N + \text{PC}_{\text{debt}} \cdot \text{kd}_N \cdot (1 - \text{ETR}) \quad (\text{S18})$$

$$\text{ROE}_N = (1 + \text{ROE}_R) \cdot (1 + N) - 1 \quad (\text{S19})$$

$$\text{kd}_N = (1 + \text{kd}_R) \cdot (1 + N) - 1 \quad (\text{S20})$$

$$N = R + I \quad (\text{S21})$$

Where the superscript “nonfuel” represents the non-fuel component; TASC_R is the total as-spent capital of a blue hydrogen production plant (\$); FCR_N is the fixed charge rate (fraction/year); CF is the plant capacity factor (%); AH is the total annual hours (8760 hours); KG_{H_2} is the hourly production rate (kg H₂/hour); CRF_N is the capital recovery factor (fraction/year); ETR is the effective tax rate (%); PV_{plant} is the present value of tax depreciation expense (fraction/year); ATWACC_N is the after-tax weighted average cost of capital (%); BL is the project book lifetime (year); d_n is the tax depreciation fraction in year n (fraction); m is the number of years of depreciation (year); $\text{PC}_{\text{equity}}$ is the percent of equity (%); ROE_N is the nominal rate of return on equity (%); PC_{debt} is the percent of debt (%); kd_N is the nominal rate of cost of debt; ROE_R is the real rate of return on equity (%); kd_R is the real rate of cost of debt (%); N is the nominal escalation rate (%); R is the real escalation rate (%); and I is the inflation rate (%).

The levelized operating and maintenance cost in nominal dollars is estimated using Supplementary Equation (22), while the levelization factor is estimated using Supplementary Equation (23):

$$\text{LOM}_N = \frac{\text{OM}_R \cdot \text{LF}_N}{(\text{CF} \cdot \text{AH}) \cdot \text{KG}_{\text{H}_2}} \quad (\text{S22})$$

$$LF = CRF_N^{\text{nonfuel}} \cdot \frac{1 - \left[\frac{1 + N}{1 + ATWACC_N} \right]^{BL}}{ATWACC_N - N} \quad (S23)$$

Where OM_R is the non-fuel operating and maintenance (O&M) cost of a blue hydrogen production plant (\$/year); and LF_N is the levelization factor for non-fuel O&M cost (unitless).

The levelized fuel price in nominal dollars is estimated using Supplementary Equation (24), where the fuel's capital recovery factor is estimated using Supplementary Equation (26) and the discount rate in nominal dollars is estimated using Supplementary Equation (27)¹⁶:

$$LFP_N = \frac{FC_N \cdot FR}{KG_{H_2}} \quad (S24)$$

$$FC_N = PV_{\text{fuel}} \cdot CRF_N^{\text{fuel}} \quad (S25)$$

$$CRF_N^{\text{fuel}} = \frac{DR_N \cdot (1 + DR_N)^{BL}}{(1 + DR_N)^{BL} - 1} \quad (S26)$$

$$DR_N = (1 + DR_R) \cdot (1 + N) - 1 \quad (S27)$$

$$PV_{\text{fuel}} = \sum_{t=1}^{BL} \frac{P_t}{(1 + DR_N)^t} \quad (S28)$$

$$P_t = P_{t-1} \cdot (1 + N) \quad (S29)$$

Where the superscript “fuel” represent the fuel component; FC_N is the natural gas cost (\$/GJ); FR is the hourly natural gas flow rate (GJ/hour); PV_{fuel} is the present value of gas cost (\$/GJ); CRF_N^{fuel} is the fuel's capital recovery factor (fraction/year); DR_N is the nominal discount rate for fuel cost (%); DR_R is the real discount rate for fuel cost (%); N is the nominal escalation rate (%); P_t is the fuel cost in year t (\$/GJ), while P_1 is the real fuel cost in the first year of a project.

In this study, the financial structure and assumptions are in alignment with those of the studies by the National Energy Technology Laboratory (NETL)^{1,2,3,16,17}. Supplementary Table 17 lists the financial parameters and summarizes their data and the sources of data. Supplementary Table 18 summarizes the estimates of FCR, levelization factor, and discount rate as a function of nominal escalation rate.

Supplementary Table 17. Financial Parameters and Assumptions

Parameter	Symbol	Unit	Value	Source(s)
Effective Tax Rate	ETR	%	25.74%	NETL (2021) ³
Inflation Rate	I	%	0–3%	Assumption
Real Escalation Rate	R	%	0%	NETL (2011) ²
Percent of Equity	PC _{equity}	%	62%	NETL (2022) ¹
Percent of Debt	PC _{debt}	%	38%	NETL (2022) ¹
Real Rate of Return on Equity	ROE _R	%	3.10%	NETL (2022) ¹
Real Rate of Cost of Debt	kd _R	%	5.15%	NETL (2022) ¹
Real Discount Rate	DR _R	%	4.73%	NETL (2019) ¹⁶
Project Book Lifetime	BL	year	30	NETL (2022) ¹
Fuel Cost	P ₁	2018\$/GJ	4.2	NETL (2022) ¹
Tax Depreciation Fraction	d ₁	%	3.75%	NETL (2021) ³ ; IRS (2016) ¹⁷
	d ₂		7.22%	
	d ₃		6.68%	
	d ₄		6.18%	
	d ₅		5.71%	

	d ₆		5.29%	
	d ₇		4.89%	
	d ₈		4.52%	
	d ₉		4.46%	
	d ₁₀		4.46%	
	d ₁₁		4.46%	
	d ₁₂		4.46%	
	d ₁₃		4.46%	
	d ₁₄		4.46%	
	d ₁₅		4.46%	
	d ₁₆		4.46%	
	d ₁₇		4.46%	
	d ₁₈		4.46%	
	d ₁₉		4.46%	
	d ₂₀		4.46%	
	d ₂₁		2.23%	

Supplementary Table 18. Financial Parameter Estimates

Parameter	Symbol	Unit	Nominal Escalation Rate			
			0%	1%	2%	3%
Fixed Charge Rate	FCR	fraction/year	0.059	0.067	0.076	0.085
Levelization Factor	LF	fraction	1.000	1.124	1.256	1.393
Discount Rate	DR	%	4.73%	5.78%	6.82%	7.87%

Additional aspects to be addressed:

Comment 1: Please include line numbers when submitting a paper for review.

Response: As suggested, line numbers have been added to the revised manuscript for review.

Comment 2: What is the U.S. Hydrogen Earthshot reported in the title and the abstract? The authors never explicitly define it. Please clarify it for the readers.

Response: We briefly defined it in the first paragraph of the introduction section. However, it looks not clear in terms of the reviewer’s feedback. To address this concern, we expanded the definition to read: “In 2021, the U.S. Department of Energy launched the Energy Earthshots Initiative that aims to accelerate breakthroughs of more abundant, affordable, and reliable clean energy solutions by 2030⁸. To catalyze technological innovation and scale in clean hydrogen, this initiative included a hydrogen shot that aims to decrease the cost of clean hydrogen to \$1 per 1 kilogram in 1 decade⁸, which is called the Hydrogen Energy Earthshot.” To be more consistent with the U.S. DOE’s definition, “Hydrogen Earthshot” was replaced by “Hydrogen Energy Earthshot” throughout the manuscript.

Comment 3: In the introduction, a quick comparison of blue hydrogen with green hydrogen should be provided. Also, I think some information on the actual emissions of current blue hydrogen projects should be illustrated, to inform the readers about the current levels of greenhouse gas emissions associated with this technology. Although they are already reported by the authors in Table 2 in the supplementary material, I think this is a crucial performance indicator of the technology. Also, a description of the contribution of different stages on the lifecycle emissions could be provided.

Response: We added to the first paragraph in the introduction section of the original manuscript a comparative statement: “Although renewable-powered green hydrogen has much less carbon emissions than blue hydrogen, the current cost of green hydrogen production can be several times higher^{6,7}, as shown later.” The newly cited reference is:

IRENA. *Making the breakthrough: Green hydrogen policies and technology costs*. ISBN 978-92-9260-314-4. (International Renewable Energy, 2021).

As in response to Comment 4, we added to Page 6 of the original version the quantitative emissions information of blue hydrogen projects: “In addition, the on-site stack CO₂ emissions from hydrogen production by SMR with CCS are 0.4 kg CO₂-eq/kg H₂, which is much less than that (1.4 kg CO₂-eq/kg H₂) from gasification with CCS⁷.”

As in response to a similar comment on life cycle emissions above, we expanded the statement in Page 11 of the original version to read: “As mentioned earlier, the 45V tax credit depends on the life cycle emissions of hydrogen production, which include greenhouse gas emissions from plant stacks, fuel supply, electric power supply, and CO₂ sequestration or management. The life cycle emissions were estimated by the National Energy Technology Laboratory to range from 3.1 to 8.9 kg CO₂-eq/kg H₂ for the gas-based blue hydrogen in the 90% confidence interval between the 5th and 95th percentile values and from 3.4 to 8.9 kg CO₂-eq/kg H₂ for the coal-based blue hydrogen, which is driven mainly by the uncertainty in fuel supply⁷. The largest contributor among the multiple stages to the life cycle emissions is the fuel supply⁷.”

Comment 4: Page 6: the authors compare the LCOH of SMR and gasification, but I think they should also mention the two different levels of GHG emissions. Gasification is not only more expensive than SMR, as authors correctly highlight, but it also leads to (1) a lower capture rate, (2) a higher net GHG emission intensity compared to SMR.

Response: As suggested, we added to Page 6 of the original version the quantitative emission information: “In addition, the on-site stack CO₂ emissions from hydrogen production by SMR with CCS are 0.4 kg CO₂-eq/kg H₂, which is much less than that (1.4 kg CO₂-eq/kg H₂) from gasification with CCS⁷.”

Comment 5: Page 6: When discussing the cost distribution in subsystems I think that the reference fuel cost for gas and coal should be provided, as their variability significantly affects the shares that are discussed. This information is of course provided by the authors together with the other parameters of the study, and also addressed later in the sensitivity analysis (at least for natural gas), but I think that including it here in the discussion may help the readers in better understanding the results that are described.

Response: As suggested, we added to Page 7 the fuel price information: “Given the gas price of \$4.2/GJ” and “Given the coal price of \$57.3/metric ton.”

Comment 6: The y scale of Figure 1 should report the 1 USD/kg that is the target aimed by the US policy.

Response: As suggested, the y-axis scale was revised to show the target cost of \$1/kg H₂.

Comment 7: Page 18, end of the page: it seems to me quite hard that advanced technologies can

contribute to the cost target by 2030, given the short time horizon. Please clarify this aspect in the discussion.

Response: We added to the paragraph further one more discussion: “However, it may be still challenging for advanced hydrogen technologies to reach the cost target by 2030.”

Comment 8: Page 19: Location of blue hydrogen hubs should also take into account the availability of CO₂ storage sites and related infrastructure. This is an important aspect that is not mentioned by the authors.

Response: Page 19 briefly mentioned this issue. To make this important issue clearer, we revised and expanded the discussion on Page 19 to read: “we revised the interpretation in the Discussion section to read: “Large-scale blue hydrogen production will consume multiple types of natural resources, such as fossil, water and land resources, and geological reservoirs and transportation infrastructure for CO₂ storage, and affect local management and planning of these natural resources. The availability and price of these natural resources vary by region or location in the country. Thus, siting blue hydrogen production plants should take into account co-location and co-availability of these natural resources.”

Comment 9: There are some typos in the manuscript (e.g. page 13, “nature gas”, page 17 “summery”, etc.).

Response: We fixed the typos.

Reviewer #2 (Remarks to the Author):

Comment: The presented manuscript "Technological evolution of large-scale blue hydrogen production toward the U.S. Hydrogen Earthshot" by W. Wu, H. Zhai, and E. Holubnyak studies the role that blue hydrogen will play in the decarbonisation of US energy system and in reaching the US Hydrogen Earthshot. The authors take data from a recent NETL study on the production cost of blue hydrogen and analyse future cost reductions through "learning-by-doing", where they assume exponential learning curves associated with increased technology deployment in coming years.

In summary, the manuscript investigates a highly relevant and timely topic of research. Yet, I see the need for substantial improvement to meet the standards expected by research articles published in Nature Communications.

Response: We agree with the reviewer on the timely importance of this study. We appreciate the reviewer for valuable comments. We made substantial revisions to address the comments to meet the expectations for publication in Nature Communications.

Comment: Here is a list of my main points of criticism:

- 1) The manuscript does not connect to the wider context of and societal/political/scientific debate about hydrogen and, specifically, blue hydrogen.
- 2) The manuscript has its findings based on a few key assumptions that are not sufficiently substantiated or discussed.
- 3) There is a lot of room for improvement for the presentation of figures and text.

Response: We appreciate the reviewer for the criticism, which has helped us significantly improve the manuscript to reach a new height of quality for publication in *Nature Communications*.

(1) To expand the introduction in a broader context, we added to the section a new paragraph:

“Clean hydrogen has the potential to help achieve 10% economy-wide emissions reductions by 2050 relative to 2005, promote energy security and resilience, and develop a new economy in the United States¹. In 2030, the hydrogen economy could create about 100,000 net new jobs for the development of new capital projects and clean hydrogen infrastructure². The U.S. Bipartisan Infrastructure Law has appropriated \$9.5 billion for clean hydrogen for the U.S. Department of Energy (DOE)¹. Both zero- and low-carbon hydrogen production technologies are key options in a diverse toolbox enabling the transition to a sustainable and equitable clean energy future¹. In October 2023, the U.S. DOE announced \$7 billion to launch seven Regional Clean Hydrogen Hubs across the nation³. Some regional hubs will use water and natural gas as the feedstock for renewable-powered electrolysis and steam methane reforming (SMR) with carbon capture and storage (CCS) to produce clean hydrogen, which are also called green and blue hydrogen in practice, respectively. Blue hydrogen is often viewed as a near-term bridge to a zero-carbon hydrogen economy. Given potential high methane leakage, however, there are scientific debates on the competitiveness of blue hydrogen^{4,5}, which make a serious call for methane abatement.”

The cited references include:

- DOE. *U.S. national clean hydrogen strategy and roadmap*. <https://www.hydrogen.energy.gov/docs/hydrogenprogramlibraries/pdfs/us-national-clean-hydrogen-strategy-roadmap.pdf> (U.S. Department of Energy, 2023).

- DOE. *Pathways to commercial liftoff: Clean hydrogen*. <https://liftoff.energy.gov/wp-content/uploads/2023/03/20230320-Liftoff-Clean-H2-vPUB.pdf> (U.S. Department of Energy, 2023).
- DOE. *Biden-Harris Administration Announces \$7 Billion For America's First Clean Hydrogen Hubs, Driving Clean Manufacturing and Delivering New Economic Opportunities Nationwide*. <https://www.energy.gov/articles/biden-harris-administration-announces-7-billion-americas-first-clean-hydrogen-hubs-driving> (U.S. Department of Energy, 2023)
- Howarth, R. W. & Jacobson, M. Z. How green is blue hydrogen? *Energy Sci. Eng.* **9**, 1676–1687 (2021).
- Romano, M. C., Antonini, C., Bardow, A., Bertsch, V., Brandon, N. P., Brouwer, J., Campanari, S., Crema, L., Dodds, P. E., Gardarsdottir, S., Gazzani, M., Gazzani, G. J., Lund, P. D., Dowell, N. M., Martelli, E., Mastropasqua, L., McKenna, R. C., Monteiro, J. G. M., Paltrinieri, N., Pollet, B. G., Reed, J. G., Schmidt, T. J., Vente, J. & Wiley, D. Comment on “How green is blue hydrogen?”. *Energy Sci. Eng.* **10**, 1944–1954 (2022).

(2) We have made significant improvements in the methodology and assumptions as the following specific comments suggested. In particular, please review our responses to Comments 8, 11, 16, and 19.

(3) We have used more specialized data analysis software to re-draw all the figures to improve the quality as suggested. Please review our response to Comment 14.

1) Wider context

Comment 1: For instance, the manuscript starts out by stating that hydrogen may potentially play a crucial role in the net-zero transition, but then focusses only on cost reduction throughout the entire rest of the paper without discussing emission savings at all. There exists a wider scientific debate about the role of natural gas in the net-zero transition (e.g. doi:10.1038/s41467-023-41105-z), about the impact of methane leakage (e.g. doi:10.1038/s41467-023-41527-9), and about how clean blue hydrogen actually is (doi:10.1002/ese3.956, doi:10.1002/ese3.1126, doi:10.1002/ese3.1154). I would expect a manuscript aiming to be published in Nature Communications to at least briefly touch upon those points and to present a balanced and nuanced picture of the role of blue hydrogen that goes beyond cost (even though I understand that this is the focus of the manuscript). I suggest the authors add at least a short discussion on the supply-chain CO₂ and CH₄ emissions of natural gas (and coal, although this is perhaps not as relevant) in the introduction and/or conclusions sections, perhaps also something on the risks of fossil lock-ins and the risks/opportunities of CCS.

Response: This comment raises a nice point in a broader context. To highlight this broader issue, we added to the end of the Discussion section a new paragraph with the suggested references:

“Last but not least, it is very important to prevent upstream methane emissions while reforming natural gas with CCS to produce blue hydrogen. Methane leakage along the natural gas supply chain can jeopardize the role of natural gas in the energy transition to a low-carbon or net-zero future, even when CCS is deployed^{43,44}. A high methane leakage rate of 3.5% or more can elevate the blue hydrogen’s carbon footprint and make it uncompetitive or even unviable in a hydrogen economy⁴. To reduce or avoid the risks of committing to high-emitting blue hydrogen, stringent standards and regulations should be imposed to limit methane leakage and promote deployment of best available technologies for methane abatement⁴⁴.”

The newly cited references are:

- Achakulwisut, P., Erickson, P., Guivarch, C., Schaeffer, R., Brutschin, E. & Pye, S. Global fossil fuel reduction pathways under different climate mitigation strategies and ambitions. *Nat. Commun.* 14, 5425 (2023).
- Shirizadeh, B., Villavicencio, M., Douguet, S., Trüby, J., Bou Issa, C., Seck, G. S., D’herbemont, V., Hache, E., Malbec, L., Sabathier, J., Venugopal, M., Lagrange, F., Saunier, S., Straus, J. & Reigstad, G. A. The impact of methane leakage on the role of natural gas in the European energy transition. *Nat. Commun.* 14, 5756 (2023).
- Howarth, R. W. & Jacobson, M. Z. How green is blue hydrogen? *Energy Sci. Eng.* 9, 1676–1687 (2021).

Comment 2: There is a brief mention of fossil fuel, water, and land resource demand, but with no clear interpretation or comparison.

Response: We tried to emphasize a couple of points that the production of blue hydrogen requires multiple types of natural resources. The large-scale deployment of blue hydrogen will affect local natural resources and the selection of sites to deploy large-scale blue hydrogen production plants should take into account co-location and availability of multiple nature resources. To make them clearer, we revised the interpretation in the Discussion section to read: “Large-scale blue hydrogen production will consume multiple types of natural resources, such as fossil, water and land resources, and geological reservoirs and transportation infrastructure for CO₂ storage, and affect local management and planning of these natural resources. The availability and price of these natural resources vary by region or location in the country. Thus, siting blue hydrogen production plants should take into account co-location and co-availability of these natural resources.”

Comment 3: Equally, natural gas and coal are discussed alongside each other purely based on a cost perspective, even though these two energy carriers are very dissimilar and also play crucially different roles in the net-zero transition.

Response: We tried to address this comment, though we were not sure if we fully captured the key point. In addition to the cost, currently announced blue hydrogen projects in the U.S. will heavily employ gas-based reformation technologies in terms of the International Energy Agency’s hydrogen project database (2023). Thus, we added to the first paragraph of the Sensitivity Analysis section an additional justification: “Blue hydrogen projects announced in the U.S. will mainly employ gas-based reformation technologies with CCS³¹.”

Comment 4: Next, I would expect to see the current project pipeline discussed. It would be very relevant to compare the 10MMTA number to announced blue-hydrogen projects in the US and to the share of announced projects with FID. Or could you at least estimate how many projects of what scale would need to be realised to reach that number?

Response: The number of projects is highly uncertain, which highly depends on the uncertain or variable capacity of specific production projects at a plant level. We think the discussion needs to focus on the total capacity instead of the number of specific projects. To address this comment and other similar comments, we added to the Discussion section a new paragraph:

“The global production capacity of low-carbon hydrogen will reach 12.3 MMTA by 2030 based on the announced, planning and committed projects³⁰. The low-carbon hydrogen capacity

in North America will reach 6.8 MMTA by 2030³⁰. However, only 1.8 MMTA³⁰ and 1.5 MMTA^{1,2} of the announced projects in North America and the U.S. have reached final investment decision (FID), mainly because many announced projects have not yet secured financing and nailed down contracted offtake^{1,2}. The hesitancy to long-term, scaled contracts is influenced by numerous factors, such as lack of price certainty, unavailability and reliability of large-scale hydrogen supply, near-term policy implementation uncertainty, and long-term political uncertainty^{1,2}.”

Comment 5a: Moreover, I would expect some mentioning of other colours of hydrogen, especially green hydrogen. Specifically, it'd be interesting to compare current cost estimates and projections of future cost reductions of green hydrogen (there are loads of studies on this) to those estimated in the presented study. Moreover, a short comparison of emission intensities would be suitable here.

Response: As in response to a similar comment from Reviewer 1 on the comparisons of current cost and emissions between blue and green hydrogen, we added to the end of Section Current Blue Hydrogen Production a new paragraph: “Currently, hydrogen is mainly produced by SMR without CCS in the U.S., which is often called grey hydrogen. Compared to it, the blue hydrogen production by SMR with CCS can decrease the stack CO₂ emission intensity by 96% but increase the LCOH by 55%⁷. The resulting CO₂ avoidance cost by blue hydrogen is \$65 per metric ton of CO₂. In contrast, the green hydrogen production by polymer electrolyte membrane electrolyzers almost has no stack CO₂ emissions but a high LCOH value ranging from \$3.0–7.5/kg H₂²⁷. The resulting CO₂ avoidance cost by green hydrogen relative to grey hydrogen varies from \$212–689 per metric ton of CO₂, which is much higher than that by blue hydrogen. Obviously, there are tradeoffs in CO₂ avoidance cost and emission savings between the blue and green production pathways. The details of emission and cost data and CO₂ avoidance cost estimation are available in Supplementary Note 2.”

We added to the Discussion section the future cost comparison between blue and green hydrogen: “The capex learning rates of green hydrogen production are 9% and 13% for alkaline electrolysis and polymer electrolyte membrane electrolysis, respectively^{35,36}, which are similar to those for SMR and PSA. Thus, the overall levelized cost of green hydrogen will likely be larger than that of blue hydrogen by 2030³⁷.”

The cited references are:

- Hydrogen Council. Path to hydrogen competitiveness a cost perspective. https://hydrogencouncil.com/wp-content/uploads/2020/01/Path-to-Hydrogen-Competitiveness_Full-Study-1.pdf (2020).
- IRENA. Green hydrogen cost reduction: scaling up electrolyzers to meet the 1.5 °C Climate Goal. ISBN 978-92-9260-295-6. (International Renewable Energy Agency, 2020).
- IRENA. Hydrogen: a renewable energy perspective. ISBN: 978-92-9260-151-5. (International Renewable Energy Agency, 2019).

We added a new supplementary section:

Supplementary Note 2: Cost of CO₂ Avoided by Blue and Green Hydrogen

The cost of CO₂ avoided by blue or green hydrogen relative to grey hydrogen can be estimated in terms of the production plant’s CO₂ emission intensity and levelized cost of hydrogen:

$$CCA = \frac{LCOH_{\text{blue/green}} - LCOH_{\text{grey}}}{EI_{\text{grey}} - EI_{\text{blue/green}}} \quad (S1)$$

Where CCA is the cost of CO₂ avoided (\$/kg CO₂); LCOH is the levelized cost of hydrogen (\$/kg H₂); EI is the stack CO₂ emission intensity of a production plant (kg CO₂/kg H₂); the subscript of “blue” represents the gas-based blue hydrogen production, the subscript of “green” represents the green hydrogen production, and the subscript of “grey” represents the gas-based grey hydrogen production without carbon capture.

Supplementary Table 5 summarizes the stack CO₂ emission intensity and LCOH of grey, blue, and green hydrogen production plants, which are collected from the literature^{1,4}. The resulting costs of CO₂ avoided by blue and green hydrogen relative to grey hydrogen are also provided in Supplementary Table 5.

Supplementary Table 5. Hydrogen Production Emission, Production Cost, and Carbon Avoidance Cost

Production Technology	Stack Emission Intensity (kg CO ₂ /kg H ₂)	Levelized Cost of Hydrogen (2018\$/kg H ₂)	Cost of CO ₂ Avoided (\$/metric ton CO ₂)
Steam Methane Reforming (SMR) (Grey)	9.35	1.06	Reference
SMR with CCS (Blue)	0.38	1.64	65
Distributed Polymer Electrolyte Membrane Electrolyzers (Green)	0 ^a	5.17 (3.0–7.5) ^b	440 (212-689)

^aZero carbon emissions are assumed for the electrolysis powered by solar photovoltaics as it produces almost no direct emissions during operation⁵.

^bThe cost was estimated for a production capacity of 1,500 kg H₂/day with an effective electricity price of 7.55 ¢/kWh⁴. The cost was further adjusted to 2018 dollars using the annual average consumer price index for all urban consumers⁶;

Comment 5b: Again, discussing project announcements and hence likely trends of installed capacity until 2030 would be relevant to compare.

Response: To address this comment and other similar comments, we added to the Discussion section a new paragraph:

“The global production capacity of low-carbon hydrogen will reach 12.3 MMTA by 2030 based on the announced, planning and committed projects³⁰. The low-carbon hydrogen capacity in North America will reach 6.8 MMTA by 2030³⁰. However, only 1.8 MMTA³⁰ and 1.5 MMTA^{1,2} of the announced projects in North America and the U.S. have reached final investment decision (FID), mainly because many announced projects have not yet secured financing and nailed down contracted offtake^{1,2}. The hesitancy to long-term, scaled contracts is influenced by numerous factors, such as lack of price certainty, unavailability and reliability of large-scale hydrogen supply, near-term policy implementation uncertainty, and long-term political uncertainty^{1,2}.”

Regarding the possible trends of installed capacity through 2030, please review the newly-added Fig. 2a and relevant discussion.

Comment 6: A big obstacle for hydrogen and large-scale industrial projects is still the lack of technological certainty. Many announced projects (see for instance the IEA hydrogen project database) still lack an FID. Perhaps the authors could discuss this in the US context and if/how their results could lay out a clearer plan for not only cost competitiveness but also for security of investment for blue hydrogen projects.

Response: To address this comment and other similar comments, we added to the Discussion section a new paragraph:

“The global production capacity of low-carbon hydrogen will reach 12.3 MMTA by 2030 based on the announced, planning and committed projects³⁰. The low-carbon hydrogen capacity in North America will reach 6.8 MMTA by 2030³⁰. However, only 1.8 MMTA³⁰ and 1.5 MMTA^{1,2} of the announced projects in North America and the U.S. have reached final investment decision (FID), mainly because many announced projects have not yet secured financing and nailed down contracted offtake^{1,2}. The hesitancy to long-term, scaled contracts is influenced by numerous factors, such as lack of price certainty, unavailability and reliability of large-scale hydrogen supply, near-term policy implementation uncertainty, and long-term political uncertainty^{1,2}. For blue hydrogen projects, enhancements in tax credits for carbon sequestration can improve the economics of hydrogen production. For example, an extension of the 45Q tax credit period from current 12 years to 18 years would significantly reduce the cumulative installed capacity required for gas-based blue hydrogen projects to reach the Hydrogen Energy Earthshot, as demonstrated in Supplementary Fig. 3. Extending the period of the 45Q tax credit for blue hydrogen projects can be considered an option to secure financing and promote long-term offtake.”

Supplementary Fig. 3. Effect of 45Q tax credit period on future cost of gas-based blue hydrogen production.

Comment 7: Finally, I would expect to see more depth added to the conclusions section and for a more explicit outline for competing strategies and policy options (e.g. further

subsidies/financial incentives; focussing on maturing one technology only; incentives to build several small vs few big plants; securing investments; speeding up planning procedures; etc).

Response: We expanded the relevant discussion by adding a new paragraph:

“Competing strategies and supportive policy and regulatory actions should be made rapidly on both the hydrogen demand and supply sides at both federal and state levels in alignment with the innovation expansion. A variety of high-level strategies are needed on the demand side to promote the widespread use of low-carbon hydrogen in industrial, transportation and power sectors and then establish large-sale markets for low-carbon hydrogen. To jumpstart a hydrogen economy, a cluster approach can be employed on the supply side to establish regional production-transportation-demand networks by co-locating feedstock supply, hydrogen production, and carbon sequestration with multiple end-users and by utilizing existing infrastructure, such as pipeline infrastructure for natural gas, CO₂, and H₂ transportation and geological reservoirs for CO₂ storage. To scale the regional hydrogen economy, secured investments in hydrogen production and supporting infrastructure are required with funding from both public and private sectors, plus subsidies and tax incentives. In addition, deploying large hydrogen production plants instead of small ones can improve engineering economics at a plant level. Given the important role of CCS in producing competitive blue hydrogen, continued support for large-scale demonstration projects should be boosted in the near term to reduce the CCS cost and its uncertainty. Investments in blue hydrogen should be prioritized to lock down sufficient financial resources for the most competitive technologies in the near term. Economic and policy incentives can be tailored with emphasis on gas-based blue hydrogen to catalyze its widespread deployment and technological evolution because of the pronounced cost advantage relative to coal-based blue hydrogen. Extending the 45Q tax credit from the current 12-year period to a longer period for gas-based blue hydrogen projects would remarkably lower the time-related cumulative installed capacity necessary to reach the Hydrogen Energy Earthshot.”

2) Key assumptions and validity of results

Comment 8: The key results of the manuscript crucially depend on the learning rates assumed for the cost reductions of the individual components. The authors assume these values from two primary sources (Rubin et al from 2007, and Schoots et al from 2008). These cited studies are now more than 15 years old and the data they refer to ranges from the 60s to the early 2000s. I wonder how reliable and valid the derived learning curves are for today's SMR and CO₂ capture technologies. I understand that these numbers are very tricky to derive with certainty, and one often has to resort to regressions of historical data. Yet, given that this is such a key assumption of this manuscript, I would expect a much more detailed analysis and discussion of these assumed parameters. My first impression is that a learning rate for SMR is somewhat high, given that this is a rather established technology.

Response: This study mainly cited the two pioneering learning curve studies by Rubin et al. (2007) and Schoots et al. (2008). Although a large number of technological learning studies have been done in the past years, almost no breakthroughs have been made on learning rates beyond the two pioneering studies in the past years. The learning rates from the two pioneering studies have been frequently adopted in the recent studies. In other words, there are no more rigorous estimates of learning rates in the literature than the two pioneering studies cited in our manuscript. To make it clearer, we added to the relevant paragraph in the introduction section on Page 5 of the original version new sentences with recent references:

“Although a large number of learning curve studies have been done in the past years, almost no breakthroughs have been made in estimating the learning rates for CCS, gasification, and SMR beyond the two pioneering studies by Rubin et al. (2007)¹⁴ and Schoots et al. (2008)¹³. Recent studies frequently adopted the learning rates from the two pioneering studies for a variety of applications^{17,18,19,20,21,22,23,24,25}.”

As given in Table 2, the base learning rate in operating and maintenance costs are zero for SMR based on the study by Schoots et al (2008). However, the capital cost still can benefit from learning by manufacturing and deploying massive reactors.

We further expanded the relevant discussions in the method section, which are highlighted in blue color: “To construct a learning curve for a technology with respect to its either total capital cost or total O&M cost, three types of model parameters have to be specified, including the initial cost, initial installed capacity, and learning rate. Capital and O&M learning rates can be estimated using empirical data for mature technologies or an analogous approach for advanced technologies. For example, the learning rates for SMR were derived from its historical installed capacity and cost data¹³, whereas the learning rates for post-combustion carbon capture were estimated by referring to those of post-combustion flue-gas desulfurization as they are technically analogous^{14,42}. The data collected for these parameters are discussed later.

A hydrogen production plant involves multiple technologies or subsystems, which have different values regarding the three parameters defining a learning curve. At a blue hydrogen production, individual subsystems lie at different levels of technological maturity. Learning rates and initial installed capacity count on maturity level and then vary by subsystem. For example, at a gas-based blue production plant, SMR and PSA are mature subsystems, whereas carbon capture has not been deployed widely, though it is commercially available. As a result, the O&M learning rates are zero for SMR and PSA but 22% for carbon capture. Thus, a component-based learning curve model is applied to estimate the total cost of hydrogen production at a certain level of cumulative installed capacity as the sum of individual subsystem costs.”

To clarify how an overall plant’s learning curve is constructed, we further expanded the discussion in the result section by adding to the first paragraph of Section Future Costs of Blue Hydrogen Production without and with Tax Credits new sentences: “A blue hydrogen production plant consists of numerous subsystems. However, the maturity status of individual subsystems and their initial installed capacity are different. As a result, learning rates and initial installed capacity vary by subsystem. Thus, a component-based learning curve model is employed to construct a plant-level learning curve based on individual subsystems’ learning rates and initial installed capacity.”

Comment 9: Also note that modern plant designs will have to ensure minimal CH₄ leakage (less than 0.1%), which will necessitate extra efforts and could thereby easily drive up SMR cost and hence offset other learning effects.

Response: A study by Alhamdani et al. (2017) reports that fugitive GHG emissions from an SMR production plant are minor and about 0.05% of the stack emissions, which indicates that on-site CH₄ leakage is not a serious concern. To address this point, we added to the “Current Blue Hydrogen Production” section on Page 6 the relevant results: “In addition to stack CO₂ emissions, there may be fugitive GHG emissions from various sources at an SMR production plant, mainly from the piping equipment and fittings²⁶. However, fugitive GHG emissions are

about 0.05% of the stack GHG emissions²⁶, which indicates that plant methane leakage is not a serious issue.”

The newly cited reference is:

- Alhamdani, Y. A., Hassim, M. H., Ng, R. T. & Hurme, M. The estimation of fugitive gas emissions from hydrogen production by natural gas steam reforming. *Int. J. Hydrogen Energy*, **42**, 9342-9351 (2017).

Comment 10: Moreover, it would be interesting if the manuscript was able to differentiate more clearly, how learning-by-doing could drive down the cost of those specific components and if/how R&D could support this effort.

Response: As the reviewer suggested, we further looked into the cost reduction from learning-by-doing and the breakdown of the cost reduction by component when the cumulative installed capacity reaches 10 MMTA. To present the relevant result and implication, we added to Page 12 of the original manuscript a new paragraph:

“Learning-by-doing will reduce the cost of hydrogen production for coal- and gas-based blue hydrogen. Fig. 1e shows the cost reduction by subsystem and by the 45Q tax credit when the cumulative installed capacity of blue hydrogen reaches 10 MMTA. For blue hydrogen produced from both coal and gas resources, the overall cost reduction will be driven largely by the carbon-sequestration tax credit and the improvement in carbon capture. In contrast, other subsystems, such as SMR and PSA, will make limited contributions because they are mature technologies and have no or limited reductions from additional 10 MMTA deployment in their future costs. These results indicate the importance of continued support from both public and private sectors for CCS-related research, development and demonstration programs at federal and state levels.”

Fig. 1 Initial and future costs of blue hydrogen production without and with tax incentives. (e) Future cost reductions for coal-based and gas-based H₂ production with tax incentives.

Comment 11: The article focusses on SMR as the main technology for blue hydrogen. However, at least to my understanding, the most suitable technology for blue hydrogen with high capture rates would be ATR. Can the authors comment on why they decided to choose SMR instead of ATR?

Response: The NETL study (2022) reports that for blue hydrogen production, ATR with CCS has stack emissions of 0.51 kg CO₂/kg H₂, life cycle emissions of 5.72 kg CO₂-eq/kg H₂, and the levelized production cost of \$1.59/kg H₂. In contrast, SMR with CCS has stack emissions of 0.38 kg CO₂/kg H₂, life cycle emissions of 4.57 kg CO₂-eq/kg H₂, and the levelized production cost of \$1.64/kg H₂. To clarify why our study focuses on SMR, we added to the first section of the results section on Page 6 of the original version: “The majority of hydrogen produced in the U.S. is made via SMR. In addition, the cost of blue hydrogen produced by SMR with CCS is similar to that by autothermal reforming with CCS but the on-site and life-cycle emissions from the SMR process are less⁷. This study therefore focuses on SMR with CCS for blue hydrogen production.”

3) Presentation of figures and text

Comment 12: As mentioned above, the whole article would greatly benefit from more discussion of context and connecting to the wider scientific debate. Specifically, the introduction and conclusions sections require more contextualisation.

Response: As suggested, we have made significant improvements to expand the discussion of context in a broader context, especially in the introduction and discussion sections. In particular, please review our responses to the high-level comment above and Comments 1, 2, 4, 5a and 5b, 6, 7, 17, and 18.

Comment 13: I don't know how this article is doing on the word-count limit, but I suspect it's rather tight. I would suggest moving a lot more of the technical discussions from the main text (e.g. "Current blue hydrogen production") into the Methods section and really only discuss the results and the key assumptions (learning rates). This would give more space for discussing context rather than stating methodological details or points that the reader can read off from the figures.

Response: We cannot fully agree with this comment. We justified the adoption of learning rates and added an array of new analyses, results and discussions to address the valuable comments from the reviewer. However, we cannot move the “Current blue hydrogen production” section into the Methods as this section presents major assumptions and current production technologies’ performance and cost, such CO₂ emissions and sequestration and levelized cost of hydrogen production, including the cost distribution by subsystem, which set up initial points to formulate learning curves. Without these results, we cannot create learning curves and outline the rationale for a series of parametric analyses. In other words, we have to first quantify and characterize the current hydrogen production and then present the learning curves for hydrogen production in the future, including its variability by key factors.

Comment 14: The figures convey little information and don't allow the reader to easily grasp the main results of the manuscript at one glance. I also can't help but noticing that the figures were created in Excel. Often (not always) I find that this does not allow to create advanced plots that can densely present a lot of information in a yet elegant and comprehensible way. For a

publication in Nature Communications, I would expect the authors to work hard to create an iconic figure capturing the key results and conveying these along with an accompanying message/story in a very comprehensible manner. E.g., I would suggest to have one main figure consisting of 1) the pie chart (which is good!), 2) a stacked-bar waterfall diagram for each technology demonstrating the different components of cost savings at 10 MMTA installed capacity and with the tax credits, and 3) the learning curves of the main assumptions without the point markers (or do they serve a purpose?) combined with shaded uncertainty corridors indicating the results of the sensitivity analysis (only gas price OR learning rate OR perhaps both combined).

Response: We partially agree with this comment. The figures were created in Excel but delivered the main information or results of this manuscript. We appreciate the reviewer for pushing us to present the figures in a higher quality. To improve the quality, we used data analysis software to re-draw all the figures in the manuscript.

As suggested, we kept the pie chart and added a stacked-bar waterfall diagram as Fig. 1e. Furthermore, we added shared uncertainty ranges to the learning curves, combined the relevant figures accordingly, and added a heat map on learning rates. Point markers were added accordingly. Below are the re-drawn ones of the three figures presented in the original manuscript. In addition, we revised the relevant statements in line with the representation of all the updated figures.

Fig. 1 Initial and future costs of blue hydrogen production without and with tax incentives. (a) Distribution of initial levelized cost for gas-based H₂ production; (b) Distribution of initial levelized cost for coal-based H₂ production; (c) Learning curves for coal-based and gas-based H₂ production capital and O&M costs without tax incentives; (d) Learning curves for overall levelized cost of coal-based and gas-based H₂ production without and with tax incentives; (e) Future cost reductions for coal-based and gas-based H₂ production with tax incentives.

Fig. 3 Effect of natural gas price on future levelized cost of gas-based blue hydrogen production.

Fig. 5 Sensitivity of LCOH by SMR with CCS to learning rates (LR). (a) LCOH under two boundary scenarios of learning rates; (b) LCOH under the range of 100%–150% time base learning rates, except for O&M cost learning rates, which are equal to 5%–10% for SMR, PSA, and CO₂ compression. Note to Fig. 5b: P₁ means a percentage relative to the base learning rate, whereas P₂ means the learning rate on an absolute basis.

Comment 15: In summary, the authors investigate a very relevant and topical research question, but the way in which their manuscript is currently written, it doesn't match the portfolio of Nature Communications and would be more suitable for a more field-specific technical journal. Irrespective of whether the authors choose to resubmit to NComms, I would encourage them to substantially revise their paper by: a) adding more context, b) pointing out key take-aways more clearly, c) improving the graphical presentation of their results, and d) more thoroughly discussing/substantiating the key assumptions of their work (mainly the learning rate).

Response: We have made significant revisions to address all the comments. In particular, (a) we added new analyses, including comparative analysis between blue and green hydrogen (Response to Comment 5a), announced and FID projects (Response to Comment 4, Response to

Comment 5a, and Response to Comment 6), diffusion-of-innovation analysis (Response to Comment 17), and inflation analysis (Response to Comment 21); (b) we improved the clarifications on some key take-home message, such as Response to Comment 10 and Response to Comment 18; (c) we have re-drawn all the figures in the initial submission to improve the quality and added two new figures to the revised manuscript. Please review Response to Comment 14, Response to Comment 17, and Response to Comment 18; (d) we expanded the statements, justifications, and clarifications to thoroughly present the assumptions made in this study. In particular, please review our responses to Comments 8, 9, 11, 16, and 19.

Some further points/comments/questions/remarks:

Comment 16: What is your assumption for the interest rate/WACC? (I can't remember reading it and couldn't find it with Ctrl+F.) I wonder if the assumed WACC could also be a basis for another sensitivity analysis.

Response: It turns out that we should substantially improve the presentation of the costing method with more details. We revised the methodological description of cost metric in line with the NETL's costing method (2022). Fixed charge rate is a key parameter used to estimate the overall production cost, which is affected by the after-tax WACC. Both after-tax WACC and FCR are affected by inflation. Thus, an additional sensitivity analysis was performed to evaluate the economic effect of inflation on the LCOH in nominal dollars. Additional methodological details about how to estimate the LCOH in nominal dollars are available in the supplementary information.

“Cost metric for technology evaluation

The component-based learning curve model is applied to project the future total capital cost and total O&M cost of individual subsystems and an overall plant as a function of cumulative installed capacity and then estimate the overall cost of hydrogen production for a given cumulative installed capacity. The cost metric considered for technology evaluation is the levelized cost of hydrogen and is estimated in real dollars^{7,34}.

$$LCOH_R = LCC_R + LOM_R + LFP_R \quad (2)$$

$$LCC_R = \frac{TASC_R \cdot FCR_R}{(CF \cdot AH) \cdot KG_{H_2}} \quad (3)$$

$$LOM_R = \frac{OM_R}{(CF \cdot AH) \cdot KG_{H_2}} \quad (4)$$

$$LFP_R = \frac{FC_R \cdot FR}{KG_{H_2}} \quad (5)$$

Where the subscript “R” means the real dollars; $LCOH_R$ is the levelized cost of hydrogen of a blue hydrogen production plant (\$/kg H₂); LCC_R is the levelized capital cost (\$/kg H₂); LOM_R is the non-fuel levelized operating and maintenance cost (\$/kg H₂); LFP_R is the levelized fuel price (\$/kg H₂); $TASC_R$ is the total as-spent capital of a blue hydrogen production plant (\$); FCR_R is the fixed charge rate (fraction/year); CF is the plant capacity factor (%); AH is the total annual hours (8760 hours); KG_{H_2} is the hourly production rate (kg H₂/hour); OM_R is the total non-fuel operating and maintenance (O&M) cost (\$/year), including both the fixed and non-fuel variable O&M costs; FC_R is the natural gas cost (\$/GJ) or coal cost (\$/metric ton); FR is the hourly natural gas flow rate (GJ/hour) or coal flow rate (metric ton/hour). When estimating the LCOH

in real dollars, the FCR has to be determined using Equations (6) to (9), which varies with the project book lifetime and a panel of financial variables³⁴:

$$FCR_R = \frac{CRF_R^{\text{nonfuel}} - ETR \cdot PV_{\text{plant}}}{1 - ETR} \quad (6)$$

$$CRF_R^{\text{nonfuel}} = \frac{ATWACC_R \cdot (1 + ATWACC_R)^{BL}}{(1 + ATWACC_R)^{BL} - 1} \quad (7)$$

$$PV_{\text{plant}} = CRF_R^{\text{nonfuel}} \cdot \sum_{n=1}^m \frac{d_n}{(1 + ATWACC_R)^n} \quad (8)$$

$$ATWACC_R = PC_{\text{equity}} \cdot ROE_R + PC_{\text{debt}} \cdot kd_R \cdot (1 - ETR) \quad (9)$$

Where the superscript “nonfuel” represents the non-fuel component; CRF_R is the capital recovery factor (fraction/year); ETR is the effective tax rate (%); PV_{plant} is the present value of tax depreciation expense of a blue hydrogen project (fraction/year); $ATWACC_R$ is the after-tax weighted average cost of capital (%); BL is the project book lifetime (year); d_n is the tax depreciation fraction in a year (n) (fraction); m is the number of years of depreciation (year); PC_{equity} is the percent of equity (%); PC_{debt} is the percent of debt (%); ROE_R is the real rate of return on equity (%); kd_R is the real rate of cost of debt (%). The financial parameters and their data sources are detailed in Supplementary Table 17 and the resulting FCR is available in Supplementary Table 18.

Inflation affects discount rate, $ATWACC$, FCR , and other factors. To evaluate the effect of inflation on the cost of blue hydrogen production, the LCOH is estimated in nominal dollars. The detailed estimation of the LCOH in nominal dollars is reported in Supplementary Note 8.”

To explain how to estimate the LCOH in nominal dollars, we added a new supplementary session:

Supplementary Note 8: Hydrogen Production Cost Estimation in Nominal Dollars

The levelized cost of hydrogen and is estimated in nominal dollars¹:

$$LCOH_N = LCC_N + LOM_N + LFP_N \quad (S12)$$

$$LCOH_N^{\text{TC}} = LCC_N + LOM_N + LFP_N - LTC_N^{45Q} \quad (S13)$$

Where the subscript “N” means the nominal dollars; $LCOH_N$ is the levelized cost of hydrogen (\$/kg H₂); LCC_N is the levelized capital cost (\$/kg H₂); LOM_N is the levelized operating and maintenance cost (\$/kg H₂); LFP_N is the levelized fuel price (\$/kg H₂); $LCOH_N^{\text{TC}}$ is the levelized cost of hydrogen with a tax credit (\$/kg H₂); LTC_N^{45Q} is the levelized tax credit over the project book lifetime of a blue hydrogen project (\$/kg H₂).

The levelized capital cost in nominal dollars is estimated using Supplementary Equation (14), while the fixed charge rate is estimated using Supplementary Equations (14) to (21)^{2,3}:

$$LCC_N = \frac{TASC_R \cdot FCR_N}{(CF \cdot AH) \cdot KG_{H_2}} \quad (S14)$$

$$FCR_N = \frac{CRF_N^{\text{nonfuel}} - ETR \cdot PV_{\text{plant}}}{1 - ETR} \quad (S15)$$

$$CRF_N^{\text{nonfuel}} = \frac{ATWACC_N \cdot (1 + ATWACC_N)^{BL}}{(1 + ATWACC_N)^{BL} - 1} \quad (S16)$$

$$PV_{\text{plant}} = CRF_N^{\text{nonfuel}} \cdot \sum_{n=1}^m \frac{d_n}{(1 + ATWACC_N)^n} \quad (\text{S17})$$

$$ATWACC_N = PC_{\text{equity}} \cdot ROE_N + PC_{\text{debt}} \cdot kd_N \cdot (1 - ETR) \quad (\text{S18})$$

$$ROE_N = (1 + ROE_R) \cdot (1 + N) - 1 \quad (\text{S19})$$

$$kd_N = (1 + kd_R) \cdot (1 + N) - 1 \quad (\text{S20})$$

$$N = R + I \quad (\text{S21})$$

Where the superscript “nonfuel” represents the non-fuel component; $TASC_R$ is the total as-spent capital of a blue hydrogen production plant (\$); FCR_N is the fixed charge rate (fraction/year); CF is the plant capacity factor (%); AH is the total annual hours (8760 hours); KG_{H_2} is the hourly production rate (kg H_2 /hour); CRF_N is the capital recovery factor (fraction/year); ETR is the effective tax rate (%); PV_{plant} is the present value of tax depreciation expense (fraction/year); $ATWACC_N$ is the after-tax weighted average cost of capital (%); BL is the project book lifetime (year); d_n is the tax depreciation fraction in year n (fraction); m is the number of years of depreciation (year); PC_{equity} is the percent of equity (%); ROE_N is the nominal rate of return on equity (%); PC_{debt} is the percent of debt (%); kd_N is the nominal rate of cost of debt; ROE_R is the real rate of return on equity (%); kd_R is the real rate of cost of debt (%); N is the nominal escalation rate (%); R is the real escalation rate (%); and I is the inflation rate (%).

The levelized operating and maintenance cost in nominal dollars is estimated using Supplementary Equation (22), while the levelization factor is estimated using Supplementary Equation (23)³:

$$LOM_N = \frac{OM_R \cdot LF_N}{(CF \cdot AH) \cdot KG_{H_2}} \quad (\text{S22})$$

$$LF = CRF_N^{\text{nonfuel}} \cdot \frac{1 - \left[\frac{1 + N}{1 + ATWACC_N} \right]^{BL}}{ATWACC_N - N} \quad (\text{S23})$$

Where OM_R is the non-fuel operating and maintenance (O&M) cost of a blue hydrogen production plant (\$/year); and LF_N is the levelization factor for non-fuel O&M cost (unitless).

The levelized fuel price in nominal dollars is estimated using Supplementary Equation (24), where the fuel’s capital recovery factor is estimated using Supplementary Equation (26) and the discount rate in nominal dollars is estimated using Supplementary Equation (27)¹⁶:

$$LFP_N = \frac{FC_N \cdot FR}{KG_{H_2}} \quad (\text{S25})$$

$$FC_N = PV_{\text{fuel}} \cdot CRF_N^{\text{fuel}} \quad (\text{S25})$$

$$CRF_N^{\text{fuel}} = \frac{DR_N \cdot (1 + DR_N)^{BL}}{(1 + DR_N)^{BL} - 1} \quad (\text{S26})$$

$$DR_N = (1 + DR_R) \cdot (1 + N) - 1 \quad (\text{S27})$$

$$PV_{\text{fuel}} = \sum_{t=1}^{BL} \frac{P_t}{(1 + DR_N)^t} \quad (\text{S28})$$

$$P_t = P_{t-1} \cdot (1 + N) \quad (\text{S29})$$

Where the superscript “fuel” represent the fuel component; FC_N is the natural gas cost (\$/GJ); FR is the hourly natural gas flow rate (GJ/hour); PV_{fuel} is the present value of gas cost (\$/GJ);

CRF_N^{fuel} is the fuel’s capital recovery factor (fraction/year); DR_N is the nominal discount rate for fuel cost (%); DR_R is the real discount rate for fuel cost (%); N is the nominal escalation rate (%); P_t is the fuel cost in year t (\$/GJ), while P_1 is the real fuel cost in the first year of a project.

In this study, the financial structure and assumptions are in alignment with those of the studies by the National Energy Technology Laboratory (NETL)^{1,2,3,16,17}. Supplementary Table 17 lists the financial parameters and summarizes their data and the sources of data. Supplementary Table 18 summarizes the estimates of FCR, levelization factor, and discount rate as a function of nominal escalation rate.

Supplementary Table 17. Financial Parameters and Assumptions

Parameter	Symbol	Unit	Value	Source(s)
Effective Tax Rate	ETR	%	25.74%	NETL (2021) ³
Inflation Rate	I	%	0–3%	Assumption
Real Escalation Rate	R	%	0%	NETL (2011) ²
Percent of Equity	PC_{equity}	%	62%	NETL (2022) ¹
Percent of Debt	PC_{debt}	%	38%	NETL (2022) ¹
Real Rate of Return on Equity	ROE_R	%	3.10%	NETL (2022) ¹
Real Rate of Cost of Debt	kd_R	%	5.15%	NETL (2022) ¹
Real Discount Rate	DR_R	%	4.72%	NETL (2019) ¹⁶
Project Book Lifetime	BL	year	30	NETL (2022) ¹
Fuel Cost	P_1	2018\$/GJ	4.2	NETL (2022) ¹
Tax Depreciation Fraction	d_1	%	3.75%	NETL (2021) ³ ; IRS (2016) ¹⁷
	d_2		7.22%	
	d_3		6.68%	
	d_4		6.18%	
	d_5		5.71%	
	d_6		5.29%	
	d_7		4.89%	
	d_8		4.52%	
	d_9		4.46%	
	d_{10}		4.46%	
	d_{11}		4.46%	
	d_{12}		4.46%	
	d_{13}		4.46%	
	d_{14}		4.46%	
	d_{15}		4.46%	
	d_{16}		4.46%	
	d_{17}		4.46%	
	d_{18}		4.46%	
	d_{19}		4.46%	
	d_{20}		4.46%	
	d_{21}		2.23%	

Supplementary Table 18. Financial Parameter Estimates

Parameter	Symbol	Unit	Nominal Escalation Rate			
			0%	1%	2%	3%
Fixed Charge Rate	FCR	fraction/year	0.059	0.067	0.076	0.085
Levelization Factor	LF	fraction	1.000	1.124	1.256	1.393
Discount Rate	DR	%	4.73%	5.78%	6.82%	7.87%

Comment 17: My understanding is you are currently deducting the tax credit from the levelised cost, i.e. thereby spreading the credit across the assumed book lifetime. Perhaps it would make sense to rather discuss in more detail what will happen when the tax credit ceases? I.e. the later that projects start, the less of the credit period they will be able to obtain. Your current main figures have product capacity on the lower axis. If you assume a certain deployment rate, you could plot these figures as a function of time. The resulting figure would allow to see the trade-off between waiting till the production cost drops lower via learning vs. less tax credit (I assume the tax credit would still be the much bigger effect, hence indicating it'd be more profitable to construct now rather than later).

Response: Tax credit is available for a certain period for blue hydrogen instead of the entire book lifetime. As discussed in the manuscript, the period is 10 years for the 45 V tax credit and 12 years for the 45Q tax credit. In addition, the regulations require that to claim either the 45Q tax credit or the 45V tax credit, facilities must be placed in service before January 1, 2033. For such facilities (in service before 1/1/2033), the period of credit availability is in common, regardless of their start-of-service time. The suggested analysis looks not suitable for the tax scheme that our study examined. To make the tax scheme clearer, we added to the “Blue hydrogen production with tax credits” section new sentences: “To claim either the 45Q tax credit or the 45V tax credit, facilities must be placed in service before January 1st, 2033²⁹. The credit is available for such qualified facilities for a period. The period of credit availability is in common to eligible facilities, regardless of their start-of-service time.”

Actually, this comment raises a valuable concept regarding how the technology adoption diffuses over time. Thus, we developed a diffusion-of-innovation model and applied it to estimate the installed capacity and the corresponding cost of hydrogen production as a function time, which can collectively determine how much the cumulative installed capacity would be deployed at what cost of hydrogen production in any particular year in the future. We added the newly-developed diffusion-of-innovation model to the Methods section:

“Diffusion-of-Innovation Model

The diffusion of innovation describes how a new technology would spread over time. An S-shaped curve is often used to measure the diffusion over time, in which the adoption rate increases during the early stage, reaches a maximum level at the point of inflection, and decreases until the diffusion curve saturates⁴⁵. To estimate the annual installed capacity of low-carbon hydrogen over time, the S-shaped diffusion function is employed^{46,47}:

$$cc_t = \frac{cc_{sat}}{1 + \frac{cc_{sat} - cc_0}{cc_0} \cdot e^{-r \cdot t}} \quad (1)$$

Where cc_t is the annual installed capacity of low-carbon hydrogen in a particular year t (million metric tons per annum, MMTA); cc_{sat} is the saturation level of annual installed capacity (MMTA); cc_0 is the initial annual installed capacity (MMTA) in the start year; r is the growth rate (fraction); and t is a particular year after the start period. The function coefficients are estimated by regression based on current and future low-carbon hydrogen production capacities through 2030^{12,28,30}. Additional details about the regression and diffusion function are available in Supplementary Note 7. Once the annual installed capacity in future years is determined, the cumulative annual installed capacity can be estimated as a function of time.”

We added the relevant results to the Results section:

“Time-based diffusion of blue hydrogen production

It is helpful for hydrogen energy planning to explore if certain production capacity and cost targets can be achieved by 2030. A new study reports the cumulative installed capacity of low-carbon hydrogen production over time based on globally announced, planning and committed projects through 2030³⁰. A diffusion-of-innovation model was established based on the current and future low-carbon hydrogen capacities through 2030 to explore the time-based diffusion of gas-based blue hydrogen over a long-term planning horizon through 2050.

Fig. 2a shows the cumulative installed capacity estimates for global low-carbon hydrogen production over time. The gas-based blue hydrogen capacity accounts for 49% of the total low-carbon hydrogen capacity given in Table 1 and is estimated to be 90% in 2030 in terms of the International Energy Agency’s hydrogen project databases^{28,31}. Given the changing shares over time, Fig. 2a also shows a range of cumulative installed capacity for gas-based blue hydrogen in a particular year. The cumulative installed capacity of the global gas-based blue hydrogen may range from 6 to 12 MMTA in 2030, which implies that it would be hard for the blue hydrogen production by SMR with CCS alone in the U.S. to reach 10 MMTA in 2030.

Fig. 1c shows the overall plant LCOH as a function of cumulative installed capacity for gas-based blue hydrogen, whereas Fig. 2a show the cumulative installed capacity over time. Combining them together, Fig. 2b shows the overall plant LCOH of gas-based blue hydrogen production without tax credit over time. The result shown in Fig. 2b implies that for the fuel price and learning rates given in the base case, it would also be difficult for gas-based blue hydrogen to reach the ambitious cost target of \$1/kg H₂ by 2030 in normal scenarios without aggressive incentives and game-changing technologies.

Fig. 2 Diffusion of cumulative installed capacity of low-carbon hydrogen and time-based learning curves of gas-based blue hydrogen. (a) Diffusion of cumulative installed capacity; (b) Time-based learning curves of blue hydrogen production cost.

Comment 18: Two main takeaways for me from the manuscript are: a) substantial cost reductions via "learning-by-doing" can only be achieved if technologies are sufficiently deployed, and b) coal-based blue hydrogen seems to be uncompetitive compared to gas-based. Wouldn't it therefore be sensible to recommend focussing policy support/industry efforts on gas-based blue hydrogen to reach sufficient deployment and hence cost reductions of this one technology instead of splitting up efforts between the two options?

Response: As in response to a similar comment above, we added to the discussion section this point: “Investments in blue hydrogen should be prioritized to lock down sufficient financial resources for the most competitive technologies in the near term. Economic and policy incentives can be tailored with emphasis on gas-based blue hydrogen to catalyze its widespread deployment and technological evolution because of the pronounced cost advantage relative to coal-based blue hydrogen. Extending the 45Q tax credit from the current 12-year period to a longer period for gas-based blue hydrogen projects would remarkably lower the time-related cumulative installed capacity necessary to reach the Hydrogen Energy Earthshot.”

Comment 19: Eq. (6) looks odd. Shouldn't the term proportional to the VOM be independent of the CF? I.e. there is no cost for fuel consumption during the period when the plant is not producing.

Response: If the VOM is reported on an absolute basis (\$/yr), it is dependent on the CF. If the VOM is reported in a normalized basis, it is independent of the CF. There is no cost for fuel consumption when the plant is not producing hydrogen. As in response to a comment on the costing method above, we revised the equations in line with the NETL’s costing method, in which the VOM is reported on an absolute basis and the fuel cost is reported separately as the third category.

Comment 20: Irrespective of what you choose to do with your figures, could you make sure the x-axis ticks are the same across all of them? E.g. it's hard to find the 1\$/kg marker. Perhaps you could also add a dashed horizontal line to that graph?

Response: As suggested, we made the x-axis ticks consistent with all the figures, except for a new heat map, which is not suitable for this change. We also added a dashed horizontal line to all the relevant figures to indicate the “1 \$/kg H₂” target.

Comment 21: Given you're trying to compare cost reductions against the 1\$/kg target, it'd be valuable to discuss the role of inflation. I noticed you're using \$2018 numbers from the NETL study. It's almost a bit of a silly question because I presume the 1\$/kg target is more of a political slogan than the result of an elaborate economic analysis, but I wonder how much inflation today and in coming years would make it harder to reach that specific nominal value.

Response: As the reviewer mentioned, the \$1/kg target is the U.S. DOE’s initiative goal, which has no specific information regarding cost type (real vs. nominal dollars) and inflation. This study reports cost results in real dollars. As in response to a similar comment from Reviewer 1, we conducted an additional parametric analysis and estimated the cost results in nominal dollars to demonstrate the effect of inflation rate on the future cost of hydrogen production toward the Hydrogen Energy Earthshot. The results imply that inflation would remarkably raise challenges for blue hydrogen production to reach the cost target. We added to the Sensitivity Analysis section the new analysis and relevant results:

“Inflation Rate

In general, this study estimates the cost of hydrogen production in real dollars. When the cost is estimated in nominal dollars, however, both the initial and future LCOH estimates vary with inflation rate as it affects discount rate, fixed charge rate, and levelization factor. A parametric analysis was further performed for inflation rate to quantify its effect on the evolving cost of gas-based blue hydrogen production toward the Hydrogen Energy Earthshot. Fig. 6 shows the learning curves of blue hydrogen production with inflation. Fig. 6a and 6b show that at

a given level of cumulative installed capacity, the LCOH in nominal dollars increases when the inflation rate increases from 1% to 3%. As a result, blue hydrogen production may not reach the cost target of \$1/kg H₂ for both the scenarios without and with 45Q tax credit even when the cumulative installed capacity reaches 30 MMTA. Fig. 6c further shows that with an inflation rate of 3%, the future LCOH may get close to the cost target when cheap natural gas resources are used as the feedstock to produce blue hydrogen with the cumulative installed capacity of up to 30 MMTA.

Fig. 6a and Fig. 6b also compare the learning curves of blue hydrogen production between the two scenarios without and with inflation. As shown in Fig. 6a for the scenario without 45Q tax credit, the reduction in hydrogen production cost from deploying the cumulative installed capacity of 10 MMTA can be offset by an inflation rate of 1%. There is a similar result for the scenario with 45Q tax credit, as shown in Fig. 6b. All these results imply that inflation would remarkably raise challenges for blue hydrogen production to reach the Hydrogen Energy Earthshot in the near future.”

Fig. 6 Effect of inflation rate on future cost of gas-based blue hydrogen production without and with 45Q tax credit. (a) Levelized cost of hydrogen production with gas price of \$4.2/GJ and without 45Q tax credit; (b) Levelized cost of hydrogen production with gas price of \$4.2/GJ and 45Q tax credit; (c) Levelized cost of hydrogen production with 3% inflation rate and 45Q tax credit.

Comment 22: Can you say more explicitly what is meant by “constant dollars”?

Response: It means real dollars. The two terminologies are often used in practice. To avoid unnecessary confusing, we changed “constant dollars” to “real dollars” unless otherwise noted in the manuscript.

Comment 23: Typo: summery

Response: We fixed the typo.

Reviewer #3 (Remarks to the Author):

Comment: This paper looks at the projected cost of blue hydrogen as a function of different learning rates, taking into account the IRA tax credits for hydrogen. It relies on a recent comprehensive report by NETL, which provides process level comparisons of state-of-the-art, fossil-based hydrogen production technologies. The authors use their analysis to assess the likelihood of reaching the US Hydrogen Earthshot goal of hydrogen at 1 \$/kg by 2030. Given the importance of hydrogen in future global energy systems, and the uncertainties around technology costs, understanding the cost evolution of hydrogen technologies is a useful and timely addition to the literature. The paper focusses heavily on learning rates and gas prices. However, the novelty of the analysis and depth of the discussion could be improved.

Response: We have an agreement with the reviewer on the timely importance of this study. We appreciate the reviewer for valuable comments. We made substantial revisions to address the comments.

Comment: Firstly, the assumptions for learning rate seem simplistic and lack a discussion/justification of how they apply to an industry that is looking to evolve a mature technology: SMR is a commercially mature process, but coupling it with CCS is not. The discussion of learning rate discovery is limited to two sentences in the methods section which simply states that ‘learning rates are mainly collected from two highly-cited learning curve articles’. In particular, the authors mention the option of retro fitting existing SMR plants briefly in the discussion, but do not include this important pathway in their analysis. The manuscript needs a more in-depth discussion of how learning rates can be applied and developed to take the maturity of certain sub-systems, and the options for retro fitting into account. I note that the authors do seem to apply different learning rates to different subsystems, but do not explain if/how they considered the varying maturity of the sub-systems.

Response: Both “Method Section” and “Section Future costs of blue hydrogen without and with tax credits” on Page 7 discuss how to apply learning rates in capital and O&M costs of individual subsystem in a blue hydrogen production plant to develop an overall plant learning curve. However, we did not explain them sufficiently in terms of this feedback. As the reviewer suggested, we expanded the discussions regarding learning rate discovery and maturity issue.

We expanded the relevant discussions in the method section, which are highlighted in blue color: “To construct a learning curve for a technology with respect to its either total capital cost or total O&M cost, three types of model parameters have to be specified, including the initial cost, initial installed capacity, and learning rate. Capital and O&M learning rates can be estimated using empirical data for mature technologies or an analogous approach for advanced technologies. For example, the learning rates for SMR were derived from its historical installed capacity and cost data¹³, whereas the learning rates for post-combustion carbon capture were estimated by referring to those of post-combustion flue-gas desulfurization as they are technically analogous^{14,42}. The data collected for these parameters are discussed later.

A hydrogen production plant involves multiple technologies or subsystems, which have different values regarding the three parameters defining a learning curve. At a blue hydrogen production, individual subsystems lie at different levels of technological maturity. Learning rates and initial installed capacity count on maturity level and then vary by subsystem. For example, at a gas-based blue production plant, SMR and PSA are mature subsystems, whereas carbon

capture has not been deployed widely, though it is commercially available. As a result, the O&M learning rates are zero for SMR and PSA but 22% for carbon capture. Thus, a component-based learning curve model is applied to estimate the total cost of hydrogen production at a certain level of cumulative installed capacity as the sum of individual subsystem costs.”

To clarify how an overall plant’s learning curve is constructed, we further expanded the discussion in the result section by adding to the first paragraph of Section Future Costs of Blue Hydrogen Production without and with Tax Credits new sentences: “A blue hydrogen production plant consists of numerous subsystems. However, the maturity status of individual subsystems and their initial installed capacity are different. As a result, learning rates and initial installed capacity vary by subsystem. Thus, a component-based learning curve model is employed to construct a plant-level learning curve based on individual subsystems’ learning rates and initial installed capacity.”

Comment: Secondly, they identify gas prices as a critical parameter, but do not take the opportunity to discuss how a large blue hydrogen industry could distort or affect the gas markets in the US – essentially providing a large capacity market for US gas as its usage in other sectors declines. Given that they are calculating large cumulative hydrogen production capacities, this would seem to be a very salient point.

Response: This is a nice point. We added to the sensitivity analysis on gas price on Page 13 of the original version a new paragraph: “It is also worth noting that the gas-based hydrogen industry may have a sizable effect on the natural gas markets in the U.S., depending on the scale of blue hydrogen production in the future. For example, the production of 10 MMTA hydrogen by SMR with CCS would consume natural gas of 1.9 billion GJ per year, which is equivalent to about 17% of the national industrial natural gas consumption in 2022³³.”

Comment: Both of the points mentioned above are critical to understand the implications of the findings of the analysis in the paper, and such discussion is expected in high impact, interdisciplinary journals like Nat. Comms.

Response: We have added new discussions, data, and clarifications to address the two valuable comments above.

In addition to the two valuable points, we also added to the Discussion section a new paragraph about competing strategies and supportive policy options:

“Competing strategies and supportive policy and regulatory actions should be made rapidly on both the hydrogen demand and supply sides at both federal and state levels in alignment with the innovation expansion. A variety of high-level strategies are needed on the demand side to promote the widespread use of low-carbon hydrogen in industrial, transportation and power sectors and then establish large-sale markets for low-carbon hydrogen. To jumpstart a hydrogen economy, a cluster approach can be employed on the supply side to establish regional production-transportation-demand networks by co-locating feedstock supply, hydrogen production, and carbon sequestration with multiple end-users and by utilizing existing infrastructure, such as pipeline infrastructure for natural gas, CO₂, and H₂ transportation and geological reservoirs for CO₂ storage. To scale the regional hydrogen economy, secured investments in hydrogen production and supporting infrastructure are required with funding from both public and private sectors, plus subsidies and tax incentives. In addition, deploying large hydrogen production plants instead of small ones can improve engineering economics at a plant level. Given the important role of CCS in producing competitive blue hydrogen, continued

support for large-scale demonstration projects should be boosted in the near term to reduce the CCS cost and its uncertainty. Investments in blue hydrogen should be prioritized to lock down sufficient financial resources for the most competitive technologies in the near term. Economic and policy incentives can be tailored with emphasis on gas-based blue hydrogen to catalyze its widespread deployment and technological evolution because of the pronounced cost advantage relative to coal-based blue hydrogen. Extending the 45Q tax credit from the current 12-year period to a longer period for gas-based blue hydrogen projects would remarkably lower the time-related cumulative installed capacity necessary to reach the Hydrogen Energy Earthshot.”

Comment: Thirdly, the authors do not accurately reflect the LCA emissions for blue hydrogen in the paper. The authors provide a range of emissions intensities for blue hydrogen, then state in the supplementary that ‘In this study, all hydrogen production facilities are assumed to be eligible for the bonus rate of tax credit of Section 45V’. However, the NETL report that they cite for emissions data clearly states that the median LCA emissions intensities will exceed the 4 kgCO₂/kgH₂ limit, and hence that blue hydrogen production will not necessarily qualify for the 45V tax credit. The authors never acknowledge this and the LCA emissions are never directly compared to the requirements for the 45V tax credit (which is tucked away in the supplementary in Table 13). This is a crucial oversight at best, and misleading at worst.

Response: It turned out that we did a poor job in presenting the LCA emissions information. As in response to a similar comment from Reviewer 1 on life cycle emissions, we expanded the statement in Page 11 of the original version to read: “As mentioned earlier, the 45V tax credit depends on the life cycle emissions of hydrogen production, which include greenhouse gas emissions from plant stacks, fuel supply, electric power supply, and CO₂ sequestration or management. The life cycle emissions were estimated by the National Energy Technology Laboratory to range from 3.1 to 8.9 kg CO₂-eq/kg H₂ for the gas-based blue hydrogen in the 90% confidence interval between the 5th and 95th percentile values and from 3.4 to 8.9 kg CO₂-eq/kg H₂ for the coal-based blue hydrogen, which is driven mainly by the uncertainty in fuel supply⁷. The largest contributor among the multiple stages to the life cycle emissions is the fuel supply⁷. The median estimate of life cycle emissions is 4.6 kg CO₂-eq/kg H₂ for the gas-based blue hydrogen and 4.1 kg CO₂-eq/kg H₂ for the coal-based blue hydrogen⁷, which is close to the threshold value of 4.0 kg CO₂-eq/kg H₂ required to claim the minimum tax credit for clean hydrogen. Blue hydrogen projects have a fair possibility of earning a 45V tax credit. Thus, the production tax credit for hydrogen projects is assumed to be \$0.6 per kilogram of H₂ for 10 years. This assumption is optimistic for blue hydrogen in this study. However, there is no 45V tax credit if the life cycle emissions of specific blue hydrogen projects are more than 4.0 kg CO₂-eq/kg H₂. See Supplementary Note 5 for additional information about life cycle emissions and tax credits.”

Comment: I recommend that the authors revise the manuscript to take account of the issues mentioned above, and my detailed comments below, before resubmission.

Response: We appreciate the reviewer for these valuable comments. We have revised the manuscript as much as possible to address the detailed comments.

Detailed comments

Comment 1: Too much of the critical information and context is buried in the supplementary. While I recognize that it is necessary to keep to word limits, the authors should provide more of an overview of the system boundaries they consider in the manuscript itself. This includes a discussion of how learning rates are chosen and applied as discussed above. For example, the NETL report assumes that CO₂ and H₂ exit the plants at particular pressures, and take into account the energy and plant required for this. It is important to state these assumptions and clarify why they are chosen.

Response: We added to the beginning of the Results section one new paragraph, which gives an overview of the system analysis: “This study first characterizes greenhouse gas emissions and costs of commercial technologies for blue hydrogen production and then develops technological learning and diffusion models to assess the future costs and evolutionary trajectories of blue hydrogen production without and with tax incentives toward the U.S. Hydrogen Energy Earthshot. A series of parametric analyses are further performed to reveal the dependence of the overall hydrogen production cost on key factors, such as fuel price, CCS cost uncertainties, learning rates, and inflation rate.”

We cannot include many technical details in the main body of this manuscript due to the word count limit. Thus, many details, such as CO₂ and H₂ product streams’ pressure and purity, are included in the supplementary information document. In addition, we are worried about the distraction from the key focus if many small details were included in the main body of this manuscript. To address the reviewer’s concern, we added the suggested information to the relevant paragraph on Page 6 of the original version: “Blue hydrogen plants produce high-purity hydrogen (99.9 vol.%) at the pressure of 6.48 Mpa and transport the captured CO₂ at the pressure of 15.3 Mpa for storage in saline reservoirs, which are typical design conditions.”

As in response to a similar comment on learning rates above, we expanded discussions in the method and result sections on how learning rates are developed and applied to construct a plant-level learning curve.

Comment 2: Autothermal reforming is projected to be a cost-effective blue hydrogen technology in many other studies as well as the IEA estimates. The authors should justify why they chose not to include it – especially since the NETL report provides data on ATR plants.

Response: To clarify it, we added to the first section of the results section on Page 6 of the original version: “The majority of hydrogen produced in the U.S. is made via SMR. In addition, the cost of blue hydrogen produced by SMR with CCS is similar to that by autothermal reforming with CCS but the on-site and life-cycle emissions from the SMR process are less⁷. This study therefore focuses on SMR with CCS for blue hydrogen production.”

Comment 3: The authors should provide hydrogen production capacity in such a way that is can be compared between data sets. Table 1 quotes H₂ production capacity in cubic meters per hour; Table 2 in GW_{th}; and Figures 1-3 in Mt H₂/year. (I note that Table S12 has both Mt/yr and GW_{th}). Given the focus on learning rates as a function of installed capacity, it is useful for the reader to compare these results to the existing capacity. The authors should clearly provide current installed capacities as Mt H₂/y.

Response: The capacity data presented in Table 1 were collected from the International Energy Agency (IEA)’s database. The database reports the annual production capacity just for some specific hydrogen projects. However, no annual capacity is reported for many projects in the

database. To address this comment, we added the annual capacities of those projects to Table 1, dependent on availability of annual capacity data. Below is the expanded table.

Table 1 Operational blue hydrogen production facilities^a.

Feedstock	Project Name	Country	Online Year	H ₂ Production Capacity		Captured CO ₂ (million metric tons/year)
				(m ³ /hour) ^b	(10 ³ · metric tons/year)	
Natural Gas	PCS Nitrogen	United States	2013	31,344	N/A	0.25
	Port Arthur	United States	2013	125,376	118	1.00
	Enid Fertiliser	United States	1982	87,764	N/A	0.70
	Port Jerome	France	2015	12,538	39	0.10
	Quest	Canada	2015	125,376	300	1.00
	Nutrien (Former Agrium) Fertilizer	Canada	2020	37,613	N/A	0.30
	Al Reyadah CCUS	United Arab Emirates	2016	47,876	N/A	0.80
Coal	Sinopec Qilu Petrochemical CCS	China	2022	41,892	N/A	0.70
	Sinopec Zhongyuan Oilfield EOR	China	2015	5,985	N/A	0.10
	Changqing Oil Field EOR	China	2015	2,992	N/A	0.05
	Great Plains Synfuel Plant and Weyburn-Midale	United States	2000	179,536	N/A	3.00
	Coffeyville Fertilizer Plant	United States	2013	125,376	N/A	1.00
Oil	Shell Heavy Residue Gasification CCU - Pernis Refinery ^c	Netherlands	2005	23,938	1,000	0.40
	Karamay Dunhua Oil Technology CCUS EOR Project	China	2015	5,985	N/A	0.10
	North West Sturgeon Refinery	Canada	2020	77,799	N/A	1.30
	Horizon Oil Sands	Canada	2009 ^d	26,212	N/A	0.44

Table 2 reports the initial installed capacity estimates of individual subsystems in GW_{th} as some subsystems are installed in non-hydrogen production plants. In other words, the installed capacity of a given subsystem includes estimates for hydrogen and/or non-hydrogen production plants or energy systems. To estimate the total installed capacity, the installed capacity estimates for hydrogen and non-hydrogen production plants need to be converted to a common metric, which can also be used for comparisons among the multiple subsystems within a hydrogen plant. In addition, this metric is also often used in the literature. We think that the reporting in Table 2 is appropriate. However, to address the reviewer’s concern about current installed capacities in million metric ton per year, we added to the first paragraph on Page 10 the current capacity information: “At a global scale, the initial installed capacity of hydrogen production in 2021 was estimated to be 0.31 MMTA for gas-based blue hydrogen and 0.15 MMTA for coal-based blue hydrogen^{12,28}.”

Comment 4: Blue hydrogen plants tend to be large, so even a large production capacity increase could represent a small number of new plants, and hence limited learnings. Is this taken into account in the learning rate analysis?

Response: No, the technological learning was evaluated in terms of the cumulative installed capacity of blue hydrogen. To make it clearer, we added to “Section Future costs of blue hydrogen production without and with tax credits” on Page 7 of the original version a new sentence: “In addition, the technological learning is evaluated in terms of the cumulative installed capacity of blue hydrogen instead of the number of new hydrogen production plants.”

Comment 5: Given that the aim is to assess the likelihood of reaching the US Hydrogen Earthshot goal of hydrogen at 1 \$/kg by 2030, the authors should comment on how long it would take to reach the installed capacities required to drive the costs down – would this occur within the next 7 years? This is obviously a difficult task, but should be at least mentioned. For example, how many blue hydrogen plants that would need to be deployed? how quickly can blue hydrogen plants typically be deployed? what is the current pipeline of announced projects?

Response: This comment raises a challenging but valuable task, which involves the diffusion of innovation over time. To address it, we collected additional data, developed a diffusion-of-innovation model, and then applied it to estimate the time needed to reach a certain cost target.

We added the newly-developed diffusion-of-innovation model to the Methods section:

“Diffusion-of-Innovation Model

The diffusion of innovation describes how a new technology would spread over time. An S-shaped curve is often used to measure the diffusion over time, in which the adoption rate increases during the early stage, reaches a maximum level at the point of inflection, and decreases until the diffusion curve saturates⁴⁵. To estimate the annual installed capacity of low-carbon hydrogen over time, the S-shaped diffusion function is employed^{46,47}:

$$cc_t = \frac{cc_{sat}}{1 + \frac{cc_{sat} - cc_0}{cc_0} \cdot e^{-r \cdot t}} \quad (1)$$

Where cc_t is the annual installed capacity of low-carbon hydrogen in a particular year t (million metric tons per annum, MMTA); cc_{sat} is the saturation level of annual installed capacity (MMTA); cc_0 is the initial annual installed capacity (MMTA) in the start year; r is the growth rate (fraction); and t is a particular year after the start period. The function coefficients are estimated by regression based on current and future low-carbon hydrogen production capacities through 2030^{12,28,30}. Additional details about the regression and diffusion function are available in Supplementary Note 7. Once the annual installed capacity in future years is determined, the cumulative annual installed capacity can be estimated as a function of time.”

We added the relevant results to the Results section:

“Time-based diffusion of blue hydrogen production

It is helpful for hydrogen energy planning to explore if certain production capacity and cost targets can be achieved by 2030. A new study reports the cumulative installed capacity of low-carbon hydrogen production over time based on globally announced, planning and committed projects through 2030³⁰. A diffusion-of-innovation model was established based on the current and future low-carbon hydrogen capacities through 2030 to explore the time-based diffusion of gas-based blue hydrogen over a long-term planning horizon through 2050.

Fig. 2a shows the cumulative installed capacity estimates for global low-carbon hydrogen production over time. The gas-based blue hydrogen capacity accounts for 49% of the total low-carbon hydrogen capacity given in Table 1 and is estimated to be 90% in 2030 in terms of the International Energy Agency’s hydrogen project databases^{28,31}. Given the changing shares over time, Fig. 2a also shows a range of cumulative installed capacity for gas-based blue hydrogen in a particular year. The cumulative installed capacity of the global gas-based blue hydrogen may range from 6 to 12 MMTA in 2030, which implies that it would be hard for the blue hydrogen production by SMR with CCS alone in the U.S. to reach 10 MMTA in 2030.

Fig. 1c shows the overall plant LCOH as a function of cumulative installed capacity for gas-based blue hydrogen, whereas Fig. 2a show the cumulative installed capacity over time. Combining them together, Fig. 2b shows the overall plant LCOH of gas-based blue hydrogen production without tax credit over time. The result shown in Fig. 2b implies that for the fuel price and learning rates given in the base case, it would also be difficult for gas-based blue hydrogen to reach the ambitious cost target of \$1/kg H₂ by 2030 in normal scenarios without aggressive incentives and game-changing technologies.

Fig. 2 Diffusion of cumulative installed capacity of low-carbon hydrogen and time-based learning curves of gas-based blue hydrogen. (a) Diffusion of cumulative installed capacity; (b) Time-based learning curves of blue hydrogen production cost.

Additional details about the regression and diffusion model were added to the SI:

“Supplementary Note 7: Development of a Time-Based Diffusion Model of Low-Carbon Hydrogen

To estimate the cumulative installed capacity over time, a diffusion-of-innovation model is developed based on the current hydrogen capacity and the future hydrogen capacity by 2030 that includes announced, planning and committed projects. The scatter points shown in Supplementary Fig. 5a represent the installed capacity of global fossil fuels with CCUS for low-carbon hydrogen production in 2021 and the cumulative installed capacity of low-carbon production, which will be produced in 2024–2030. These data were retrieved from the IEA studies and a new study by the Hydrogen Council and McKinsey & Company^{8,9,14}, respectively. The fitting curve was regressed on the scatter points shown in Supplementary Fig. 5a, which is a second-order polynomial function. This fitting curve was then used to calculate the annual installed capacity of low-carbon hydrogen from 2021 to 2029, as shown in Supplementary Fig. 5b. A diffusion model was further formulated by regression on the annual installed capacity

estimates through 2029. Supplementary Fig. 6 shows the diffusion model and the annual installed capacity of global low-carbon hydrogen through 2050.

Supplementary Fig. 5. Cumulative and annual installed capacity of global low-carbon hydrogen over time. (a) Cumulative installed capacity; (b) Annual installed capacity.

Supplementary Fig. 6. Projection of annual installed capacity of global low-carbon hydrogen through 2050.

Comment 6: The authors state that ‘the cost of blue hydrogen produced by SMR with CCS approximates to the Hydrogen Earthshot, as shown in Fig. 1e.’, however, this is difficult to see on the graph. Suggest that the authors add a line to indicate the earth shot cost – as is done in Fig 2 b-d and Fig 3.

Response: As suggested, a dash line was added to all the relevant figures to indicate the Hydrogen Energy Earthshot cost (\$1/kg H₂).

Issues with range of LCA given and discussion around tax credit 45V:

Comment 7: The authors state that “the production tax credit for hydrogen projects is \$0.6 per kilogram of H₂ for 10 years as the life cycle GHG emissions of gas- and coal-based production plants for blue hydrogen are estimated to be 3.13 to 8.86 and 3.40 to 8.87 kg CO₂-equivalent per kilogram of hydrogen, respectively”. However, looking into the NETL report, the median is 4.6 kgCO₂/kgH₂ (Exhibit 3-52, pg 126). The assumption that blue hydrogen plants will get the 45V tax credit is clearly flawed, and the lack of transparency around this is misleading. One way to

get round this would be to ‘look under the hood’ of the NETL analysis and highlight under which conditions blue hydrogen plants could meet the emissions requirements. This would correct the misleading assumption and improve the utility of the analysis in the paper.

Response: As the reviewer suggested, we further looked into the NETL analysis. As in response to one similar comment above, we significantly expanded the life cycle statement to read: “As mentioned earlier, the 45V tax credit depends on the life cycle emissions of hydrogen production, which include greenhouse gas emissions from plant stacks, fuel supply, electric power supply, and CO₂ sequestration or management. The life cycle emissions were estimated by the National Energy Technology Laboratory to range from 3.1 to 8.9 kg CO₂-eq/kg H₂ for the gas-based blue hydrogen in the 90% confidence interval between the 5th and 95th percentile values and from 3.4 to 8.9 kg CO₂-eq/kg H₂ for the coal-based blue hydrogen, which is driven mainly by the uncertainty in fuel supply⁷. The largest contributor among the multiple stages to the life cycle emissions is the fuel supply⁷. The median estimate of life cycle emissions is 4.6 kg CO₂-eq/kg H₂ for the gas-based blue hydrogen and 4.1 kg CO₂-eq/kg H₂ for the coal-based blue hydrogen⁷, which is close to the threshold value of 4.0 kg CO₂-eq/kg H₂ required to claim the minimum tax credit for clean hydrogen. Blue hydrogen projects have a fair possibility of earning a 45V tax credit. Thus, the production tax credit for hydrogen projects is assumed to be \$0.6 per kilogram of H₂ for 10 years. This assumption is optimistic for blue hydrogen in this study. However, there is no 45V tax credit if the life cycle emissions of specific blue hydrogen projects are more than 4.0 kg CO₂-eq/kg H₂. See Supplementary Note 5 for additional information about life cycle emissions and tax credits.”

Reviewers' Comments:

Reviewer #1:

Remarks to the Author:

The authors have addressed all my comments. I believe that the paper can be now accepted in its present form.

Reviewer #2:

Remarks to the Author:

The authors have addressed my main points of concern and the manuscript has improved substantially. I recommend publication, subject to a some more minor revisions.

It's nice to see that the manuscript has come along quite well. Congratulations to the authors, I can imagine that this quite a bit of extra work. The literature review of learning-curve studies is very interesting. It surprises me to find out that so little has happened in that field since the papers by Rubin et al and Schoots et al. The additional sensitivity analysis of the learning rate is very important and helpful given this parameter has a high uncertainty. The figures have improved a lot (I find Fig. 1e particularly helpful). Nice work about with the additional section and figure on the S-curve diffusion. Also a lot of other things got clarified and made more explicit.

Here are my remaining main points that I think should be addressed before publication. I don't think it'll require substantial changes or additional analyses, it's mainly about a few textual changes.

- I am still a little worried that that policymakers and stakeholders might perceive the storyline in this paper as an endorsement for strong investment in blue hydrogen, especially in comparison with green hydrogen. I know that this manuscript is not at all focussing on that debate but rather just on scaling up blue hydrogen, but we all know how scientific papers in high-impact journals sometimes are read and cited, even more so when published Open Access. Most importantly, while blue hydrogen may play a crucial role in bridging the hydrogen supply gap, it comes with (potentially very) high residual supply-chain emissions, which purely green hydrogen does not have. In particular:
 - The authors state CO₂ avoidance cost for blue hydrogen based on stack emissions, not lifecycle emissions (lines 186--196 and Supplementary Note 2).
 - The authors state that green hydrogen cost will stay above blue hydrogen cost, because learning rates for electrolysis are similar to those for SMR. I agree with both statements alone, but the relation between the two is flawed. A substantial share of cost reductions for green hydrogen will be driven by lower electricity cost due to declining cost of solar PV and wind and an increase in efficiency. So, while green will remain expensive in the short term, it could even reach cost parity with blue in the long run if renewables cost continue to decline as they have in the past 20 years.
 - The authors state a few times that green hydrogen is and will be more expensive, but without contextualising this with the potential of green to have lower abatement cost than blue (even when production cost remains higher), which would be relevant in the context of future tax credits and other regulation.
 - (As a side note to the CO₂ avoidance cost calculation: the authors assume grey hydrogen as the reference. Grey hydrogen today is, to the best of my knowledge, mostly used in ammonia production and in refineries. Those applications could be among the first applications to make use of blue hydrogen, despite the uncertainty about the future role of refineries in net-zero transformation pathways and the option for carbon capture on integrated "blue ammonia". Though I think the more interesting question is not the decarbonisation of those small shares of grey hydrogen produced today, but rather the replacement of direct use of fossil (natural) gas in industry more generally. Therefore, I would naturally compare blue hydrogen against fossil gas on the basis of its heating value or in the context of a specific application, e.g. direct reduction of steel. But I know this goes way too far for the presented manuscript, so sticking with grey hydrogen as a reference could be easier, while noting that CO₂ avoidance cost for other applications when competing against fossil gas will likely be quite a bit

higher.)

- As a follow-up point to the previous point: if green hydrogen plays an important role, one could arrive at the conclusion that hydrogen hubs should not (or at least not only) be co-located with the availability of fossil gas and carbon sequestration but rather with good potentials for solar PV and wind. This is a good example for why conclusions from the authors' work on blue hydrogen should not be taken out of the wider context.
- In general, the manuscript demonstrates that it will be hard for blue hydrogen cost to reach the 1 USD/kgH₂ target. It would be nice to put this into more context:
 - As the authors already agreed, the 1 USD/kgH₂ number is a bit of a tagline more than a carefully derived target, especially given it's not even defined properly (real vs nominal, inflation, etc). I would therefore encourage the authors to point in a few other directions about what targets are actually needed to drive the net-zero transition via hydrogen. Ultimately, nobody should actually care about that 1 USD/kgH₂ number, but rather about what the expected LCOH might mean for the competition with unabated fossils.
 - Moreover, if tax credits and learning-by-doing aren't enough to drive down LCOH to 1 USD/kgH₂ (=30 USD/MWh) or let alone down to the cost of fossil gas (~15 USD/MWh), then how could the US push forward its net-zero transition and the scale-up of the hydrogen economy? My focus is rather on EU policy and over here there is broad agreement among academia and policymakers that (in spite of all subsidies and support schemes) an EU-wide carbon price can and will be the backbone of the net-zero transition, especially for industry and hydrogen. In fact, many of my colleagues are still puzzled about how the US will be able to effectively cut emissions without a federal carbon tax or emissions-trading scheme. I know that again this takes the discussion quite far away from learning curves for blue hydrogen, but it would add a lot of highly relevant context and make it so much clearer why the results of the manuscript and hitting/missing that 1 USD/kgH₂ target matters in the wider policy context.
- Once more re SMR vs ATR: I think it's fine to stick with SMR and only mention ATR (alongside many other technologies) on the side. However, my understanding is that ATR really would be the technology to go for when producing blue hydrogen because it's more efficient (even though it increases CAPEX) and because it's easier to achieve high capture rates. For instance, this recent study [[doi:10.1016/j.enconman.2023.116840](https://doi.org/10.1016/j.enconman.2023.116840)] states in the abstract that "[t]he techno-economical assessment of many recent studies has indicated that the oxygen-based system, such as auto-thermal reforming and partial oxidation, is the most efficient for producing greenfield blue hydrogen". I understand that the NETL study arrives at slightly higher emissions for ATR compared to SMR. Most of the increase in lifecycle emissions is however due to grid electricity (which should be fairly easy to decarbonise), while the stack emissions (which are anyways small compared to supply-chain emissions) are slightly smaller due to the capture rate being smaller by a few percentage points (which I would imagine could be compensated without too much additional capex/fuel demand) and while supply-chain emissions from fossil gas are even lower for ATR due to higher efficiency. In summary, I would agree with the study cited above that ATR is probably the go-to solution for greenfield investment in blue hydrogen. I think it's fine that the present manuscript looks more closely at SMR rather than ATR, but perhaps the authors could more prominently flag that ATR is a strong alternative candidate for the production of blue hydrogen.

Here are a few more minor points that the authors can implement as they see fit:

- Fig. 5a: Nice. But I wonder why you are comparing different cases in the same way as in Fig. 1 (different tax credits in the same subplot, varying learning rate across the two subplots) and not as in Fig. 3 (different tax credits in different subplots, varying learning rate in the same subplot). I would expect Fig. 5a to look like Fig. 3, given that the purpose of both figures is to compare the effect of changing the respective sensitivity parameter and not the effect of the tax credit.
- I encourage the authors to double-check throughout the manuscript that when writing "CO₂ avoided", they really mean only (!) CO₂ avoided and not CO₂eq/GHG avoided. In particular, in reference to my previous point, I would suggest that the manuscript should mostly talk about GHG avoided and not just CO₂ avoided, especially given the IRA tax credits also refer to GHG avoided, not CO₂ avoided.

- The authors talk about CCS in several places throughout the manuscript (eg CCS cost, CCS learning rates, etc). I wonder if this could be made a bit more precise by referring to CO2 capture, compression, transportation, and sequestration as separate things.
- Sorry, maybe I didn't get this or I didn't express that point clearly enough in the previous round but — do the markers in Fig. 1c, 1d, 5a, and 6 serve a purpose? Why don't all these figures look like Fig. 3? E.g. my immediate impression is that they could correspond to some real-world historic data that is fitted by the curve – but that's not the case, unless I really missed something. Can't this just be a simple line? Or at least, could the markers be equally spaced? The authors may feel free to leave them in if they really think those markers are useful, but I personally just get distracted and keep looking for an explanation for what they are.
- The colour scheme often uses black, blue, and green. The immediate impression I get from this is that those would correspond to black, blue, and green hydrogen (which they don't). You could switch to other colours (eg black, red, orange, brown, etc), which would reduce that risk. Sorry, this is me wanting to understand the key messages of Nature Journal articles in 3 minutes by looking at only the main figures (which I suppose more people other than myself do).

In summary, I once more congratulate the authors on substantially improving their manuscript. It's in a good shape and should get published in NCOMMS. My remaining suggested points are really there to ensure the manuscript is well embedded in the wider academic and political debate about technologies and regulation related to hydrogen, such that the derived results will receive the attention they deserve.

Reviewer #3:

Remarks to the Author:

As mentioned in my initial review, this paper looks at the projected cost of blue hydrogen as a function of different learning rates, taking into account the IRA tax credits for hydrogen. It relies on a recent comprehensive report by NETL, which provides process level comparisons of state-of-the-art, fossil-based hydrogen production technologies. The authors use their analysis to assess the likelihood of reaching the US Hydrogen Earthshot goal of hydrogen at 1 \$/kg by 2030.

Given the importance of hydrogen in future global energy systems, and the uncertainties around technology costs, understanding the cost evolution of hydrogen technologies is a useful and timely addition to the literature.

The authors have significantly improved the analysis of the paper to address my initial concerns regarding the application of the learning rates and the emissions intensities of blue hydrogen. They include a new analysis incorporating a technology diffusion model to like learning rates in terms of capacity to the 2030 target, and in doing so are able to draw novel and important conclusions which very much improve the paper. They have also addressed all of my comments and concerns to my satisfaction.

My final suggestion would be to avoid statements about the future competitiveness of green hydrogen with blue hydrogen. This is a very big question with a lot of different aspects to it. Instead, authors could potentially acknowledge this and cite one of the many works looking at this point directly.

The comments from all the three reviewers are very valuable, which helped us significantly improve the manuscript to reach a new height of quality for publication. To express our sincere appreciation, we expanded the acknowledgement section by adding: “In addition, the authors also want to deliver special thanks to the three anonymous reviewers for their great comments, which improved the quality of this study during the revision process.”

REVIEWER COMMENTS

Reviewer #1 (Remarks to the Author):

Comment: The authors present an analysis on future blue hydrogen costs in the US based on different technologies, learning rates and natural gas prices. They compare the effect of two different tax credits that are in place, in the perspective of reaching a target cost of 1 USD/kg for blue hydrogen.

Response: The authors greatly appreciate the reviewer for time and valuable comments. Noted.

Comment: The results are of interest for the research community, given the importance of the topic. Unfortunately, I believe that the present version of the manuscript has some important issues, and it is not suitable for publication on an international journal such as *Nature Communications*. I report my comments below.

Response: As the reviewer pointed out, this manuscript addresses a timely important topic on low-carbon hydrogen production and provides important results, findings, and new insights, which are of interest to the community in this field. We have performed a large amount of additional quantitative and qualitative analyses, discussions, and clarifications to address the reviewer and other two reviewers’ critical comments, which have helped us significantly improve the manuscript. We think the revised manuscript is a fit for broad readership of multidisciplinary journals like *Nature Communications*.

Comment: The economic analysis provided by the authors is detailed and include many different sub-systems and parameters. However, in a general perspective, I believe that other very important aspects are not discussed at all. The crucial point is the CO₂ emission savings that can be achieved by blue hydrogen, especially compared to green hydrogen, which appears to be the elephant in the room in this analysis. The authors briefly mention lifecycle emission ranges at page 11, but no additional information is given nor discussed. I think indeed this is a crucial point that should be addressed to support the main results. The authors could also estimate a cost per avoided tonne of CO₂ emission to be compared to other technologies and studies.

Response: We appreciate the reviewer for pushing us to think more broadly about this critical topic. We collected additional data of stack CO₂ emissions from the production of grey hydrogen, blue hydrogen, and green hydrogen to quantify the CO₂ emission savings by blue hydrogen relative to grey hydrogen and further compare it with green hydrogen. We further estimated the CO₂ avoidance costs by blue and green hydrogen and made a comparison between the two low-carbon hydrogen production pathways. Additional information of life cycle emissions was added to the main paper.

While revising the main paper in response to this comment, we added to the end of Section Current Blue Hydrogen Production a new paragraph: “Currently, hydrogen is mainly produced by SMR without CCS in the U.S., which is often called grey hydrogen. Compared to it, the blue hydrogen production by SMR with CCS can decrease the stack CO₂ emission intensity by 96% but increase the LCOH by 55%⁷. The resulting CO₂ avoidance cost by blue hydrogen is \$65 per metric ton of CO₂. In contrast, the green hydrogen production by polymer electrolyte membrane electrolyzers almost has no stack CO₂ emissions but a high LCOH value ranging from \$3.0–7.5/kg H₂²⁷. The resulting CO₂ avoidance cost by green hydrogen relative to grey hydrogen varies from \$212–689 per metric ton of CO₂, which is much higher than that by blue hydrogen. Obviously, there are tradeoffs in CO₂ avoidance cost and emission savings between the blue and green production pathways. The details of emission and cost data and CO₂ avoidance cost estimation are available in Supplementary Note 2.”

We added a new supplementary section:

Supplementary Note 2: Cost of CO₂ Avoided by Blue and Green Hydrogen

The cost of CO₂ avoided by blue or green hydrogen relative to grey hydrogen can be estimated in terms of the production plant’s CO₂ emission intensity and levelized cost of hydrogen:

$$CCA = \frac{LCOH_{\text{blue/green}} - LCOH_{\text{grey}}}{EI_{\text{grey}} - EI_{\text{blue/green}}} \quad (S1)$$

Where CCA is the cost of CO₂ avoided (\$/kg CO₂); LCOH is the levelized cost of hydrogen (\$/kg H₂); EI is the stack CO₂ emission intensity of a production plant (kg CO₂/kg H₂); the subscript of “blue” represents the gas-based blue hydrogen production, the subscript of “green” represents the green hydrogen production, and the subscript of “grey” represents the gas-based grey hydrogen production without carbon capture.

Supplementary Table 5 summarizes the stack CO₂ emission intensity and LCOH of grey, blue, and green hydrogen production plants, which are collected from the literature^{1,4}. The resulting costs of CO₂ avoided by blue and green hydrogen relative to grey hydrogen are also provided in Supplementary Table 5.

Supplementary Table 5. Hydrogen Production Emission, Production Cost, and Carbon Avoidance Cost

Production Technology	Stack Emission Intensity (kg CO ₂ /kg H ₂)	Levelized Cost of Hydrogen (2018\$/kg H ₂)	Cost of CO ₂ Avoided (\$/metric ton CO ₂)
Steam Methane Reforming (SMR) (Grey)	9.35	1.06	Reference
SMR with CCS (Blue)	0.38	1.64	65
Distributed Polymer Electrolyte Membrane Electrolyzers (Green)	0 ^a	5.17 (3.0–7.5) ^b	440 (212-689)

^aZero carbon emissions are assumed for the electrolysis powered by solar photovoltaics as it produces almost no direct emissions during operation⁵.

^bThe cost was estimated for a production capacity of 1,500 kg H₂/day with an effective electricity price of 7.55 ¢/kWh⁴. The cost was further adjusted to 2018 dollars using the annual average consumer price index for all urban consumers⁶.

To clarify the life cycle emission estimation in detail, we expanded the statement in Page 11 of the original version to read: “As mentioned earlier, the 45V tax credit depends on the life

cycle emissions of hydrogen production, which include greenhouse gas emissions from plant stacks, fuel supply, electric power supply, and CO₂ sequestration or management. The life cycle emissions were estimated by the National Energy Technology Laboratory to range from 3.1 to 8.9 kg CO₂-eq/kg H₂ for the gas-based blue hydrogen in the 90% confidence interval between the 5th and 95th percentile values and from 3.4 to 8.9 kg CO₂-eq/kg H₂ for the coal-based blue hydrogen, which is driven mainly by the uncertainty in fuel supply⁷. The largest contributor among the multiple stages to the life cycle emissions is the fuel supply⁷. The median estimate of life cycle emissions is 4.6 kg CO₂-eq/kg H₂ for the gas-based blue hydrogen and 4.1 kg CO₂-eq/kg H₂ for the coal-based blue hydrogen⁷, which is close to the threshold value of 4.0 kg CO₂-eq/kg H₂ required to claim the minimum tax credit for clean hydrogen. Blue hydrogen projects have a fair possibility of earning a 45V tax credit. Thus, the production tax credit for hydrogen projects is assumed to be \$0.6 per kilogram of H₂ for 10 years. This assumption is optimistic for blue hydrogen in this study. However, there is no 45V tax credit if the life cycle emissions of specific blue hydrogen projects are more than 4.0 kg CO₂-eq/kg H₂. See Supplementary Note 5 for additional information about life cycle emissions and tax credits.”

Comment: The literature review seems quite limited, especially given the large number of studies on the subject in the last years. The learning rates reported by the authors (page 5) rely on literature that is more than 15 years old, which seems quite outdated considering the recent technological evolution. I think that using more recent data would improve the quality and reliability of the results.

Response: This study mainly cited the two pioneering learning curve studies by Rubin et al. (2007) and Schoots et al. (2008). Although a large number of technological learning studies have been done in the past years, almost no breakthroughs have been made on learning rates beyond the two pioneering studies in the past years. The learning rates from the two pioneering studies have been frequently adopted in the recent studies. In other words, there are no more rigorous estimates of learning rates in the literature than the two pioneering studies cited in our manuscript. To make it clear, we added to the relevant paragraph in the introduction section on Page 5 of the original version new sentences:

“Although a large number of learning curve studies have been done in the past years, almost no breakthroughs have been made in estimating the learning rates for CCS, gasification, and SMR beyond the two pioneering studies by Rubin et al. (2007)¹⁴ and Schoots et al. (2008)¹³. Recent studies frequently adopted the learning rates from the two pioneering studies for a variety of applications¹⁷⁻²⁵.”

The added references include:

- Böhm, H., Zauner, A., Rosenfeld, D. C. & Tichler, R. Projecting cost development for future large-scale power-to-gas implementations by scaling effects. *Appl. Energy* **264**, 114780 (2020).
- Fan, J. L., Li, Z., Li, K. & Zhang, X. Modelling plant-level abatement costs and effects of incentive policies for coal-fired power generation retrofitted with CCUS. *Energy Policy* **165**, 112959 (2022).
- Fan, J. L., Xu, M., Yang, L., Zhang, X. & Li, F. How can carbon capture utilization and storage be incentivized in China? A perspective based on the 45Q tax credit provisions. *Energy Policy* **132**, 1229–1240 (2019).

- George, J. F., Müller, V. P., Winkler, J. & Ragwitz, M. Is blue hydrogen a bridging technology? The limits of a CO₂ price and the role of state-induced price components for green hydrogen production in Germany. *Energy Policy* **167**, 113072 (2022).
- Kang, J. N., Wei, Y. M., Liu, L., Han, R., Chen, H., Li, J., Wang, J. W. & Yu, B. Y. The prospects of carbon capture and storage in China's power sector under the 2 °C target: a component-based learning curve approach. *Int. J. Greenhouse Gas Control* **101**, 103149 (2020).
- Lee, H., Lee, J. & Koo, Y. Economic impacts of carbon capture and storage on the steel industry - A hybrid energy system model incorporating technological change. *Appl. Energy* **317**, 119208 (2022).
- Malhotra, A., & Schmidt, T. S. Accelerating low-carbon innovation. *Joule* **4**, 2259–2267 (2020).
- Nicodemus, J. H. Technological learning and the future of solar H₂: A component learning comparison of solar thermochemical cycles and electrolysis with solar PV. *Energy Policy* **120**, 100–109 (2018).
- Yang, L., Xu, M., Yang, Y., Fan, J. & Zhang, X. Comparison of subsidy schemes for carbon capture utilization and storage (CCUS) investment based on real option approach: Evidence from China. *Appl. Energy* **255**, 113828 (2019).

Comment: Furthermore, CCS in real applications has shown a lower economic performance than expected, and I think this potential issue should be addressed in the discussion and maybe also in the sensitivity analysis.

Response: There are uncertainties in CCS cost estimates. As the reviewer suggested, we performed a parametric analysis on CCS systems' process and project contingences and added the relevant analysis, results, and discussion to the sensitivity analysis section:

“There are uncertainties in the process and project contingences of two CO₂ removal systems employed for producing low-carbon hydrogen from natural gas resources. Such uncertainties affect the total as-spent capital (TASC) and LCOH of a hydrogen production plant. The process contingency depends on the maturity level of a technology, whereas the project contingency depends on the availability of site-specific project details. In the base case, the process contingency is 18% of the bare erected cost (BEC) for the Cansolv system and 0% for the MDEA system, while the project contingency is 25% of the sum of BEC, engineering, construction management, home office and fees, and process contingency and 25% for the Cansolv unit and 20% for the MDEA unit⁷. A parametric analysis is then conducted to reveal the collective impacts of uncertain process and project contingencies, which takes into account low and high contingencies. In the low contingencies scenario, the process contingency is 10% for the Cansolv system and 0% for the MDEA system, while the project contingency is 10% for both the CO₂ removal systems. In the high contingencies scenario, the process contingency is 40% for both the CO₂ removal systems, while the project contingency is 30% for both the CO₂ removal systems³⁴.

As shown in Figure 4, the uncertainties in CCS cost estimates have a sizable effect on the hydrogen production plant's TASC. As a result, the plant LCOH varies from \$1.45-1.48/kg H₂ at

the cumulative installed capacity of 10 MMTA. To reach the cost of \$1.46/kg H₂, the cumulative installed capacity requirements vary from 7 to 16 MMTA. These results imply that cost uncertainties in CCS systems may result in pronounced variations in the estimation of the cumulative installed capacity necessary to reach a cost target.”

Fig. 4 Effects of process and project contingencies of CO₂ removal systems on initial capital cost and future levelized cost of gas-based blue hydrogen production.

Comment: An additional issue about CCS is that the authors do not seem to consider any investment cost for storage (tables 10 and 11). I think this is a major issue, as these systems should be built, and are also expected to require an important part of the tax credit that the authors are completely allocating to blue hydrogen production. This aspect should be better discussed in the paper, especially considering the important debate about the suitability of CCS.

Response: The investment in CO₂ storage was considered. Both the CO₂ transport and storage costs were considered in this study and were treated as O&M costs (\$/metric ton CO₂) in the overall cost estimation instead of a capital cost component, which are shown in Supplementary Tables 10 and 11 of the original manuscript. To make it clearer, we added to Supplementary Table 1 the total CO₂ transport and storage cost: “total CO₂ transport and storage cost: \$10/metric ton CO₂.”

Comment: The authors consider a tax credit that can be claimed for lifecycle GHG emissions lower than 4 kgCO₂/kgH₂. However, the GHG emission ranges that they provide at page 11 show median values that are much higher than 4. How do they justify this assumption? Please clarify this issue.

Response: The median values of life cycle emissions are close to the threshold value of 4.0 kg CO₂-eq/kg H₂. As we addressed in response to a similar comment above, we expanded the discussion with additional information: “The life cycle emissions were estimated by the National Energy Technology Laboratory to range from 3.1 to 8.9 kg CO₂-eq/kg H₂ for the gas-based blue hydrogen in the 90% confidence interval between the 5th and 95th percentile values and from 3.4 to 8.9 kg CO₂-eq/kg H₂ for the coal-based blue hydrogen, which is driven mainly by the uncertainty in fuel supply⁷. The largest contributor among the multiple stages to the life cycle emissions is the fuel supply⁷. The median estimate of life cycle emissions is 4.6 kg CO₂-eq/kg H₂ for the gas-based blue hydrogen and 4.1 kg CO₂-eq/kg H₂ for the coal-based blue hydrogen⁷,

which is close to the threshold value of 4.0 kg CO₂-eq/kg H₂ required to claim the minimum tax credit for clean hydrogen. Blue hydrogen projects have a fair possibility of earning a 45V tax credit. Thus, the production tax credit for hydrogen projects is assumed to be \$0.6 per kilogram of H₂ for 10 years. This assumption is optimistic for blue hydrogen in this study. However, there is no 45V tax credit if the life cycle emissions of specific blue hydrogen projects are more than 4.0 kg CO₂-eq/kg H₂.”

Comment: The results of the authors are expressed with respect to the potential blue hydrogen deployment capacity. How these capacities compare to the expected demand of blue hydrogen by 2030? I think this information is fundamental to put the results in the right perspective.

Response: The U.S. Department of Energy’s National Clean Hydrogen Strategy and Roadmap report provides an estimate of total national clean hydrogen from both renewable and decarbonized fossil resources by 2030 instead of blue hydrogen alone. To address this comment on the broad perspective, we added to the relevant results on Page 10 a new sentence: “The annual demand for clean hydrogen produced from renewable and decarbonized fossil resources in the U.S. may reach 10 million metric tons of hydrogen per year by 2030¹.”

We further added to the Discussion section a new paragraph:

“The global production capacity of low-carbon hydrogen will reach 12.3 MMTA by 2030 based on the announced, planning and committed projects³⁰. The low-carbon hydrogen capacity in North America will reach 6.8 MMTA by 2030³⁰. However, only 1.8 MMTA³⁰ and 1.5 MMTA^{1,2} of the announced projects in North America and the U.S. have reached final investment decision (FID), mainly because many announced projects have not yet secured financing and nailed down contracted offtake^{1,2}. The hesitancy to long-term, scaled contracts is influenced by numerous factors, such as lack of price certainty, unavailability and reliability of large-scale hydrogen supply, near-term policy implementation uncertainty, and long-term political uncertainty^{1,2}. For blue hydrogen projects, enhancements in tax credits for carbon sequestration can improve the economics of hydrogen production. For example, an extension of the 45Q tax credit period from current 12 years to 18 years would significantly reduce the cumulative installed capacity required for gas-based blue hydrogen projects to reach the Hydrogen Energy Earthshot, as demonstrated in Supplementary Fig. 3. Extending the period of the 45Q tax credit for blue hydrogen projects can be considered an option to secure financing and promote long-term offtake.”

Supplementary Fig. 3. Effect of 45Q tax credit period on future cost of gas-based blue hydrogen production.

Comment: The sensitivity analysis (or at least the discussion) may also address the potential effect of different levels of inflation (and possibly other economic parameters), given the significant variations over the last years. Would this affect the reliability of the results?

Response: The \$1/kg target is the U.S. DOE’s initiative goal, which has no specific information regarding cost type (real vs. nominal dollars) and inflation. This study reports cost results in real dollars. To address this comment, we conducted an additional parametric analysis and estimated the cost results in nominal dollars to demonstrate the effect of inflation rate on the future cost of hydrogen production toward the Hydrogen Energy Earthshot. The results imply that inflation would remarkably raise challenges for blue hydrogen production to reach the cost target. We added to the Sensitivity Analysis section the new analysis and relevant results:

“Inflation Rate

In general, this study estimates the cost of hydrogen production in real dollars. When the cost is estimated in nominal dollars, however, both the initial and future LCOH estimates vary with inflation rate as it affects discount rate, fixed charge rate, and levelization factor. A parametric analysis was further performed for inflation rate to quantify its effect on the evolving cost of gas-based blue hydrogen production toward the Hydrogen Energy Earthshot. Fig. 6 shows the learning curves of blue hydrogen production with inflation. Fig. 6a and 6b show that at a given level of cumulative installed capacity, the LCOH in nominal dollars increases when the inflation rate increases from 1% to 3%. As a result, blue hydrogen production may not reach the cost target of \$1/kg H₂ for both the scenarios without and with 45Q tax credit even when the cumulative installed capacity reaches 30 MMTA. Fig. 6c further shows that with an inflation rate of 3%, the future LCOH may get close to the cost target when cheap natural gas resources are used as the feedstock to produce blue hydrogen with the cumulative installed capacity of up to 30 MMTA.

Fig. 6a and 6b also compare the learning curves of blue hydrogen production between the two scenarios without and with inflation. As shown in Fig. 6a for the scenario without 45Q tax credit, the reduction in hydrogen production cost from deploying the cumulative installed capacity of 10 MMTA can be offset by an inflation rate of 1%. There is a similar result for the scenario with 45Q tax credit, as shown in Fig. 6b. All these results imply that inflation would

remarkably raise challenges for blue hydrogen production to reach the Hydrogen Energy Earthshot in the near future.”

Fig. 6 Effect of inflation rate on future cost of gas-based blue hydrogen production without and with 45Q tax credit. (a) Levelized cost of hydrogen production with gas price of \$4.2/GJ and without 45Q tax credit; (b) Levelized cost of hydrogen production with gas price of \$4.2/GJ and 45Q tax credit; (c) Levelized cost of hydrogen production with 3% inflation rate and 45Q tax credit.

To explain how to estimate the LCOH in nominal dollars, we added a new supplementary session:

“Supplementary Note 8: Hydrogen Production Cost Estimation in Nominal Dollars

The levelized cost of hydrogen and is estimated in nominal dollars¹:

$$\text{LCOH}_N = \text{LCC}_N + \text{LOM}_N + \text{LFP}_N \quad (\text{S12})$$

$$\text{LCOH}_N^{\text{TC}} = \text{LCC}_N + \text{LOM}_N + \text{LFP}_N - \text{LTC}_N^{45\text{Q}} \quad (\text{S13})$$

Where the subscript “N” means the nominal dollars; LCOH_N is the levelized cost of hydrogen (\$/kg H₂); LCC_N is the levelized capital cost (\$/kg H₂); LOM_N is the non-fuel levelized operating and maintenance cost (\$/kg H₂); LFP_N is the levelized fuel price (\$/kg H₂); $\text{LCOH}_N^{\text{TC}}$ is the levelized cost of hydrogen with a tax credit (\$/kg H₂); $\text{LTC}_N^{45\text{Q}}$ is the levelized tax credit over the project book lifetime of a blue hydrogen project (\$/kg H₂).

The levelized capital cost in nominal dollars is estimated using Supplementary Equation (14), while the fixed charge rate is estimated using Supplementary Equations (15) to (21)^{2,3,15}:

$$\text{LCC}_N = \frac{\text{TASC}_R \cdot \text{FCR}_N}{(\text{CF} \cdot \text{AH}) \cdot \text{KG}_{\text{H}_2}} \quad (\text{S14})$$

$$\text{FCR}_N = \frac{\text{CRF}_N^{\text{nonfuel}} - \text{ETR} \cdot \text{PV}_{\text{plant}}}{1 - \text{ETR}} \quad (\text{S15})$$

$$\text{CRF}_N^{\text{nonfuel}} = \frac{\text{ATWACC}_N \cdot (1 + \text{ATWACC}_N)^{\text{BL}}}{(1 + \text{ATWACC}_N)^{\text{BL}} - 1} \quad (\text{S16})$$

$$\text{PV}_{\text{plant}} = \text{CRF}_N^{\text{nonfuel}} \cdot \sum_{n=1}^m \frac{d_n}{(1 + \text{ATWACC}_N)^n} \quad (\text{S17})$$

$$\text{ATWACC}_N = \text{PC}_{\text{equity}} \cdot \text{ROE}_N + \text{PC}_{\text{debt}} \cdot \text{kd}_N \cdot (1 - \text{ETR}) \quad (\text{S18})$$

$$\text{ROE}_N = (1 + \text{ROE}_R) \cdot (1 + N) - 1 \quad (\text{S19})$$

$$\text{kd}_N = (1 + \text{kd}_R) \cdot (1 + N) - 1 \quad (\text{S20})$$

$$N = R + I \quad (\text{S21})$$

Where the superscript “nonfuel” represents the non-fuel component; TASC_R is the total as-spent capital of a blue hydrogen production plant (\$); FCR_N is the fixed charge rate (fraction/year); CF is the plant capacity factor (%); AH is the total annual hours (8760 hours); KG_{H_2} is the hourly production rate (kg H₂/hour); CRF_N is the capital recovery factor (fraction/year); ETR is the effective tax rate (%); PV_{plant} is the present value of tax depreciation expense (fraction/year); ATWACC_N is the after-tax weighted average cost of capital (%); BL is the project book lifetime (year); d_n is the tax depreciation fraction in year n (fraction); m is the number of years of depreciation (year); $\text{PC}_{\text{equity}}$ is the percent of equity (%); ROE_N is the nominal rate of return on equity (%); PC_{debt} is the percent of debt (%); kd_N is the nominal rate of cost of debt; ROE_R is the real rate of return on equity (%); kd_R is the real rate of cost of debt (%); N is the nominal escalation rate (%); R is the real escalation rate (%); and I is the inflation rate (%).

The levelized operating and maintenance cost in nominal dollars is estimated using Supplementary Equation (22), while the levelization factor is estimated using Supplementary Equation (23):

$$\text{LOM}_N = \frac{\text{OM}_R \cdot \text{LF}_N}{(\text{CF} \cdot \text{AH}) \cdot \text{KG}_{\text{H}_2}} \quad (\text{S22})$$

$$LF = CRF_N^{\text{nonfuel}} \cdot \frac{1 - \left[\frac{1 + N}{1 + ATWACC_N} \right]^{BL}}{ATWACC_N - N} \quad (S23)$$

Where OM_R is the non-fuel operating and maintenance (O&M) cost of a blue hydrogen production plant (\$/year); and LF_N is the levelization factor for non-fuel O&M cost (unitless).

The levelized fuel price in nominal dollars is estimated using Supplementary Equation (24), where the fuel's capital recovery factor is estimated using Supplementary Equation (26) and the discount rate in nominal dollars is estimated using Supplementary Equation (27)¹⁶:

$$LFP_N = \frac{FC_N \cdot FR}{KG_{H_2}} \quad (S24)$$

$$FC_N = PV_{\text{fuel}} \cdot CRF_N^{\text{fuel}} \quad (S25)$$

$$CRF_N^{\text{fuel}} = \frac{DR_N \cdot (1 + DR_N)^{BL}}{(1 + DR_N)^{BL} - 1} \quad (S26)$$

$$DR_N = (1 + DR_R) \cdot (1 + N) - 1 \quad (S27)$$

$$PV_{\text{fuel}} = \sum_{t=1}^{BL} \frac{P_t}{(1 + DR_N)^t} \quad (S28)$$

$$P_t = P_{t-1} \cdot (1 + N) \quad (S29)$$

Where the superscript “fuel” represent the fuel component; FC_N is the natural gas cost (\$/GJ); FR is the hourly natural gas flow rate (GJ/hour); PV_{fuel} is the present value of gas cost (\$/GJ); CRF_N^{fuel} is the fuel's capital recovery factor (fraction/year); DR_N is the nominal discount rate for fuel cost (%); DR_R is the real discount rate for fuel cost (%); N is the nominal escalation rate (%); P_t is the fuel cost in year t (\$/GJ), while P_1 is the real fuel cost in the first year of a project.

In this study, the financial structure and assumptions are in alignment with those of the studies by the National Energy Technology Laboratory (NETL)^{1,2,3,16,17}. Supplementary Table 17 lists the financial parameters and summarizes their data and the sources of data. Supplementary Table 18 summarizes the estimates of FCR, levelization factor, and discount rate as a function of nominal escalation rate.

Supplementary Table 17. Financial Parameters and Assumptions

Parameter	Symbol	Unit	Value	Source(s)
Effective Tax Rate	ETR	%	25.74%	NETL (2021) ³
Inflation Rate	I	%	0–3%	Assumption
Real Escalation Rate	R	%	0%	NETL (2011) ²
Percent of Equity	PC _{equity}	%	62%	NETL (2022) ¹
Percent of Debt	PC _{debt}	%	38%	NETL (2022) ¹
Real Rate of Return on Equity	ROE _R	%	3.10%	NETL (2022) ¹
Real Rate of Cost of Debt	kd _R	%	5.15%	NETL (2022) ¹
Real Discount Rate	DR _R	%	4.73%	NETL (2019) ¹⁶
Project Book Lifetime	BL	year	30	NETL (2022) ¹
Fuel Cost	P ₁	2018\$/GJ	4.2	NETL (2022) ¹
Tax Depreciation Fraction	d ₁	%	3.75%	NETL (2021) ³ ; IRS (2016) ¹⁷
	d ₂		7.22%	
	d ₃		6.68%	
	d ₄		6.18%	
	d ₅		5.71%	

	d ₆		5.29%	
	d ₇		4.89%	
	d ₈		4.52%	
	d ₉		4.46%	
	d ₁₀		4.46%	
	d ₁₁		4.46%	
	d ₁₂		4.46%	
	d ₁₃		4.46%	
	d ₁₄		4.46%	
	d ₁₅		4.46%	
	d ₁₆		4.46%	
	d ₁₇		4.46%	
	d ₁₈		4.46%	
	d ₁₉		4.46%	
	d ₂₀		4.46%	
	d ₂₁		2.23%	

Supplementary Table 18. Financial Parameter Estimates

Parameter	Symbol	Unit	Nominal Escalation Rate			
			0%	1%	2%	3%
Fixed Charge Rate	FCR	fraction/year	0.059	0.067	0.076	0.085
Levelization Factor	LF	fraction	1.000	1.124	1.256	1.393
Discount Rate	DR	%	4.73%	5.78%	6.82%	7.87%

Additional aspects to be addressed:

Comment 1: Please include line numbers when submitting a paper for review.

Response: As suggested, line numbers have been added to the revised manuscript for review.

Comment 2: What is the U.S. Hydrogen Earthshot reported in the title and the abstract? The authors never explicitly define it. Please clarify it for the readers.

Response: We briefly defined it in the first paragraph of the introduction section. However, it looks not clear in terms of the reviewer’s feedback. To address this concern, we expanded the definition to read: “In 2021, the U.S. Department of Energy launched the Energy Earthshots Initiative that aims to accelerate breakthroughs of more abundant, affordable, and reliable clean energy solutions by 2030⁸. To catalyze technological innovation and scale in clean hydrogen, this initiative included a hydrogen shot that aims to decrease the cost of clean hydrogen to \$1 per 1 kilogram in 1 decade⁸, which is called the Hydrogen Energy Earthshot.” To be more consistent with the U.S. DOE’s definition, “Hydrogen Earthshot” was replaced by “Hydrogen Energy Earthshot” throughout the manuscript.

Comment 3: In the introduction, a quick comparison of blue hydrogen with green hydrogen should be provided. Also, I think some information on the actual emissions of current blue hydrogen projects should be illustrated, to inform the readers about the current levels of greenhouse gas emissions associated with this technology. Although they are already reported by the authors in Table 2 in the supplementary material, I think this is a crucial performance indicator of the technology. Also, a description of the contribution of different stages on the lifecycle emissions could be provided.

Response: We added to the first paragraph in the introduction section of the original manuscript a comparative statement: “Although renewable-powered green hydrogen has much less carbon emissions than blue hydrogen, the current cost of green hydrogen production can be several times higher^{6,7}, as shown later.” The newly cited reference is:

IRENA. *Making the breakthrough: Green hydrogen policies and technology costs*. ISBN 978-92-9260-314-4. (International Renewable Energy, 2021).

As in response to Comment 4, we added to Page 6 of the original version the quantitative emissions information of blue hydrogen projects: “In addition, the on-site stack CO₂ emissions from hydrogen production by SMR with CCS are 0.4 kg CO₂-eq/kg H₂, which is much less than that (1.4 kg CO₂-eq/kg H₂) from gasification with CCS⁷.”

As in response to a similar comment on life cycle emissions above, we expanded the statement in Page 11 of the original version to read: “As mentioned earlier, the 45V tax credit depends on the life cycle emissions of hydrogen production, which include greenhouse gas emissions from plant stacks, fuel supply, electric power supply, and CO₂ sequestration or management. The life cycle emissions were estimated by the National Energy Technology Laboratory to range from 3.1 to 8.9 kg CO₂-eq/kg H₂ for the gas-based blue hydrogen in the 90% confidence interval between the 5th and 95th percentile values and from 3.4 to 8.9 kg CO₂-eq/kg H₂ for the coal-based blue hydrogen, which is driven mainly by the uncertainty in fuel supply⁷. The largest contributor among the multiple stages to the life cycle emissions is the fuel supply⁷.”

Comment 4: Page 6: the authors compare the LCOH of SMR and gasification, but I think they should also mention the two different levels of GHG emissions. Gasification is not only more expensive than SMR, as authors correctly highlight, but it also leads to (1) a lower capture rate, (2) a higher net GHG emission intensity compared to SMR.

Response: As suggested, we added to Page 6 of the original version the quantitative emission information: “In addition, the on-site stack CO₂ emissions from hydrogen production by SMR with CCS are 0.4 kg CO₂-eq/kg H₂, which is much less than that (1.4 kg CO₂-eq/kg H₂) from gasification with CCS⁷.”

Comment 5: Page 6: When discussing the cost distribution in subsystems I think that the reference fuel cost for gas and coal should be provided, as their variability significantly affects the shares that are discussed. This information is of course provided by the authors together with the other parameters of the study, and also addressed later in the sensitivity analysis (at least for natural gas), but I think that including it here in the discussion may help the readers in better understanding the results that are described.

Response: As suggested, we added to Page 7 the fuel price information: “Given the gas price of \$4.2/GJ” and “Given the coal price of \$57.3/metric ton.”

Comment 6: The y scale of Figure 1 should report the 1 USD/kg that is the target aimed by the US policy.

Response: As suggested, the y-axis scale was revised to show the target cost of \$1/kg H₂.

Comment 7: Page 18, end of the page: it seems to me quite hard that advanced technologies can

contribute to the cost target by 2030, given the short time horizon. Please clarify this aspect in the discussion.

Response: We added to the paragraph further one more discussion: “However, it may be still challenging for advanced hydrogen technologies to reach the cost target by 2030.”

Comment 8: Page 19: Location of blue hydrogen hubs should also take into account the availability of CO₂ storage sites and related infrastructure. This is an important aspect that is not mentioned by the authors.

Response: Page 19 briefly mentioned this issue. To make this important issue clearer, we revised and expanded the discussion on Page 19 to read: “we revised the interpretation in the Discussion section to read: “Large-scale blue hydrogen production will consume multiple types of natural resources, such as fossil, water and land resources, and geological reservoirs and transportation infrastructure for CO₂ storage, and affect local management and planning of these natural resources. The availability and price of these natural resources vary by region or location in the country. Thus, siting blue hydrogen production plants should take into account co-location and co-availability of these natural resources.”

Comment 9: There are some typos in the manuscript (e.g. page 13, “nature gas”, page 17 “summery”, etc.).

Response: We fixed the typos.

Reviewer #2 (Remarks to the Author):

Comment: The presented manuscript "Technological evolution of large-scale blue hydrogen production toward the U.S. Hydrogen Earthshot" by W. Wu, H. Zhai, and E. Holubnyak studies the role that blue hydrogen will play in the decarbonisation of US energy system and in reaching the US Hydrogen Earthshot. The authors take data from a recent NETL study on the production cost of blue hydrogen and analyse future cost reductions through "learning-by-doing", where they assume exponential learning curves associated with increased technology deployment in coming years.

In summary, the manuscript investigates a highly relevant and timely topic of research. Yet, I see the need for substantial improvement to meet the standards expected by research articles published in Nature Communications.

Response: We agree with the reviewer on the timely importance of this study. We appreciate the reviewer for valuable comments. We made substantial revisions to address the comments to meet the expectations for publication in Nature Communications.

Comment: Here is a list of my main points of criticism:

- 1) The manuscript does not connect to the wider context of and societal/political/scientific debate about hydrogen and, specifically, blue hydrogen.
- 2) The manuscript has its findings based on a few key assumptions that are not sufficiently substantiated or discussed.
- 3) There is a lot of room for improvement for the presentation of figures and text.

Response: We appreciate the reviewer for the criticism, which has helped us significantly improve the manuscript to reach a new height of quality for publication in *Nature Communications*.

(1) To expand the introduction in a broader context, we added to the section a new paragraph:

“Clean hydrogen has the potential to help achieve 10% economy-wide emissions reductions by 2050 relative to 2005, promote energy security and resilience, and develop a new economy in the United States¹. In 2030, the hydrogen economy could create about 100,000 net new jobs for the development of new capital projects and clean hydrogen infrastructure². The U.S. Bipartisan Infrastructure Law has appropriated \$9.5 billion for clean hydrogen for the U.S. Department of Energy (DOE)¹. Both zero- and low-carbon hydrogen production technologies are key options in a diverse toolbox enabling the transition to a sustainable and equitable clean energy future¹. In October 2023, the U.S. DOE announced \$7 billion to launch seven Regional Clean Hydrogen Hubs across the nation³. Some regional hubs will use water and natural gas as the feedstock for renewable-powered electrolysis and steam methane reforming (SMR) with carbon capture and storage (CCS) to produce clean hydrogen, which are also called green and blue hydrogen in practice, respectively. Blue hydrogen is often viewed as a near-term bridge to a zero-carbon hydrogen economy. Given potential high methane leakage, however, there are scientific debates on the competitiveness of blue hydrogen^{4,5}, which make a serious call for methane abatement.”

The cited references include:

- DOE. *U.S. national clean hydrogen strategy and roadmap*. <https://www.hydrogen.energy.gov/docs/hydrogenprogramlibraries/pdfs/us-national-clean-hydrogen-strategy-roadmap.pdf> (U.S. Department of Energy, 2023).

- DOE. *Pathways to commercial liftoff: Clean hydrogen*. <https://liftoff.energy.gov/wp-content/uploads/2023/03/20230320-Liftoff-Clean-H2-vPUB.pdf> (U.S. Department of Energy, 2023).
- DOE. *Biden-Harris Administration Announces \$7 Billion For America's First Clean Hydrogen Hubs, Driving Clean Manufacturing and Delivering New Economic Opportunities Nationwide*. <https://www.energy.gov/articles/biden-harris-administration-announces-7-billion-americas-first-clean-hydrogen-hubs-driving> (U.S. Department of Energy, 2023)
- Howarth, R. W. & Jacobson, M. Z. How green is blue hydrogen? *Energy Sci. Eng.* **9**, 1676–1687 (2021).
- Romano, M. C., Antonini, C., Bardow, A., Bertsch, V., Brandon, N. P., Brouwer, J., Campanari, S., Crema, L., Dodds, P. E., Gardarsdottir, S., Gazzani, M., Gazzani, G. J., Lund, P. D., Dowell, N. M., Martelli, E., Mastropasqua, L., McKenna, R. C., Monteiro, J. G. M., Paltrinieri, N., Pollet, B. G., Reed, J. G., Schmidt, T. J., Vente, J. & Wiley, D. Comment on “How green is blue hydrogen?”. *Energy Sci. Eng.* **10**, 1944–1954 (2022).

(2) We have made significant improvements in the methodology and assumptions as the following specific comments suggested. In particular, please review our responses to Comments 8, 11, 16, and 19.

(3) We have used more specialized data analysis software to re-draw all the figures to improve the quality as suggested. Please review our response to Comment 14.

1) Wider context

Comment 1: For instance, the manuscript starts out by stating that hydrogen may potentially play a crucial role in the net-zero transition, but then focusses only on cost reduction throughout the entire rest of the paper without discussing emission savings at all. There exists a wider scientific debate about the role of natural gas in the net-zero transition (e.g. doi:10.1038/s41467-023-41105-z), about the impact of methane leakage (e.g. doi:10.1038/s41467-023-41527-9), and about how clean blue hydrogen actually is (doi:10.1002/ese3.956, doi:10.1002/ese3.1126, doi:10.1002/ese3.1154). I would expect a manuscript aiming to be published in Nature Communications to at least briefly touch upon those points and to present a balanced and nuanced picture of the role of blue hydrogen that goes beyond cost (even though I understand that this is the focus of the manuscript). I suggest the authors add at least a short discussion on the supply-chain CO₂ and CH₄ emissions of natural gas (and coal, although this is perhaps not as relevant) in the introduction and/or conclusions sections, perhaps also something on the risks of fossil lock-ins and the risks/opportunities of CCS.

Response: This comment raises a nice point in a broader context. To highlight this broader issue, we added to the end of the Discussion section a new paragraph with the suggested references:

“Last but not least, it is very important to prevent upstream methane emissions while reforming natural gas with CCS to produce blue hydrogen. Methane leakage along the natural gas supply chain can jeopardize the role of natural gas in the energy transition to a low-carbon or net-zero future, even when CCS is deployed^{43,44}. A high methane leakage rate of 3.5% or more can elevate the blue hydrogen’s carbon footprint and make it uncompetitive or even unviable in a hydrogen economy⁴. To reduce or avoid the risks of committing to high-emitting blue hydrogen, stringent standards and regulations should be imposed to limit methane leakage and promote deployment of best available technologies for methane abatement⁴⁴.”

The newly cited references are:

- Achakulwisut, P., Erickson, P., Guivarch, C., Schaeffer, R., Brutschin, E. & Pye, S. Global fossil fuel reduction pathways under different climate mitigation strategies and ambitions. *Nat. Commun.* 14, 5425 (2023).
- Shirizadeh, B., Villavicencio, M., Douguet, S., Trüby, J., Bou Issa, C., Seck, G. S., D’herbemont, V., Hache, E., Malbec, L., Sabathier, J., Venugopal, M., Lagrange, F., Saunier, S., Straus, J. & Reigstad, G. A. The impact of methane leakage on the role of natural gas in the European energy transition. *Nat. Commun.* 14, 5756 (2023).
- Howarth, R. W. & Jacobson, M. Z. How green is blue hydrogen? *Energy Sci. Eng.* 9, 1676–1687 (2021).

Comment 2: There is a brief mention of fossil fuel, water, and land resource demand, but with no clear interpretation or comparison.

Response: We tried to emphasize a couple of points that the production of blue hydrogen requires multiple types of natural resources. The large-scale deployment of blue hydrogen will affect local natural resources and the selection of sites to deploy large-scale blue hydrogen production plants should take into account co-location and availability of multiple nature resources. To make them clearer, we revised the interpretation in the Discussion section to read: “Large-scale blue hydrogen production will consume multiple types of natural resources, such as fossil, water and land resources, and geological reservoirs and transportation infrastructure for CO₂ storage, and affect local management and planning of these natural resources. The availability and price of these natural resources vary by region or location in the country. Thus, siting blue hydrogen production plants should take into account co-location and co-availability of these natural resources.”

Comment 3: Equally, natural gas and coal are discussed alongside each other purely based on a cost perspective, even though these two energy carriers are very dissimilar and also play crucially different roles in the net-zero transition.

Response: We tried to address this comment, though we were not sure if we fully captured the key point. In addition to the cost, currently announced blue hydrogen projects in the U.S. will heavily employ gas-based reformation technologies in terms of the International Energy Agency’s hydrogen project database (2023). Thus, we added to the first paragraph of the Sensitivity Analysis section an additional justification: “Blue hydrogen projects announced in the U.S. will mainly employ gas-based reformation technologies with CCS³¹.”

Comment 4: Next, I would expect to see the current project pipeline discussed. It would be very relevant to compare the 10MMTA number to announced blue-hydrogen projects in the US and to the share of announced projects with FID. Or could you at least estimate how many projects of what scale would need to be realised to reach that number?

Response: The number of projects is highly uncertain, which highly depends on the uncertain or variable capacity of specific production projects at a plant level. We think the discussion needs to focus on the total capacity instead of the number of specific projects. To address this comment and other similar comments, we added to the Discussion section a new paragraph:

“The global production capacity of low-carbon hydrogen will reach 12.3 MMTA by 2030 based on the announced, planning and committed projects³⁰. The low-carbon hydrogen capacity

in North America will reach 6.8 MMTA by 2030³⁰. However, only 1.8 MMTA³⁰ and 1.5 MMTA^{1,2} of the announced projects in North America and the U.S. have reached final investment decision (FID), mainly because many announced projects have not yet secured financing and nailed down contracted offtake^{1,2}. The hesitancy to long-term, scaled contracts is influenced by numerous factors, such as lack of price certainty, unavailability and reliability of large-scale hydrogen supply, near-term policy implementation uncertainty, and long-term political uncertainty^{1,2}.”

Comment 5a: Moreover, I would expect some mentioning of other colours of hydrogen, especially green hydrogen. Specifically, it'd be interesting to compare current cost estimates and projections of future cost reductions of green hydrogen (there are loads of studies on this) to those estimated in the presented study. Moreover, a short comparison of emission intensities would be suitable here.

Response: As in response to a similar comment from Reviewer 1 on the comparisons of current cost and emissions between blue and green hydrogen, we added to the end of Section Current Blue Hydrogen Production a new paragraph: “Currently, hydrogen is mainly produced by SMR without CCS in the U.S., which is often called grey hydrogen. Compared to it, the blue hydrogen production by SMR with CCS can decrease the stack CO₂ emission intensity by 96% but increase the LCOH by 55%⁷. The resulting CO₂ avoidance cost by blue hydrogen is \$65 per metric ton of CO₂. In contrast, the green hydrogen production by polymer electrolyte membrane electrolyzers almost has no stack CO₂ emissions but a high LCOH value ranging from \$3.0–7.5/kg H₂²⁷. The resulting CO₂ avoidance cost by green hydrogen relative to grey hydrogen varies from \$212–689 per metric ton of CO₂, which is much higher than that by blue hydrogen. Obviously, there are tradeoffs in CO₂ avoidance cost and emission savings between the blue and green production pathways. The details of emission and cost data and CO₂ avoidance cost estimation are available in Supplementary Note 2.”

We added to the Discussion section the future cost comparison between blue and green hydrogen: “The capex learning rates of green hydrogen production are 9% and 13% for alkaline electrolysis and polymer electrolyte membrane electrolysis, respectively^{35,36}, which are similar to those for SMR and PSA. Thus, the overall levelized cost of green hydrogen will likely be larger than that of blue hydrogen by 2030³⁷.”

The cited references are:

- Hydrogen Council. Path to hydrogen competitiveness a cost perspective. https://hydrogencouncil.com/wp-content/uploads/2020/01/Path-to-Hydrogen-Competitiveness_Full-Study-1.pdf (2020).
- IRENA. Green hydrogen cost reduction: scaling up electrolyzers to meet the 1.5 °C Climate Goal. ISBN 978-92-9260-295-6. (International Renewable Energy Agency, 2020).
- IRENA. Hydrogen: a renewable energy perspective. ISBN: 978-92-9260-151-5. (International Renewable Energy Agency, 2019).

We added a new supplementary section:

Supplementary Note 2: Cost of CO₂ Avoided by Blue and Green Hydrogen

The cost of CO₂ avoided by blue or green hydrogen relative to grey hydrogen can be estimated in terms of the production plant’s CO₂ emission intensity and levelized cost of hydrogen:

$$CCA = \frac{LCOH_{\text{blue/green}} - LCOH_{\text{grey}}}{EI_{\text{grey}} - EI_{\text{blue/green}}} \quad (S1)$$

Where CCA is the cost of CO₂ avoided (\$/kg CO₂); LCOH is the levelized cost of hydrogen (\$/kg H₂); EI is the stack CO₂ emission intensity of a production plant (kg CO₂/kg H₂); the subscript of “blue” represents the gas-based blue hydrogen production, the subscript of “green” represents the green hydrogen production, and the subscript of “grey” represents the gas-based grey hydrogen production without carbon capture.

Supplementary Table 5 summarizes the stack CO₂ emission intensity and LCOH of grey, blue, and green hydrogen production plants, which are collected from the literature^{1,4}. The resulting costs of CO₂ avoided by blue and green hydrogen relative to grey hydrogen are also provided in Supplementary Table 5.

Supplementary Table 5. Hydrogen Production Emission, Production Cost, and Carbon Avoidance Cost

Production Technology	Stack Emission Intensity (kg CO ₂ /kg H ₂)	Levelized Cost of Hydrogen (2018\$/kg H ₂)	Cost of CO ₂ Avoided (\$/metric ton CO ₂)
Steam Methane Reforming (SMR) (Grey)	9.35	1.06	Reference
SMR with CCS (Blue)	0.38	1.64	65
Distributed Polymer Electrolyte Membrane Electrolyzers (Green)	0 ^a	5.17 (3.0–7.5) ^b	440 (212-689)

^aZero carbon emissions are assumed for the electrolysis powered by solar photovoltaics as it produces almost no direct emissions during operation⁵.

^bThe cost was estimated for a production capacity of 1,500 kg H₂/day with an effective electricity price of 7.55 ¢/kWh⁴. The cost was further adjusted to 2018 dollars using the annual average consumer price index for all urban consumers⁶;

Comment 5b: Again, discussing project announcements and hence likely trends of installed capacity until 2030 would be relevant to compare.

Response: To address this comment and other similar comments, we added to the Discussion section a new paragraph:

“The global production capacity of low-carbon hydrogen will reach 12.3 MMTA by 2030 based on the announced, planning and committed projects³⁰. The low-carbon hydrogen capacity in North America will reach 6.8 MMTA by 2030³⁰. However, only 1.8 MMTA³⁰ and 1.5 MMTA^{1,2} of the announced projects in North America and the U.S. have reached final investment decision (FID), mainly because many announced projects have not yet secured financing and nailed down contracted offtake^{1,2}. The hesitancy to long-term, scaled contracts is influenced by numerous factors, such as lack of price certainty, unavailability and reliability of large-scale hydrogen supply, near-term policy implementation uncertainty, and long-term political uncertainty^{1,2}.”

Regarding the possible trends of installed capacity through 2030, please review the newly-added Fig. 2a and relevant discussion.

Comment 6: A big obstacle for hydrogen and large-scale industrial projects is still the lack of technological certainty. Many announced projects (see for instance the IEA hydrogen project database) still lack an FID. Perhaps the authors could discuss this in the US context and if/how their results could lay out a clearer plan for not only cost competitiveness but also for security of investment for blue hydrogen projects.

Response: To address this comment and other similar comments, we added to the Discussion section a new paragraph:

“The global production capacity of low-carbon hydrogen will reach 12.3 MMTA by 2030 based on the announced, planning and committed projects³⁰. The low-carbon hydrogen capacity in North America will reach 6.8 MMTA by 2030³⁰. However, only 1.8 MMTA³⁰ and 1.5 MMTA^{1,2} of the announced projects in North America and the U.S. have reached final investment decision (FID), mainly because many announced projects have not yet secured financing and nailed down contracted offtake^{1,2}. The hesitancy to long-term, scaled contracts is influenced by numerous factors, such as lack of price certainty, unavailability and reliability of large-scale hydrogen supply, near-term policy implementation uncertainty, and long-term political uncertainty^{1,2}. For blue hydrogen projects, enhancements in tax credits for carbon sequestration can improve the economics of hydrogen production. For example, an extension of the 45Q tax credit period from current 12 years to 18 years would significantly reduce the cumulative installed capacity required for gas-based blue hydrogen projects to reach the Hydrogen Energy Earthshot, as demonstrated in Supplementary Fig. 3. Extending the period of the 45Q tax credit for blue hydrogen projects can be considered an option to secure financing and promote long-term offtake.”

Supplementary Fig. 3. Effect of 45Q tax credit period on future cost of gas-based blue hydrogen production.

Comment 7: Finally, I would expect to see more depth added to the conclusions section and for a more explicit outline for competing strategies and policy options (e.g. further

subsidies/financial incentives; focussing on maturing one technology only; incentives to build several small vs few big plants; securing investments; speeding up planning procedures; etc).

Response: We expanded the relevant discussion by adding a new paragraph:

“Competing strategies and supportive policy and regulatory actions should be made rapidly on both the hydrogen demand and supply sides at both federal and state levels in alignment with the innovation expansion. A variety of high-level strategies are needed on the demand side to promote the widespread use of low-carbon hydrogen in industrial, transportation and power sectors and then establish large-sale markets for low-carbon hydrogen. To jumpstart a hydrogen economy, a cluster approach can be employed on the supply side to establish regional production-transportation-demand networks by co-locating feedstock supply, hydrogen production, and carbon sequestration with multiple end-users and by utilizing existing infrastructure, such as pipeline infrastructure for natural gas, CO₂, and H₂ transportation and geological reservoirs for CO₂ storage. To scale the regional hydrogen economy, secured investments in hydrogen production and supporting infrastructure are required with funding from both public and private sectors, plus subsidies and tax incentives. In addition, deploying large hydrogen production plants instead of small ones can improve engineering economics at a plant level. Given the important role of CCS in producing competitive blue hydrogen, continued support for large-scale demonstration projects should be boosted in the near term to reduce the CCS cost and its uncertainty. Investments in blue hydrogen should be prioritized to lock down sufficient financial resources for the most competitive technologies in the near term. Economic and policy incentives can be tailored with emphasis on gas-based blue hydrogen to catalyze its widespread deployment and technological evolution because of the pronounced cost advantage relative to coal-based blue hydrogen. Extending the 45Q tax credit from the current 12-year period to a longer period for gas-based blue hydrogen projects would remarkably lower the time-related cumulative installed capacity necessary to reach the Hydrogen Energy Earthshot.”

2) Key assumptions and validity of results

Comment 8: The key results of the manuscript crucially depend on the learning rates assumed for the cost reductions of the individual components. The authors assume these values from two primary sources (Rubin et al from 2007, and Schoots et al from 2008). These cited studies are now more than 15 years old and the data they refer to ranges from the 60s to the early 2000s. I wonder how reliable and valid the derived learning curves are for today's SMR and CO₂ capture technologies. I understand that these numbers are very tricky to derive with certainty, and one often has to resort to regressions of historical data. Yet, given that this is such a key assumption of this manuscript, I would expect a much more detailed analysis and discussion of these assumed parameters. My first impression is that a learning rate for SMR is somewhat high, given that this is a rather established technology.

Response: This study mainly cited the two pioneering learning curve studies by Rubin et al. (2007) and Schoots et al. (2008). Although a large number of technological learning studies have been done in the past years, almost no breakthroughs have been made on learning rates beyond the two pioneering studies in the past years. The learning rates from the two pioneering studies have been frequently adopted in the recent studies. In other words, there are no more rigorous estimates of learning rates in the literature than the two pioneering studies cited in our manuscript. To make it clearer, we added to the relevant paragraph in the introduction section on Page 5 of the original version new sentences with recent references:

“Although a large number of learning curve studies have been done in the past years, almost no breakthroughs have been made in estimating the learning rates for CCS, gasification, and SMR beyond the two pioneering studies by Rubin et al. (2007)¹⁴ and Schoots et al. (2008)¹³. Recent studies frequently adopted the learning rates from the two pioneering studies for a variety of applications^{17,18,19,20,21,22,23,24,25}.”

As given in Table 2, the base learning rate in operating and maintenance costs are zero for SMR based on the study by Schoots et al (2008). However, the capital cost still can benefit from learning by manufacturing and deploying massive reactors.

We further expanded the relevant discussions in the method section, which are highlighted in blue color: “To construct a learning curve for a technology with respect to its either total capital cost or total O&M cost, three types of model parameters have to be specified, including the initial cost, initial installed capacity, and learning rate. Capital and O&M learning rates can be estimated using empirical data for mature technologies or an analogous approach for advanced technologies. For example, the learning rates for SMR were derived from its historical installed capacity and cost data¹³, whereas the learning rates for post-combustion carbon capture were estimated by referring to those of post-combustion flue-gas desulfurization as they are technically analogous^{14,42}. The data collected for these parameters are discussed later.

A hydrogen production plant involves multiple technologies or subsystems, which have different values regarding the three parameters defining a learning curve. At a blue hydrogen production, individual subsystems lie at different levels of technological maturity. Learning rates and initial installed capacity count on maturity level and then vary by subsystem. For example, at a gas-based blue production plant, SMR and PSA are mature subsystems, whereas carbon capture has not been deployed widely, though it is commercially available. As a result, the O&M learning rates are zero for SMR and PSA but 22% for carbon capture. Thus, a component-based learning curve model is applied to estimate the total cost of hydrogen production at a certain level of cumulative installed capacity as the sum of individual subsystem costs.”

To clarify how an overall plant’s learning curve is constructed, we further expanded the discussion in the result section by adding to the first paragraph of Section Future Costs of Blue Hydrogen Production without and with Tax Credits new sentences: “A blue hydrogen production plant consists of numerous subsystems. However, the maturity status of individual subsystems and their initial installed capacity are different. As a result, learning rates and initial installed capacity vary by subsystem. Thus, a component-based learning curve model is employed to construct a plant-level learning curve based on individual subsystems’ learning rates and initial installed capacity.”

Comment 9: Also note that modern plant designs will have to ensure minimal CH₄ leakage (less than 0.1%), which will necessitate extra efforts and could thereby easily drive up SMR cost and hence offset other learning effects.

Response: A study by Alhamdani et al. (2017) reports that fugitive GHG emissions from an SMR production plant are minor and about 0.05% of the stack emissions, which indicates that on-site CH₄ leakage is not a serious concern. To address this point, we added to the “Current Blue Hydrogen Production” section on Page 6 the relevant results: “In addition to stack CO₂ emissions, there may be fugitive GHG emissions from various sources at an SMR production plant, mainly from the piping equipment and fittings²⁶. However, fugitive GHG emissions are

about 0.05% of the stack GHG emissions²⁶, which indicates that plant methane leakage is not a serious issue.”

The newly cited reference is:

- Alhamdani, Y. A., Hassim, M. H., Ng, R. T. & Hurme, M. The estimation of fugitive gas emissions from hydrogen production by natural gas steam reforming. *Int. J. Hydrogen Energy*, **42**, 9342-9351 (2017).

Comment 10: Moreover, it would be interesting if the manuscript was able to differentiate more clearly, how learning-by-doing could drive down the cost of those specific components and if/how R&D could support this effort.

Response: As the reviewer suggested, we further looked into the cost reduction from learning-by-doing and the breakdown of the cost reduction by component when the cumulative installed capacity reaches 10 MMTA. To present the relevant result and implication, we added to Page 12 of the original manuscript a new paragraph:

“Learning-by-doing will reduce the cost of hydrogen production for coal- and gas-based blue hydrogen. Fig. 1e shows the cost reduction by subsystem and by the 45Q tax credit when the cumulative installed capacity of blue hydrogen reaches 10 MMTA. For blue hydrogen produced from both coal and gas resources, the overall cost reduction will be driven largely by the carbon-sequestration tax credit and the improvement in carbon capture. In contrast, other subsystems, such as SMR and PSA, will make limited contributions because they are mature technologies and have no or limited reductions from additional 10 MMTA deployment in their future costs. These results indicate the importance of continued support from both public and private sectors for CCS-related research, development and demonstration programs at federal and state levels.”

Fig. 1 Initial and future costs of blue hydrogen production without and with tax incentives. (e) Future cost reductions for coal-based and gas-based H₂ production with tax incentives.

Comment 11: The article focusses on SMR as the main technology for blue hydrogen. However, at least to my understanding, the most suitable technology for blue hydrogen with high capture rates would be ATR. Can the authors comment on why they decided to choose SMR instead of ATR?

Response: The NETL study (2022) reports that for blue hydrogen production, ATR with CCS has stack emissions of 0.51 kg CO₂/kg H₂, life cycle emissions of 5.72 kg CO₂-eq/kg H₂, and the levelized production cost of \$1.59/kg H₂. In contrast, SMR with CCS has stack emissions of 0.38 kg CO₂/kg H₂, life cycle emissions of 4.57 kg CO₂-eq/kg H₂, and the levelized production cost of \$1.64/kg H₂. To clarify why our study focuses on SMR, we added to the first section of the results section on Page 6 of the original version: “The majority of hydrogen produced in the U.S. is made via SMR. In addition, the cost of blue hydrogen produced by SMR with CCS is similar to that by autothermal reforming with CCS but the on-site and life-cycle emissions from the SMR process are less⁷. This study therefore focuses on SMR with CCS for blue hydrogen production.”

3) Presentation of figures and text

Comment 12: As mentioned above, the whole article would greatly benefit from more discussion of context and connecting to the wider scientific debate. Specifically, the introduction and conclusions sections require more contextualisation.

Response: As suggested, we have made significant improvements to expand the discussion of context in a broader context, especially in the introduction and discussion sections. In particular, please review our responses to the high-level comment above and Comments 1, 2, 4, 5a and 5b, 6, 7, 17, and 18.

Comment 13: I don't know how this article is doing on the word-count limit, but I suspect it's rather tight. I would suggest moving a lot more of the technical discussions from the main text (e.g. "Current blue hydrogen production") into the Methods section and really only discuss the results and the key assumptions (learning rates). This would give more space for discussing context rather than stating methodological details or points that the reader can read off from the figures.

Response: We cannot fully agree with this comment. We justified the adoption of learning rates and added an array of new analyses, results and discussions to address the valuable comments from the reviewer. However, we cannot move the “Current blue hydrogen production” section into the Methods as this section presents major assumptions and current production technologies’ performance and cost, such CO₂ emissions and sequestration and levelized cost of hydrogen production, including the cost distribution by subsystem, which set up initial points to formulate learning curves. Without these results, we cannot create learning curves and outline the rationale for a series of parametric analyses. In other words, we have to first quantify and characterize the current hydrogen production and then present the learning curves for hydrogen production in the future, including its variability by key factors.

Comment 14: The figures convey little information and don't allow the reader to easily grasp the main results of the manuscript at one glance. I also can't help but noticing that the figures were created in Excel. Often (not always) I find that this does not allow to create advanced plots that can densely present a lot of information in a yet elegant and comprehensible way. For a

publication in Nature Communications, I would expect the authors to work hard to create an iconic figure capturing the key results and conveying these along with an accompanying message/story in a very comprehensible manner. E.g., I would suggest to have one main figure consisting of 1) the pie chart (which is good!), 2) a stacked-bar waterfall diagram for each technology demonstrating the different components of cost savings at 10 MMTA installed capacity and with the tax credits, and 3) the learning curves of the main assumptions without the point markers (or do they serve a purpose?) combined with shaded uncertainty corridors indicating the results of the sensitivity analysis (only gas price OR learning rate OR perhaps both combined).

Response: We partially agree with this comment. The figures were created in Excel but delivered the main information or results of this manuscript. We appreciate the reviewer for pushing us to present the figures in a higher quality. To improve the quality, we used data analysis software to re-draw all the figures in the manuscript.

As suggested, we kept the pie chart and added a stacked-bar waterfall diagram as Fig. 1e. Furthermore, we added shared uncertainty ranges to the learning curves, combined the relevant figures accordingly, and added a heat map on learning rates. Point markers were added accordingly. Below are the re-drawn ones of the three figures presented in the original manuscript. In addition, we revised the relevant statements in line with the representation of all the updated figures.

Fig. 1 Initial and future costs of blue hydrogen production without and with tax incentives. (a) Distribution of initial levelized cost for gas-based H₂ production; (b) Distribution of initial levelized cost for coal-based H₂ production; (c) Learning curves for coal-based and gas-based H₂ production capital and O&M costs without tax incentives; (d) Learning curves for overall levelized cost of coal-based and gas-based H₂ production without and with tax incentives; (e) Future cost reductions for coal-based and gas-based H₂ production with tax incentives.

Fig. 3 Effect of natural gas price on future levelized cost of gas-based blue hydrogen production.

Fig. 5 Sensitivity of LCOH by SMR with CCS to learning rates (LR). (a) LCOH under two boundary scenarios of learning rates; (b) LCOH under the range of 100%–150% time base learning rates, except for O&M cost learning rates, which are equal to 5%–10% for SMR, PSA, and CO₂ compression. Note to Fig. 5b: P₁ means a percentage relative to the base learning rate, whereas P₂ means the learning rate on an absolute basis.

Comment 15: In summary, the authors investigate a very relevant and topical research question, but the way in which their manuscript is currently written, it doesn't match the portfolio of Nature Communications and would be more suitable for a more field-specific technical journal. Irrespective of whether the authors choose to resubmit to NComms, I would encourage them to substantially revise their paper by: a) adding more context, b) pointing out key take-aways more clearly, c) improving the graphical presentation of their results, and d) more thoroughly discussing/substantiating the key assumptions of their work (mainly the learning rate).

Response: We have made significant revisions to address all the comments. In particular, (a) we added new analyses, including comparative analysis between blue and green hydrogen (Response to Comment 5a), announced and FID projects (Response to Comment 4, Response to

Comment 5a, and Response to Comment 6), diffusion-of-innovation analysis (Response to Comment 17), and inflation analysis (Response to Comment 21); (b) we improved the clarifications on some key take-home message, such as Response to Comment 10 and Response to Comment 18; (c) we have re-drawn all the figures in the initial submission to improve the quality and added two new figures to the revised manuscript. Please review Response to Comment 14, Response to Comment 17, and Response to Comment 18; (d) we expanded the statements, justifications, and clarifications to thoroughly present the assumptions made in this study. In particular, please review our responses to Comments 8, 9, 11, 16, and 19.

Some further points/comments/questions/remarks:

Comment 16: What is your assumption for the interest rate/WACC? (I can't remember reading it and couldn't find it with Ctrl+F.) I wonder if the assumed WACC could also be a basis for another sensitivity analysis.

Response: It turns out that we should substantially improve the presentation of the costing method with more details. We revised the methodological description of cost metric in line with the NETL's costing method (2022). Fixed charge rate is a key parameter used to estimate the overall production cost, which is affected by the after-tax WACC. Both after-tax WACC and FCR are affected by inflation. Thus, an additional sensitivity analysis was performed to evaluate the economic effect of inflation on the LCOH in nominal dollars. Additional methodological details about how to estimate the LCOH in nominal dollars are available in the supplementary information.

“Cost metric for technology evaluation

The component-based learning curve model is applied to project the future total capital cost and total O&M cost of individual subsystems and an overall plant as a function of cumulative installed capacity and then estimate the overall cost of hydrogen production for a given cumulative installed capacity. The cost metric considered for technology evaluation is the levelized cost of hydrogen and is estimated in real dollars^{7,34}.

$$LCOH_R = LCC_R + LOM_R + LFP_R \quad (2)$$

$$LCC_R = \frac{TASC_R \cdot FCR_R}{(CF \cdot AH) \cdot KG_{H_2}} \quad (3)$$

$$LOM_R = \frac{OM_R}{(CF \cdot AH) \cdot KG_{H_2}} \quad (4)$$

$$LFP_R = \frac{FC_R \cdot FR}{KG_{H_2}} \quad (5)$$

Where the subscript “R” means the real dollars; $LCOH_R$ is the levelized cost of hydrogen of a blue hydrogen production plant (\$/kg H₂); LCC_R is the levelized capital cost (\$/kg H₂); LOM_R is the non-fuel levelized operating and maintenance cost (\$/kg H₂); LFP_R is the levelized fuel price (\$/kg H₂); $TASC_R$ is the total as-spent capital of a blue hydrogen production plant (\$); FCR_R is the fixed charge rate (fraction/year); CF is the plant capacity factor (%); AH is the total annual hours (8760 hours); KG_{H_2} is the hourly production rate (kg H₂/hour); OM_R is the total non-fuel operating and maintenance (O&M) cost (\$/year), including both the fixed and non-fuel variable O&M costs; FC_R is the natural gas cost (\$/GJ) or coal cost (\$/metric ton); FR is the hourly natural gas flow rate (GJ/hour) or coal flow rate (metric ton/hour). When estimating the LCOH

in real dollars, the FCR has to be determined using Equations (6) to (9), which varies with the project book lifetime and a panel of financial variables³⁴:

$$FCR_R = \frac{CRF_R^{\text{nonfuel}} - ETR \cdot PV_{\text{plant}}}{1 - ETR} \quad (6)$$

$$CRF_R^{\text{nonfuel}} = \frac{ATWACC_R \cdot (1 + ATWACC_R)^{BL}}{(1 + ATWACC_R)^{BL} - 1} \quad (7)$$

$$PV_{\text{plant}} = CRF_R^{\text{nonfuel}} \cdot \sum_{n=1}^m \frac{d_n}{(1 + ATWACC_R)^n} \quad (8)$$

$$ATWACC_R = PC_{\text{equity}} \cdot ROE_R + PC_{\text{debt}} \cdot kd_R \cdot (1 - ETR) \quad (9)$$

Where the superscript “nonfuel” represents the non-fuel component; CRF_R is the capital recovery factor (fraction/year); ETR is the effective tax rate (%); PV_{plant} is the present value of tax depreciation expense of a blue hydrogen project (fraction/year); $ATWACC_R$ is the after-tax weighted average cost of capital (%); BL is the project book lifetime (year); d_n is the tax depreciation fraction in a year (n) (fraction); m is the number of years of depreciation (year); PC_{equity} is the percent of equity (%); PC_{debt} is the percent of debt (%); ROE_R is the real rate of return on equity (%); kd_R is the real rate of cost of debt (%). The financial parameters and their data sources are detailed in Supplementary Table 17 and the resulting FCR is available in Supplementary Table 18.

Inflation affects discount rate, ATWACC, FCR, and other factors. To evaluate the effect of inflation on the cost of blue hydrogen production, the LCOH is estimated in nominal dollars. The detailed estimation of the LCOH in nominal dollars is reported in Supplementary Note 8.”

To explain how to estimate the LCOH in nominal dollars, we added a new supplementary session:

Supplementary Note 8: Hydrogen Production Cost Estimation in Nominal Dollars

The levelized cost of hydrogen and is estimated in nominal dollars¹:

$$LCOH_N = LCC_N + LOM_N + LFP_N \quad (S12)$$

$$LCOH_N^{\text{TC}} = LCC_N + LOM_N + LFP_N - LTC_N^{45Q} \quad (S13)$$

Where the subscript “N” means the nominal dollars; $LCOH_N$ is the levelized cost of hydrogen (\$/kg H₂); LCC_N is the levelized capital cost (\$/kg H₂); LOM_N is the levelized operating and maintenance cost (\$/kg H₂); LFP_N is the levelized fuel price (\$/kg H₂); $LCOH_N^{\text{TC}}$ is the levelized cost of hydrogen with a tax credit (\$/kg H₂); LTC_N^{45Q} is the levelized tax credit over the project book lifetime of a blue hydrogen project (\$/kg H₂).

The levelized capital cost in nominal dollars is estimated using Supplementary Equation (14), while the fixed charge rate is estimated using Supplementary Equations (14) to (21)^{2,3}:

$$LCC_N = \frac{TASC_R \cdot FCR_N}{(CF \cdot AH) \cdot KG_{H_2}} \quad (S14)$$

$$FCR_N = \frac{CRF_N^{\text{nonfuel}} - ETR \cdot PV_{\text{plant}}}{1 - ETR} \quad (S15)$$

$$CRF_N^{\text{nonfuel}} = \frac{ATWACC_N \cdot (1 + ATWACC_N)^{BL}}{(1 + ATWACC_N)^{BL} - 1} \quad (S16)$$

$$PV_{\text{plant}} = CRF_N^{\text{nonfuel}} \cdot \sum_{n=1}^m \frac{d_n}{(1 + ATWACC_N)^n} \quad (\text{S17})$$

$$ATWACC_N = PC_{\text{equity}} \cdot ROE_N + PC_{\text{debt}} \cdot kd_N \cdot (1 - ETR) \quad (\text{S18})$$

$$ROE_N = (1 + ROE_R) \cdot (1 + N) - 1 \quad (\text{S19})$$

$$kd_N = (1 + kd_R) \cdot (1 + N) - 1 \quad (\text{S20})$$

$$N = R + I \quad (\text{S21})$$

Where the superscript “nonfuel” represents the non-fuel component; $TASC_R$ is the total as-spent capital of a blue hydrogen production plant (\$); FCR_N is the fixed charge rate (fraction/year); CF is the plant capacity factor (%); AH is the total annual hours (8760 hours); KG_{H_2} is the hourly production rate (kg H_2 /hour); CRF_N is the capital recovery factor (fraction/year); ETR is the effective tax rate (%); PV_{plant} is the present value of tax depreciation expense (fraction/year); $ATWACC_N$ is the after-tax weighted average cost of capital (%); BL is the project book lifetime (year); d_n is the tax depreciation fraction in year n (fraction); m is the number of years of depreciation (year); PC_{equity} is the percent of equity (%); ROE_N is the nominal rate of return on equity (%); PC_{debt} is the percent of debt (%); kd_N is the nominal rate of cost of debt; ROE_R is the real rate of return on equity (%); kd_R is the real rate of cost of debt (%); N is the nominal escalation rate (%); R is the real escalation rate (%); and I is the inflation rate (%).

The levelized operating and maintenance cost in nominal dollars is estimated using Supplementary Equation (22), while the levelization factor is estimated using Supplementary Equation (23)³:

$$LOM_N = \frac{OM_R \cdot LF_N}{(CF \cdot AH) \cdot KG_{H_2}} \quad (\text{S22})$$

$$LF = CRF_N^{\text{nonfuel}} \cdot \frac{1 - \left[\frac{1 + N}{1 + ATWACC_N} \right]^{BL}}{ATWACC_N - N} \quad (\text{S23})$$

Where OM_R is the non-fuel operating and maintenance (O&M) cost of a blue hydrogen production plant (\$/year); and LF_N is the levelization factor for non-fuel O&M cost (unitless).

The levelized fuel price in nominal dollars is estimated using Supplementary Equation (24), where the fuel’s capital recovery factor is estimated using Supplementary Equation (26) and the discount rate in nominal dollars is estimated using Supplementary Equation (27)¹⁶:

$$LFP_N = \frac{FC_N \cdot FR}{KG_{H_2}} \quad (\text{S25})$$

$$FC_N = PV_{\text{fuel}} \cdot CRF_N^{\text{fuel}} \quad (\text{S25})$$

$$CRF_N^{\text{fuel}} = \frac{DR_N \cdot (1 + DR_N)^{BL}}{(1 + DR_N)^{BL} - 1} \quad (\text{S26})$$

$$DR_N = (1 + DR_R) \cdot (1 + N) - 1 \quad (\text{S27})$$

$$PV_{\text{fuel}} = \sum_{t=1}^{BL} \frac{P_t}{(1 + DR_N)^t} \quad (\text{S28})$$

$$P_t = P_{t-1} \cdot (1 + N) \quad (\text{S29})$$

Where the superscript “fuel” represent the fuel component; FC_N is the natural gas cost (\$/GJ); FR is the hourly natural gas flow rate (GJ/hour); PV_{fuel} is the present value of gas cost (\$/GJ);

CRF_N^{fuel} is the fuel’s capital recovery factor (fraction/year); DR_N is the nominal discount rate for fuel cost (%); DR_R is the real discount rate for fuel cost (%); N is the nominal escalation rate (%); P_t is the fuel cost in year t (\$/GJ), while P_1 is the real fuel cost in the first year of a project.

In this study, the financial structure and assumptions are in alignment with those of the studies by the National Energy Technology Laboratory (NETL)^{1,2,3,16,17}. Supplementary Table 17 lists the financial parameters and summarizes their data and the sources of data. Supplementary Table 18 summarizes the estimates of FCR, levelization factor, and discount rate as a function of nominal escalation rate.

Supplementary Table 17. Financial Parameters and Assumptions

Parameter	Symbol	Unit	Value	Source(s)
Effective Tax Rate	ETR	%	25.74%	NETL (2021) ³
Inflation Rate	I	%	0–3%	Assumption
Real Escalation Rate	R	%	0%	NETL (2011) ²
Percent of Equity	PC_{equity}	%	62%	NETL (2022) ¹
Percent of Debt	PC_{debt}	%	38%	NETL (2022) ¹
Real Rate of Return on Equity	ROE_R	%	3.10%	NETL (2022) ¹
Real Rate of Cost of Debt	kd_R	%	5.15%	NETL (2022) ¹
Real Discount Rate	DR_R	%	4.72%	NETL (2019) ¹⁶
Project Book Lifetime	BL	year	30	NETL (2022) ¹
Fuel Cost	P_1	2018\$/GJ	4.2	NETL (2022) ¹
Tax Depreciation Fraction	d_1	%	3.75%	NETL (2021) ³ ; IRS (2016) ¹⁷
	d_2		7.22%	
	d_3		6.68%	
	d_4		6.18%	
	d_5		5.71%	
	d_6		5.29%	
	d_7		4.89%	
	d_8		4.52%	
	d_9		4.46%	
	d_{10}		4.46%	
	d_{11}		4.46%	
	d_{12}		4.46%	
	d_{13}		4.46%	
	d_{14}		4.46%	
	d_{15}		4.46%	
	d_{16}		4.46%	
	d_{17}		4.46%	
	d_{18}		4.46%	
	d_{19}		4.46%	
	d_{20}		4.46%	
	d_{21}		2.23%	

Supplementary Table 18. Financial Parameter Estimates

Parameter	Symbol	Unit	Nominal Escalation Rate			
			0%	1%	2%	3%
Fixed Charge Rate	FCR	fraction/year	0.059	0.067	0.076	0.085
Levelization Factor	LF	fraction	1.000	1.124	1.256	1.393
Discount Rate	DR	%	4.73%	5.78%	6.82%	7.87%

Comment 17: My understanding is you are currently deducting the tax credit from the levelised cost, i.e. thereby spreading the credit across the assumed book lifetime. Perhaps it would make sense to rather discuss in more detail what will happen when the tax credit ceases? I.e. the later that projects start, the less of the credit period they will be able to obtain. Your current main figures have product capacity on the lower axis. If you assume a certain deployment rate, you could plot these figures as a function of time. The resulting figure would allow to see the trade-off between waiting till the production cost drops lower via learning vs. less tax credit (I assume the tax credit would still be the much bigger effect, hence indicating it'd be more profitable to construct now rather than later).

Response: Tax credit is available for a certain period for blue hydrogen instead of the entire book lifetime. As discussed in the manuscript, the period is 10 years for the 45 V tax credit and 12 years for the 45Q tax credit. In addition, the regulations require that to claim either the 45Q tax credit or the 45V tax credit, facilities must be placed in service before January 1, 2033. For such facilities (in service before 1/1/2033), the period of credit availability is in common, regardless of their start-of-service time. The suggested analysis looks not suitable for the tax scheme that our study examined. To make the tax scheme clearer, we added to the “Blue hydrogen production with tax credits” section new sentences: “To claim either the 45Q tax credit or the 45V tax credit, facilities must be placed in service before January 1st, 2033²⁹. The credit is available for such qualified facilities for a period. The period of credit availability is in common to eligible facilities, regardless of their start-of-service time.”

Actually, this comment raises a valuable concept regarding how the technology adoption diffuses over time. Thus, we developed a diffusion-of-innovation model and applied it to estimate the installed capacity and the corresponding cost of hydrogen production as a function time, which can collectively determine how much the cumulative installed capacity would be deployed at what cost of hydrogen production in any particular year in the future. We added the newly-developed diffusion-of-innovation model to the Methods section:

“Diffusion-of-Innovation Model

The diffusion of innovation describes how a new technology would spread over time. An S-shaped curve is often used to measure the diffusion over time, in which the adoption rate increases during the early stage, reaches a maximum level at the point of inflection, and decreases until the diffusion curve saturates⁴⁵. To estimate the annual installed capacity of low-carbon hydrogen over time, the S-shaped diffusion function is employed^{46,47}:

$$cc_t = \frac{cc_{sat}}{1 + \frac{cc_{sat} - cc_0}{cc_0} \cdot e^{-r \cdot t}} \quad (1)$$

Where cc_t is the annual installed capacity of low-carbon hydrogen in a particular year t (million metric tons per annum, MMTA); cc_{sat} is the saturation level of annual installed capacity (MMTA); cc_0 is the initial annual installed capacity (MMTA) in the start year; r is the growth rate (fraction); and t is a particular year after the start period. The function coefficients are estimated by regression based on current and future low-carbon hydrogen production capacities through 2030^{12,28,30}. Additional details about the regression and diffusion function are available in Supplementary Note 7. Once the annual installed capacity in future years is determined, the cumulative annual installed capacity can be estimated as a function of time.”

We added the relevant results to the Results section:

“Time-based diffusion of blue hydrogen production

It is helpful for hydrogen energy planning to explore if certain production capacity and cost targets can be achieved by 2030. A new study reports the cumulative installed capacity of low-carbon hydrogen production over time based on globally announced, planning and committed projects through 2030³⁰. A diffusion-of-innovation model was established based on the current and future low-carbon hydrogen capacities through 2030 to explore the time-based diffusion of gas-based blue hydrogen over a long-term planning horizon through 2050.

Fig. 2a shows the cumulative installed capacity estimates for global low-carbon hydrogen production over time. The gas-based blue hydrogen capacity accounts for 49% of the total low-carbon hydrogen capacity given in Table 1 and is estimated to be 90% in 2030 in terms of the International Energy Agency’s hydrogen project databases^{28,31}. Given the changing shares over time, Fig. 2a also shows a range of cumulative installed capacity for gas-based blue hydrogen in a particular year. The cumulative installed capacity of the global gas-based blue hydrogen may range from 6 to 12 MMTA in 2030, which implies that it would be hard for the blue hydrogen production by SMR with CCS alone in the U.S. to reach 10 MMTA in 2030.

Fig. 1c shows the overall plant LCOH as a function of cumulative installed capacity for gas-based blue hydrogen, whereas Fig. 2a show the cumulative installed capacity over time. Combining them together, Fig. 2b shows the overall plant LCOH of gas-based blue hydrogen production without tax credit over time. The result shown in Fig. 2b implies that for the fuel price and learning rates given in the base case, it would also be difficult for gas-based blue hydrogen to reach the ambitious cost target of \$1/kg H₂ by 2030 in normal scenarios without aggressive incentives and game-changing technologies.

Fig. 2 Diffusion of cumulative installed capacity of low-carbon hydrogen and time-based learning curves of gas-based blue hydrogen. (a) Diffusion of cumulative installed capacity; (b) Time-based learning curves of blue hydrogen production cost.

Comment 18: Two main takeaways for me from the manuscript are: a) substantial cost reductions via "learning-by-doing" can only be achieved if technologies are sufficiently deployed, and b) coal-based blue hydrogen seems to be uncompetitive compared to gas-based. Wouldn't it therefore be sensible to recommend focussing policy support/industry efforts on gas-based blue hydrogen to reach sufficient deployment and hence cost reductions of this one technology instead of splitting up efforts between the two options?

Response: As in response to a similar comment above, we added to the discussion section this point: “Investments in blue hydrogen should be prioritized to lock down sufficient financial resources for the most competitive technologies in the near term. Economic and policy incentives can be tailored with emphasis on gas-based blue hydrogen to catalyze its widespread deployment and technological evolution because of the pronounced cost advantage relative to coal-based blue hydrogen. Extending the 45Q tax credit from the current 12-year period to a longer period for gas-based blue hydrogen projects would remarkably lower the time-related cumulative installed capacity necessary to reach the Hydrogen Energy Earthshot.”

Comment 19: Eq. (6) looks odd. Shouldn't the term proportional to the VOM be independent of the CF? I.e. there is no cost for fuel consumption during the period when the plant is not producing.

Response: If the VOM is reported on an absolute basis (\$/yr), it is dependent on the CF. If the VOM is reported in a normalized basis, it is independent of the CF. There is no cost for fuel consumption when the plant is not producing hydrogen. As in response to a comment on the costing method above, we revised the equations in line with the NETL’s costing method, in which the VOM is reported on an absolute basis and the fuel cost is reported separately as the third category.

Comment 20: Irrespective of what you choose to do with your figures, could you make sure the x-axis ticks are the same across all of them? E.g. it's hard to find the 1\$/kg marker. Perhaps you could also add a dashed horizontal line to that graph?

Response: As suggested, we made the x-axis ticks consistent with all the figures, except for a new heat map, which is not suitable for this change. We also added a dashed horizontal line to all the relevant figures to indicate the “1 \$/kg H₂” target.

Comment 21: Given you're trying to compare cost reductions against the 1\$/kg target, it'd be valuable to discuss the role of inflation. I noticed you're using \$2018 numbers from the NETL study. It's almost a bit of a silly question because I presume the 1\$/kg target is more of a political slogan than the result of an elaborate economic analysis, but I wonder how much inflation today and in coming years would make it harder to reach that specific nominal value.

Response: As the reviewer mentioned, the \$1/kg target is the U.S. DOE’s initiative goal, which has no specific information regarding cost type (real vs. nominal dollars) and inflation. This study reports cost results in real dollars. As in response to a similar comment from Reviewer 1, we conducted an additional parametric analysis and estimated the cost results in nominal dollars to demonstrate the effect of inflation rate on the future cost of hydrogen production toward the Hydrogen Energy Earthshot. The results imply that inflation would remarkably raise challenges for blue hydrogen production to reach the cost target. We added to the Sensitivity Analysis section the new analysis and relevant results:

“Inflation Rate

In general, this study estimates the cost of hydrogen production in real dollars. When the cost is estimated in nominal dollars, however, both the initial and future LCOH estimates vary with inflation rate as it affects discount rate, fixed charge rate, and levelization factor. A parametric analysis was further performed for inflation rate to quantify its effect on the evolving cost of gas-based blue hydrogen production toward the Hydrogen Energy Earthshot. Fig. 6 shows the learning curves of blue hydrogen production with inflation. Fig. 6a and 6b show that at

a given level of cumulative installed capacity, the LCOH in nominal dollars increases when the inflation rate increases from 1% to 3%. As a result, blue hydrogen production may not reach the cost target of \$1/kg H₂ for both the scenarios without and with 45Q tax credit even when the cumulative installed capacity reaches 30 MMTA. Fig. 6c further shows that with an inflation rate of 3%, the future LCOH may get close to the cost target when cheap natural gas resources are used as the feedstock to produce blue hydrogen with the cumulative installed capacity of up to 30 MMTA.

Fig. 6a and Fig. 6b also compare the learning curves of blue hydrogen production between the two scenarios without and with inflation. As shown in Fig. 6a for the scenario without 45Q tax credit, the reduction in hydrogen production cost from deploying the cumulative installed capacity of 10 MMTA can be offset by an inflation rate of 1%. There is a similar result for the scenario with 45Q tax credit, as shown in Fig. 6b. All these results imply that inflation would remarkably raise challenges for blue hydrogen production to reach the Hydrogen Energy Earthshot in the near future.”

Fig. 6 Effect of inflation rate on future cost of gas-based blue hydrogen production without and with 45Q tax credit. (a) Levelized cost of hydrogen production with gas price of \$4.2/GJ and without 45Q tax credit; (b) Levelized cost of hydrogen production with gas price of \$4.2/GJ and 45Q tax credit; (c) Levelized cost of hydrogen production with 3% inflation rate and 45Q tax credit.

Comment 22: Can you say more explicitly what is meant by “constant dollars”?

Response: It means real dollars. The two terminologies are often used in practice. To avoid unnecessary confusing, we changed “constant dollars” to “real dollars” unless otherwise noted in the manuscript.

Comment 23: Typo: summery

Response: We fixed the typo.

Reviewer #3 (Remarks to the Author):

Comment: This paper looks at the projected cost of blue hydrogen as a function of different learning rates, taking into account the IRA tax credits for hydrogen. It relies on a recent comprehensive report by NETL, which provides process level comparisons of state-of-the-art, fossil-based hydrogen production technologies. The authors use their analysis to assess the likelihood of reaching the US Hydrogen Earthshot goal of hydrogen at 1 \$/kg by 2030. Given the importance of hydrogen in future global energy systems, and the uncertainties around technology costs, understanding the cost evolution of hydrogen technologies is a useful and timely addition to the literature. The paper focusses heavily on learning rates and gas prices. However, the novelty of the analysis and depth of the discussion could be improved.

Response: We have an agreement with the reviewer on the timely importance of this study. We appreciate the reviewer for valuable comments. We made substantial revisions to address the comments.

Comment: Firstly, the assumptions for learning rate seem simplistic and lack a discussion/justification of how they apply to an industry that is looking to evolve a mature technology: SMR is a commercially mature process, but coupling it with CCS is not. The discussion of learning rate discovery is limited to two sentences in the methods section which simply states that ‘learning rates are mainly collected from two highly-cited learning curve articles’. In particular, the authors mention the option of retro fitting existing SMR plants briefly in the discussion, but do not include this important pathway in their analysis. The manuscript needs a more in-depth discussion of how learning rates can be applied and developed to take the maturity of certain sub-systems, and the options for retro fitting into account. I note that the authors do seem to apply different learning rates to different subsystems, but do not explain if/how they considered the varying maturity of the sub-systems.

Response: Both “Method Section” and “Section Future costs of blue hydrogen without and with tax credits” on Page 7 discuss how to apply learning rates in capital and O&M costs of individual subsystem in a blue hydrogen production plant to develop an overall plant learning curve. However, we did not explain them sufficiently in terms of this feedback. As the reviewer suggested, we expanded the discussions regarding learning rate discovery and maturity issue.

We expanded the relevant discussions in the method section, which are highlighted in blue color: “To construct a learning curve for a technology with respect to its either total capital cost or total O&M cost, three types of model parameters have to be specified, including the initial cost, initial installed capacity, and learning rate. Capital and O&M learning rates can be estimated using empirical data for mature technologies or an analogous approach for advanced technologies. For example, the learning rates for SMR were derived from its historical installed capacity and cost data¹³, whereas the learning rates for post-combustion carbon capture were estimated by referring to those of post-combustion flue-gas desulfurization as they are technically analogous^{14,42}. The data collected for these parameters are discussed later.

A hydrogen production plant involves multiple technologies or subsystems, which have different values regarding the three parameters defining a learning curve. At a blue hydrogen production, individual subsystems lie at different levels of technological maturity. Learning rates and initial installed capacity count on maturity level and then vary by subsystem. For example, at a gas-based blue production plant, SMR and PSA are mature subsystems, whereas carbon

capture has not been deployed widely, though it is commercially available. As a result, the O&M learning rates are zero for SMR and PSA but 22% for carbon capture. Thus, a component-based learning curve model is applied to estimate the total cost of hydrogen production at a certain level of cumulative installed capacity as the sum of individual subsystem costs.”

To clarify how an overall plant’s learning curve is constructed, we further expanded the discussion in the result section by adding to the first paragraph of Section Future Costs of Blue Hydrogen Production without and with Tax Credits new sentences: “A blue hydrogen production plant consists of numerous subsystems. However, the maturity status of individual subsystems and their initial installed capacity are different. As a result, learning rates and initial installed capacity vary by subsystem. Thus, a component-based learning curve model is employed to construct a plant-level learning curve based on individual subsystems’ learning rates and initial installed capacity.”

Comment: Secondly, they identify gas prices as a critical parameter, but do not take the opportunity to discuss how a large blue hydrogen industry could distort or affect the gas markets in the US – essentially providing a large capacity market for US gas as its usage in other sectors declines. Given that they are calculating large cumulative hydrogen production capacities, this would seem to be a very salient point.

Response: This is a nice point. We added to the sensitivity analysis on gas price on Page 13 of the original version a new paragraph: “It is also worth noting that the gas-based hydrogen industry may have a sizable effect on the natural gas markets in the U.S., depending on the scale of blue hydrogen production in the future. For example, the production of 10 MMTA hydrogen by SMR with CCS would consume natural gas of 1.9 billion GJ per year, which is equivalent to about 17% of the national industrial natural gas consumption in 2022³³.”

Comment: Both of the points mentioned above are critical to understand the implications of the findings of the analysis in the paper, and such discussion is expected in high impact, interdisciplinary journals like Nat. Comms.

Response: We have added new discussions, data, and clarifications to address the two valuable comments above.

In addition to the two valuable points, we also added to the Discussion section a new paragraph about competing strategies and supportive policy options:

“Competing strategies and supportive policy and regulatory actions should be made rapidly on both the hydrogen demand and supply sides at both federal and state levels in alignment with the innovation expansion. A variety of high-level strategies are needed on the demand side to promote the widespread use of low-carbon hydrogen in industrial, transportation and power sectors and then establish large-sale markets for low-carbon hydrogen. To jumpstart a hydrogen economy, a cluster approach can be employed on the supply side to establish regional production-transportation-demand networks by co-locating feedstock supply, hydrogen production, and carbon sequestration with multiple end-users and by utilizing existing infrastructure, such as pipeline infrastructure for natural gas, CO₂, and H₂ transportation and geological reservoirs for CO₂ storage. To scale the regional hydrogen economy, secured investments in hydrogen production and supporting infrastructure are required with funding from both public and private sectors, plus subsidies and tax incentives. In addition, deploying large hydrogen production plants instead of small ones can improve engineering economics at a plant level. Given the important role of CCS in producing competitive blue hydrogen, continued

support for large-scale demonstration projects should be boosted in the near term to reduce the CCS cost and its uncertainty. Investments in blue hydrogen should be prioritized to lock down sufficient financial resources for the most competitive technologies in the near term. Economic and policy incentives can be tailored with emphasis on gas-based blue hydrogen to catalyze its widespread deployment and technological evolution because of the pronounced cost advantage relative to coal-based blue hydrogen. Extending the 45Q tax credit from the current 12-year period to a longer period for gas-based blue hydrogen projects would remarkably lower the time-related cumulative installed capacity necessary to reach the Hydrogen Energy Earthshot.”

Comment: Thirdly, the authors do not accurately reflect the LCA emissions for blue hydrogen in the paper. The authors provide a range of emissions intensities for blue hydrogen, then state in the supplementary that ‘In this study, all hydrogen production facilities are assumed to be eligible for the bonus rate of tax credit of Section 45V’. However, the NETL report that they cite for emissions data clearly states that the median LCA emissions intensities will exceed the 4 kgCO₂/kgH₂ limit, and hence that blue hydrogen production will not necessarily qualify for the 45V tax credit. The authors never acknowledge this and the LCA emissions are never directly compared to the requirements for the 45V tax credit (which is tucked away in the supplementary in Table 13). This is a crucial oversight at best, and misleading at worst.

Response: It turned out that we did a poor job in presenting the LCA emissions information. As in response to a similar comment from Reviewer 1 on life cycle emissions, we expanded the statement in Page 11 of the original version to read: “As mentioned earlier, the 45V tax credit depends on the life cycle emissions of hydrogen production, which include greenhouse gas emissions from plant stacks, fuel supply, electric power supply, and CO₂ sequestration or management. The life cycle emissions were estimated by the National Energy Technology Laboratory to range from 3.1 to 8.9 kg CO₂-eq/kg H₂ for the gas-based blue hydrogen in the 90% confidence interval between the 5th and 95th percentile values and from 3.4 to 8.9 kg CO₂-eq/kg H₂ for the coal-based blue hydrogen, which is driven mainly by the uncertainty in fuel supply⁷. The largest contributor among the multiple stages to the life cycle emissions is the fuel supply⁷. The median estimate of life cycle emissions is 4.6 kg CO₂-eq/kg H₂ for the gas-based blue hydrogen and 4.1 kg CO₂-eq/kg H₂ for the coal-based blue hydrogen⁷, which is close to the threshold value of 4.0 kg CO₂-eq/kg H₂ required to claim the minimum tax credit for clean hydrogen. Blue hydrogen projects have a fair possibility of earning a 45V tax credit. Thus, the production tax credit for hydrogen projects is assumed to be \$0.6 per kilogram of H₂ for 10 years. This assumption is optimistic for blue hydrogen in this study. However, there is no 45V tax credit if the life cycle emissions of specific blue hydrogen projects are more than 4.0 kg CO₂-eq/kg H₂. See Supplementary Note 5 for additional information about life cycle emissions and tax credits.”

Comment: I recommend that the authors revise the manuscript to take account of the issues mentioned above, and my detailed comments below, before resubmission.

Response: We appreciate the reviewer for these valuable comments. We have revised the manuscript as much as possible to address the detailed comments.

Detailed comments

Comment 1: Too much of the critical information and context is buried in the supplementary. While I recognize that it is necessary to keep to word limits, the authors should provide more of an overview of the system boundaries they consider in the manuscript itself. This includes a discussion of how learning rates are chosen and applied as discussed above. For example, the NETL report assumes that CO₂ and H₂ exit the plants at particular pressures, and take into account the energy and plant required for this. It is important to state these assumptions and clarify why they are chosen.

Response: We added to the beginning of the Results section one new paragraph, which gives an overview of the system analysis: “This study first characterizes greenhouse gas emissions and costs of commercial technologies for blue hydrogen production and then develops technological learning and diffusion models to assess the future costs and evolutionary trajectories of blue hydrogen production without and with tax incentives toward the U.S. Hydrogen Energy Earthshot. A series of parametric analyses are further performed to reveal the dependence of the overall hydrogen production cost on key factors, such as fuel price, CCS cost uncertainties, learning rates, and inflation rate.”

We cannot include many technical details in the main body of this manuscript due to the word count limit. Thus, many details, such as CO₂ and H₂ product streams’ pressure and purity, are included in the supplementary information document. In addition, we are worried about the distraction from the key focus if many small details were included in the main body of this manuscript. To address the reviewer’s concern, we added the suggested information to the relevant paragraph on Page 6 of the original version: “Blue hydrogen plants produce high-purity hydrogen (99.9 vol.%) at the pressure of 6.48 Mpa and transport the captured CO₂ at the pressure of 15.3 Mpa for storage in saline reservoirs, which are typical design conditions.”

As in response to a similar comment on learning rates above, we expanded discussions in the method and result sections on how learning rates are developed and applied to construct a plant-level learning curve.

Comment 2: Autothermal reforming is projected to be a cost-effective blue hydrogen technology in many other studies as well as the IEA estimates. The authors should justify why they chose not to include it – especially since the NETL report provides data on ATR plants.

Response: To clarify it, we added to the first section of the results section on Page 6 of the original version: “The majority of hydrogen produced in the U.S. is made via SMR. In addition, the cost of blue hydrogen produced by SMR with CCS is similar to that by autothermal reforming with CCS but the on-site and life-cycle emissions from the SMR process are less⁷. This study therefore focuses on SMR with CCS for blue hydrogen production.”

Comment 3: The authors should provide hydrogen production capacity in such a way that is can be compared between data sets. Table 1 quotes H₂ production capacity in cubic meters per hour; Table 2 in GW_{th}; and Figures 1-3 in Mt H₂/year. (I note that Table S12 has both Mt/yr and GW_{th}). Given the focus on learning rates as a function of installed capacity, it is useful for the reader to compare these results to the existing capacity. The authors should clearly provide current installed capacities as Mt H₂/y.

Response: The capacity data presented in Table 1 were collected from the International Energy Agency (IEA)’s database. The database reports the annual production capacity just for some specific hydrogen projects. However, no annual capacity is reported for many projects in the

database. To address this comment, we added the annual capacities of those projects to Table 1, dependent on availability of annual capacity data. Below is the expanded table.

Table 1 Operational blue hydrogen production facilities^a.

Feedstock	Project Name	Country	Online Year	H ₂ Production Capacity		Captured CO ₂ (million metric tons/year)
				(m ³ /hour) ^b	(10 ³ · metric tons/year)	
Natural Gas	PCS Nitrogen	United States	2013	31,344	N/A	0.25
	Port Arthur	United States	2013	125,376	118	1.00
	Enid Fertiliser	United States	1982	87,764	N/A	0.70
	Port Jerome	France	2015	12,538	39	0.10
	Quest	Canada	2015	125,376	300	1.00
	Nutrien (Former Agrium) Fertilizer	Canada	2020	37,613	N/A	0.30
	Al Reyadah CCUS	United Arab Emirates	2016	47,876	N/A	0.80
Coal	Sinopec Qilu Petrochemical CCS	China	2022	41,892	N/A	0.70
	Sinopec Zhongyuan Oilfield EOR	China	2015	5,985	N/A	0.10
	Changqing Oil Field EOR	China	2015	2,992	N/A	0.05
	Great Plains Synfuel Plant and Weyburn-Midale	United States	2000	179,536	N/A	3.00
	Coffeyville Fertilizer Plant	United States	2013	125,376	N/A	1.00
Oil	Shell Heavy Residue Gasification CCU - Pernis Refinery ^c	Netherlands	2005	23,938	1,000	0.40
	Karamay Dunhua Oil Technology CCUS EOR Project	China	2015	5,985	N/A	0.10
	North West Sturgeon Refinery	Canada	2020	77,799	N/A	1.30
	Horizon Oil Sands	Canada	2009 ^d	26,212	N/A	0.44

Table 2 reports the initial installed capacity estimates of individual subsystems in GW_{th} as some subsystems are installed in non-hydrogen production plants. In other words, the installed capacity of a given subsystem includes estimates for hydrogen and/or non-hydrogen production plants or energy systems. To estimate the total installed capacity, the installed capacity estimates for hydrogen and non-hydrogen production plants need to be converted to a common metric, which can also be used for comparisons among the multiple subsystems within a hydrogen plant. In addition, this metric is also often used in the literature. We think that the reporting in Table 2 is appropriate. However, to address the reviewer’s concern about current installed capacities in million metric ton per year, we added to the first paragraph on Page 10 the current capacity information: “At a global scale, the initial installed capacity of hydrogen production in 2021 was estimated to be 0.31 MMTA for gas-based blue hydrogen and 0.15 MMTA for coal-based blue hydrogen^{12,28}.”

Comment 4: Blue hydrogen plants tend to be large, so even a large production capacity increase could represent a small number of new plants, and hence limited learnings. Is this taken into account in the learning rate analysis?

Response: No, the technological learning was evaluated in terms of the cumulative installed capacity of blue hydrogen. To make it clearer, we added to “Section Future costs of blue hydrogen production without and with tax credits” on Page 7 of the original version a new sentence: “In addition, the technological learning is evaluated in terms of the cumulative installed capacity of blue hydrogen instead of the number of new hydrogen production plants.”

Comment 5: Given that the aim is to assess the likelihood of reaching the US Hydrogen Earthshot goal of hydrogen at 1 \$/kg by 2030, the authors should comment on how long it would take to reach the installed capacities required to drive the costs down – would this occur within the next 7 years? This is obviously a difficult task, but should be at least mentioned. For example, how many blue hydrogen plants that would need to be deployed? how quickly can blue hydrogen plants typically be deployed? what is the current pipeline of announced projects?

Response: This comment raises a challenging but valuable task, which involves the diffusion of innovation over time. To address it, we collected additional data, developed a diffusion-of-innovation model, and then applied it to estimate the time needed to reach a certain cost target.

We added the newly-developed diffusion-of-innovation model to the Methods section:

“Diffusion-of-Innovation Model

The diffusion of innovation describes how a new technology would spread over time. An S-shaped curve is often used to measure the diffusion over time, in which the adoption rate increases during the early stage, reaches a maximum level at the point of inflection, and decreases until the diffusion curve saturates⁴⁵. To estimate the annual installed capacity of low-carbon hydrogen over time, the S-shaped diffusion function is employed^{46,47}:

$$cc_t = \frac{cc_{sat}}{1 + \frac{cc_{sat} - cc_0}{cc_0} \cdot e^{-r \cdot t}} \quad (1)$$

Where cc_t is the annual installed capacity of low-carbon hydrogen in a particular year t (million metric tons per annum, MMTA); cc_{sat} is the saturation level of annual installed capacity (MMTA); cc_0 is the initial annual installed capacity (MMTA) in the start year; r is the growth rate (fraction); and t is a particular year after the start period. The function coefficients are estimated by regression based on current and future low-carbon hydrogen production capacities through 2030^{12,28,30}. Additional details about the regression and diffusion function are available in Supplementary Note 7. Once the annual installed capacity in future years is determined, the cumulative annual installed capacity can be estimated as a function of time.”

We added the relevant results to the Results section:

“Time-based diffusion of blue hydrogen production

It is helpful for hydrogen energy planning to explore if certain production capacity and cost targets can be achieved by 2030. A new study reports the cumulative installed capacity of low-carbon hydrogen production over time based on globally announced, planning and committed projects through 2030³⁰. A diffusion-of-innovation model was established based on the current and future low-carbon hydrogen capacities through 2030 to explore the time-based diffusion of gas-based blue hydrogen over a long-term planning horizon through 2050.

Fig. 2a shows the cumulative installed capacity estimates for global low-carbon hydrogen production over time. The gas-based blue hydrogen capacity accounts for 49% of the total low-carbon hydrogen capacity given in Table 1 and is estimated to be 90% in 2030 in terms of the International Energy Agency’s hydrogen project databases^{28,31}. Given the changing shares over time, Fig. 2a also shows a range of cumulative installed capacity for gas-based blue hydrogen in a particular year. The cumulative installed capacity of the global gas-based blue hydrogen may range from 6 to 12 MMTA in 2030, which implies that it would be hard for the blue hydrogen production by SMR with CCS alone in the U.S. to reach 10 MMTA in 2030.

Fig. 1c shows the overall plant LCOH as a function of cumulative installed capacity for gas-based blue hydrogen, whereas Fig. 2a show the cumulative installed capacity over time. Combining them together, Fig. 2b shows the overall plant LCOH of gas-based blue hydrogen production without tax credit over time. The result shown in Fig. 2b implies that for the fuel price and learning rates given in the base case, it would also be difficult for gas-based blue hydrogen to reach the ambitious cost target of \$1/kg H₂ by 2030 in normal scenarios without aggressive incentives and game-changing technologies.

Fig. 2 Diffusion of cumulative installed capacity of low-carbon hydrogen and time-based learning curves of gas-based blue hydrogen. (a) Diffusion of cumulative installed capacity; (b) Time-based learning curves of blue hydrogen production cost.

Additional details about the regression and diffusion model were added to the SI:

“Supplementary Note 7: Development of a Time-Based Diffusion Model of Low-Carbon Hydrogen

To estimate the cumulative installed capacity over time, a diffusion-of-innovation model is developed based on the current hydrogen capacity and the future hydrogen capacity by 2030 that includes announced, planning and committed projects. The scatter points shown in Supplementary Fig. 5a represent the installed capacity of global fossil fuels with CCUS for low-carbon hydrogen production in 2021 and the cumulative installed capacity of low-carbon production, which will be produced in 2024–2030. These data were retrieved from the IEA studies and a new study by the Hydrogen Council and McKinsey & Company^{8,9,14}, respectively. The fitting curve was regressed on the scatter points shown in Supplementary Fig. 5a, which is a second-order polynomial function. This fitting curve was then used to calculate the annual installed capacity of low-carbon hydrogen from 2021 to 2029, as shown in Supplementary Fig. 5b. A diffusion model was further formulated by regression on the annual installed capacity

estimates through 2029. Supplementary Fig. 6 shows the diffusion model and the annual installed capacity of global low-carbon hydrogen through 2050.

Supplementary Fig. 5. Cumulative and annual installed capacity of global low-carbon hydrogen over time. (a) Cumulative installed capacity; (b) Annual installed capacity.

Supplementary Fig. 6. Projection of annual installed capacity of global low-carbon hydrogen through 2050.

Comment 6: The authors state that ‘the cost of blue hydrogen produced by SMR with CCS approximates to the Hydrogen Earthshot, as shown in Fig. 1e.’, however, this is difficult to see on the graph. Suggest that the authors add a line to indicate the earth shot cost – as is done in Fig 2 b-d and Fig 3.

Response: As suggested, a dash line was added to all the relevant figures to indicate the Hydrogen Energy Earthshot cost (\$1/kg H₂).

Issues with range of LCA given and discussion around tax credit 45V:

Comment 7: The authors state that “the production tax credit for hydrogen projects is \$0.6 per kilogram of H₂ for 10 years as the life cycle GHG emissions of gas- and coal-based production plants for blue hydrogen are estimated to be 3.13 to 8.86 and 3.40 to 8.87 kg CO₂-equivalent per kilogram of hydrogen, respectively”. However, looking into the NETL report, the median is 4.6 kgCO₂/kgH₂ (Exhibit 3-52, pg 126). The assumption that blue hydrogen plants will get the 45V tax credit is clearly flawed, and the lack of transparency around this is misleading. One way to

get round this would be to ‘look under the hood’ of the NETL analysis and highlight under which conditions blue hydrogen plants could meet the emissions requirements. This would correct the misleading assumption and improve the utility of the analysis in the paper.

Response: As the reviewer suggested, we further looked into the NETL analysis. As in response to one similar comment above, we significantly expanded the life cycle statement to read: “As mentioned earlier, the 45V tax credit depends on the life cycle emissions of hydrogen production, which include greenhouse gas emissions from plant stacks, fuel supply, electric power supply, and CO₂ sequestration or management. The life cycle emissions were estimated by the National Energy Technology Laboratory to range from 3.1 to 8.9 kg CO₂-eq/kg H₂ for the gas-based blue hydrogen in the 90% confidence interval between the 5th and 95th percentile values and from 3.4 to 8.9 kg CO₂-eq/kg H₂ for the coal-based blue hydrogen, which is driven mainly by the uncertainty in fuel supply⁷. The largest contributor among the multiple stages to the life cycle emissions is the fuel supply⁷. The median estimate of life cycle emissions is 4.6 kg CO₂-eq/kg H₂ for the gas-based blue hydrogen and 4.1 kg CO₂-eq/kg H₂ for the coal-based blue hydrogen⁷, which is close to the threshold value of 4.0 kg CO₂-eq/kg H₂ required to claim the minimum tax credit for clean hydrogen. Blue hydrogen projects have a fair possibility of earning a 45V tax credit. Thus, the production tax credit for hydrogen projects is assumed to be \$0.6 per kilogram of H₂ for 10 years. This assumption is optimistic for blue hydrogen in this study. However, there is no 45V tax credit if the life cycle emissions of specific blue hydrogen projects are more than 4.0 kg CO₂-eq/kg H₂. See Supplementary Note 5 for additional information about life cycle emissions and tax credits.”